# The Parkinson's disease drug entacapone disrupts gut microbiome homeostasis via iron sequestration

Fátima C. Pereira [1,2,11] ✉, Xiaowei Ge [3,11], Jannie M. Kristensen [1], Rasmus H. Kirkegaard [1,4], Klara Maritsch[1], Dávid Szamosvári[5], Stefanie Imminger [1], David Seki[1], Juwairiyah B. Shazzad [2], Yifan Zhu[6], Marie Decorte[1], Bela Hausmann [5,7], David Berry [1], Kenneth Wasmund[1,10], Arno Schintlmeister [1], Thomas Böttcher [1,5], Ji-Xin Cheng [3,8] ✉ & Michael Wagner [1,9] ✉

Many human-targeted drugs alter the gut microbiome, leading to implications for host health. However, the mechanisms underlying these effects are not well known. Here we combined quantitative microbiome profiling, long-read metagenomics, stable isotope probing and single-cell chemical imaging to investigate the impact of two widely prescribed drugs on the gut microbiome. Physiologically relevant concentrations of entacapone, a treatment for Parkinson's disease, or loxapine succinate, used to treat schizophrenia, were incubated ex vivo with human faecal samples. Both drugs significantly impact microbial activity, more so than microbial abundance. Mechanistically, entacapone can complex and deplete available iron resulting in gut microbiome composition and function changes. Microbial growth can be rescued by replenishing levels of microbiota-accessible iron. Further, entacapone-induced iron starvation selected for iron-scavenging gut microbiome members encoding antimicrobial resistance and virulence genes. These findings reveal the impact of two under-investigated drugs on whole microbiomes and identify metal sequestration as a mechanism of drug-induced microbiome disturbance.

Drugs initially designed to specifically target human cells can often affect microbes as well[1]. As a result of poor gastrointestinal absorption and/or biliary secretion, many of these drugs reach the large intestine where they encounter and potentially interact with hundreds to thousands of different microbial species that play important roles in various aspects of human physiology[1–3]. Indeed, several cohort studies have reported significant associations between the use of medication and shifts in gut microbial composition and function[4–6]. While the cross-sectional nature of these large cohort studies hinders the establishment of causality, in vitro studies have been pivotal for systematically evaluating direct effects of human-targeted drugs on gut

microbes. A landmark study assessing the antimicrobial effect of 835 human-targeted drugs against a panel of 40 cultured gut microbes revealed that a substantial proportion (24%) of drugs could inhibit the growth of at least one gut bacterial strain in vitro[1]. An additional study testing the effect of a smaller panel of drugs on faecal samples by metaproteomics demonstrated selective anti- and/or promicrobial activity for the great majority of drugs tested, with a considerable fraction of drugs also shifting microbiome composition[7]. Of note, this study demonstrated that bacterial function could shift in response to drugs without a change in taxon abundance, thus highlighting the need for using metrics other than abundance when investigating the

impact of drugs. Importantly, the interaction between the microbiome and drugs is bidirectional, with many studies clearly demonstrating that gut microbes can also actively metabolize[8–10], and under certain circumstances bioaccumulate[11], pharmaceutical drugs. While several studies so far have shed light on the nature and extent of microbe-driven drug transformations[8,10], mechanistic details for human-targeted drug impact on the microbiome remain to be elucidated.

One hypothesis is that drugs can change intestinal microenvironments such as pH or osmolarity, and by doing so, directly affect bacterial growth[12]. Another explanation is that drugs interact with structural analogues of their human targets within bacteria, thus also interfering with cellular processes in microbes[13]. Drug–microbiome interactions have been shown to modulate the therapeutic effect of drugs, contribute to their side effects, or both[14–17]. Importantly, in certain cases, drug-induced microbiome changes might also contribute to other diseases. For instance, proton pump inhibitors (PPIs) cause major shifts in the gut microbiome[18], leading to decreased resistance to colonization by enteric pathogens[19]. Among the human-targeted drugs tested in vitro, compounds that target the nervous system, such as antipsychotics and antidepressants, seem to exhibit stronger anti-commensal activity against gut bacteria compared with other tested drug classes[1,20,21]. This is concerning, given the growing number of studies implicating the microbiome in many neuropsychiatric disorders[22] and the widespread and rising use of this class of pharmaceuticals worldwide[23]. Thus, a better mechanistic understanding of drug–microbiome interactions in the context of nervous system-targeted medications may facilitate innovative ways to improve efficacy and/or minimize side effects of therapies for such disorders.

Here we investigate the effects of two nervous system-targeted drugs on whole gut microbiomes using a suite of complementary functional microbiome approaches. These aimed at investigating the effects of the drugs in the context of whole microbial communities, which more closely resembles their effects in situ, and to therefore examine most key members of the gut microbiome. We studied two commonly used drugs: (1) entacapone, a catechol-O-methyltransferase (COMT) inhibitor that acts by preventing the degradation of levodopa, the main drug used in the treatment of Parkinson's disease[24]; and (2) loxapine succinate, a tricyclic antipsychotic medication primarily used in the treatment of schizophrenia[25]. These drugs, whose yearly total prescription exceeds 30 million tablets in the United States alone[26], belong to different therapeutic classes and were shown to be selective in their anti-commensal activity against a panel of 40 cultured gut strains, with entacapone predominantly targeting Gram-positive organisms of the phylum Firmicutes and loxapine succinate targeting only taxa within the Gram-negative order Bacteroidales[1]. We demonstrate that the impact of either drug on microbiome composition and/or activity extends beyond the taxa initially detected in vitro in pure culture, with entacapone causing strong shifts in microbiome composition. This prompted us to look for the cause of these shifts. We identified microbial iron deprivation, driven by the ability of entacapone to complex iron, as the main mechanism behind entacapone's strong modulatory effect.

## Results

### Nervous system-targeted drugs affect the microbiome

To evaluate the impact of entacapone (ENT) and loxapine succinate (LOX) on whole gut microbiome communities, we incubated freshly collected faecal samples from six healthy adult individuals with two drug concentrations, a low and a high (Hi) concentration (Fig. 1a and Methods). Supplementation of ENT-Hi significantly reduced the numbers of microbial cells over time when compared with all other tested conditions (Fig. 1b and Supplementary Table 1). Major shifts in the microbial community composition, as determined by 16S ribosomal RNA gene amplicon sequencing analyses, were detected in response to ENT-Hi, LOX-Hi and LOX-Low treatments (Fig. 1c), and in the case

of ENT-Hi, these were also accompanied by shifts in alpha diversity (Extended Data Fig. 1a,b, and Supplementary Tables 2 and 3).

By integrating total microbial counts with 16S rRNA gene amplicon sequencing data[27], we determined absolute abundances for all detected taxa (Fig. 1d). Importantly, absolute abundance data confirmed that the employed incubation conditions enabled an increase in abundance for nearly all the taxa initially present in faecal samples (Fig. 1e: No drug versus inoculum; Supplementary Table 4) and thus allowed tracking of drug-induced activity and abundance changes of microbiome members. Absolute abundances of genera such as *Bacteroides* or *Clostridium sensu stricto 1* decreased in both ENT-Hi and LOX-Hi samples compared with the dimethylsulfoxide (DMSO) control (Fig. 1d). However, many of the detected effects were drug specific, with ENT-Hi decreasing and LOX-Hi increasing total abundances of the genera *Anaerostipes*, *Fusicatenibacter*, *Ruminococcus torques* group, *Eubacterium hallii* group, *Erysipelotrichaceae* group UG-003 and *Roseburia*. Abundances of several genera were significantly altered in response to ENT-Hi only: *Escherichia-Shigella* and *Ruminococcus* increased in abundance, and genera such as *Alistipes*, *Streptococcus* or *Blautia* decreased (Fig. 1d,e and Supplementary Table 5). A similar impact of ENT-Hi on microbial biomass accumulation and on the overall community composition could also be observed when we used a different, nutrient-rich medium (BHI) for the incubations with the same six donors (Extended Data Fig. 1c–g) as well as in additional incubations set up using samples from three additional individual donors (Extended Data Fig. 2 and Supplementary Table 6).

Differential abundance analysis indicated that the abundance of 29.4% of all 16S rRNA gene amplicon sequencing variants (ASVs) were significantly impacted by ENT-Hi after 24 h of incubation, 11.8% of ASVs were impacted by LOX-Hi, and only 3.6% and 6.0% of ASVs were impacted by ENT-Low and LOX-Low, respectively (Methods and Supplementary Table 5). Interestingly, LOX-Low resulted in growth inhibition patterns that differ from the ones observed when the same drug was supplemented to gut members grown in pure culture[1], where it only specifically inhibited growth of Bacteroidales strains. Our results show that other Gram-negative and several Gram-positive species are affected by LOX-Low in the context of whole microbiome communities. These include Erysipelotrichaceae spp., Oscillospiraceae spp. and Lachnospiraceae spp., suggesting a cross-sensitization to loxapine in the context of the community. For ENT-Low, only very few organisms were significantly impacted, most of which were Firmicutes, which agrees with previous reports[1].

Using long-read metagenomic sequencing of the starting faecal sample material, we retrieved a total of 1,049 metagenome-assembled genomes (MAGs) (Supplementary Tables 7 and 8). By linking ASVs to MAGs that contained 16S rRNA genes, we could follow drug-driven community shifts at a higher taxonomic resolution (Fig. 1e,f). This revealed that ASVs classified as *Escherichia* and *Ruminococcus* taxa thriving in ENT-Hi give exact hits to the 16S rRNA genes in genomes of *E. coli* and *R. bromii*. These analyses also indicated that LOX-Hi conditions selectively promoted the growth of taxa such as *Mediterraneibacter faecis*, *Faecalibacillus faecis* and *Blautia_A*. Species such as *Clostridium* sp900539375 (*Clostridium sensu stricto* 1 based on SILVA taxonomy) and several *Bacteroides* species (*B. uniformis*, *B. stercoris*, *Phocaeicola dorei* (formerly *B. dorei*) and *P. vulgatus* (formely *B. vulgatus*)) were totally or partially inhibited by the presence of high concentrations of either drug, but the effect of ENT-Hi is much more pronounced than that of LOX-Hi (Fig. 1e,f). On the other hand, *Prevotella* sp900546535 and several Gram-positive organisms such as *Streptococcus lutetiensis*, *Collinsella* sp003487125, *Bifidobacterium longum*, *Mediterraneibacter faecis* or *Faecalibacillus faecis* showed growth inhibition by ENT-Hi only (Fig. 1e,f). All together, these results reveal a strong but distinct effect of entacapone and loxapine at physiological concentrations on microbiota composition and abundance in the short term, with entacapone having a much more pronounced effect than loxapine.

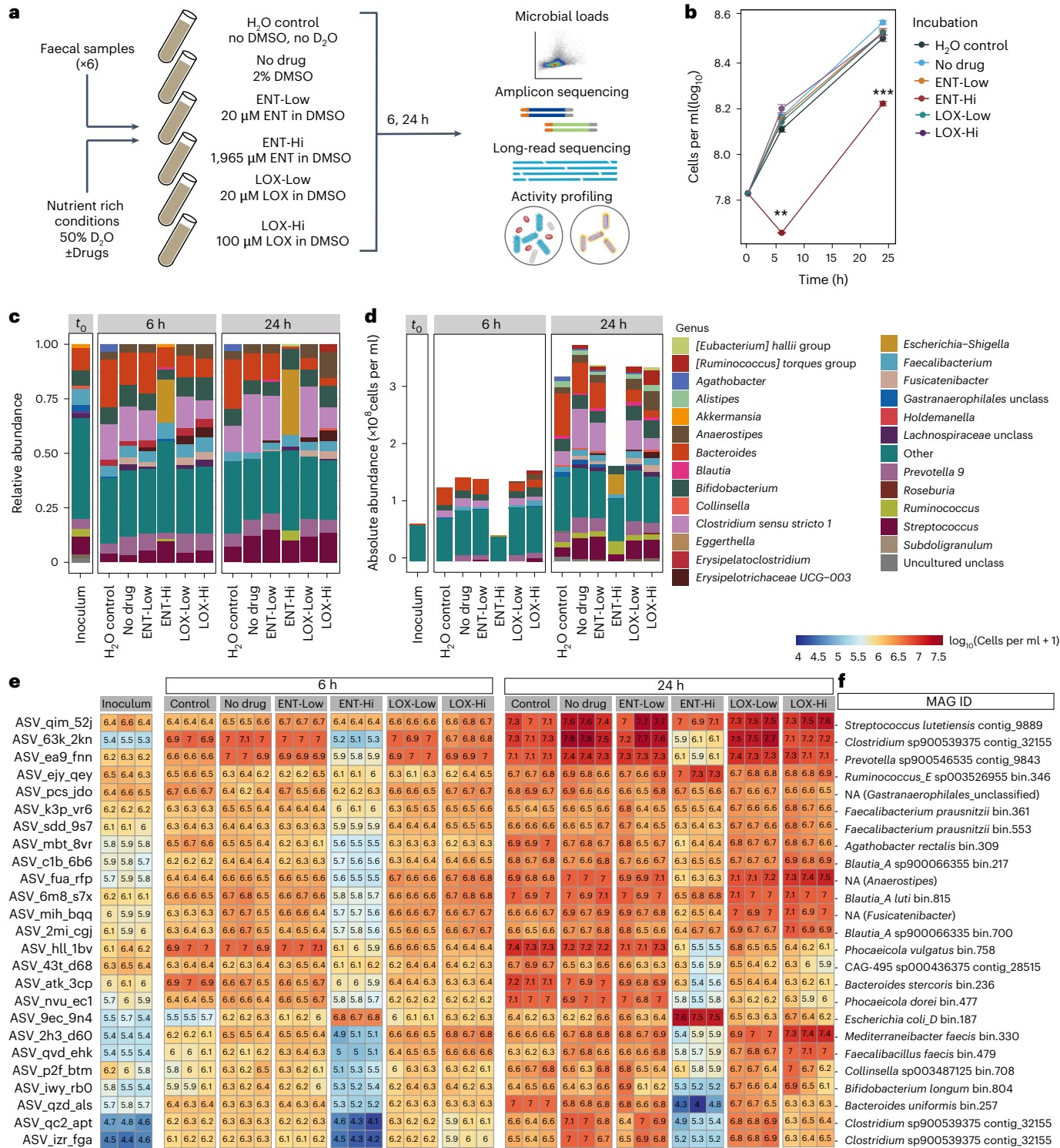

**Fig. 1 | Drug supplementation of faecal samples impacts biomass accumulation and microbiota composition. a**, Schematic representation of faecal sample incubations with the drugs entacapone (ENT) and loxapine (LOX). Samples from 6 different donors were mixed and supplemented with 2 different concentrations of each drug (Low and Hi). The $H_2O$ control consisted of medium (in $H_2O$) without $D_2O$ or DMSO. After 6 or 24 h of incubation, samples were collected for further analyses (Methods). All incubations were performed in triplicates. **b**, Total microbial cell loads in faecal sample incubations described in **a** at the start of the incubation (time = 0 h) and after 6 and 24 h, in sM9 medium. Total cell loads were assessed by flow cytometry and are presented in Supplementary Table 1. **P = 0.006, ***P = 0.00015, unpaired two-sided *t*-test with 'No drug' used as the reference. Data represent the mean ± s.e.m. of 3 incubation vials per condition. **c,d**, Relative (**c**) and absolute (**d**) genus abundance profiles of faecal microbial communities incubated

for 6 or 24 h as described in **b** and assessed by 16S rRNA gene amplicon sequencing. Relative and absolute abundance data are presented in Supplementary Tables 3 and 4, respectively. The community composition at 0 h (inoculum) is also shown. All genera present at a relative abundance below 0.025 or absolute abundance below 6 × 10^6 cells per ml were assigned to the category 'Other'. Unclass, unclassified. Each bar represents the mean of triplicates. **e**, Heat map showing the absolute abundance, expressed in $\log_{10}$(cells per ml + 1), of the 25 most abundant ASVs detected across all samples. Each column displays data from one replicate. **f**, MAGs with 16S rRNA gene sequences matching ASVs shown in **e**. MAGs were retrieved by metagenomic sequencing of the initial faecal samples using Oxford Nanopore sequencing (Methods and Supplementary Table 8). NA indicates no match of the ASV sequence to each MAG 16S rRNA gene sequence.

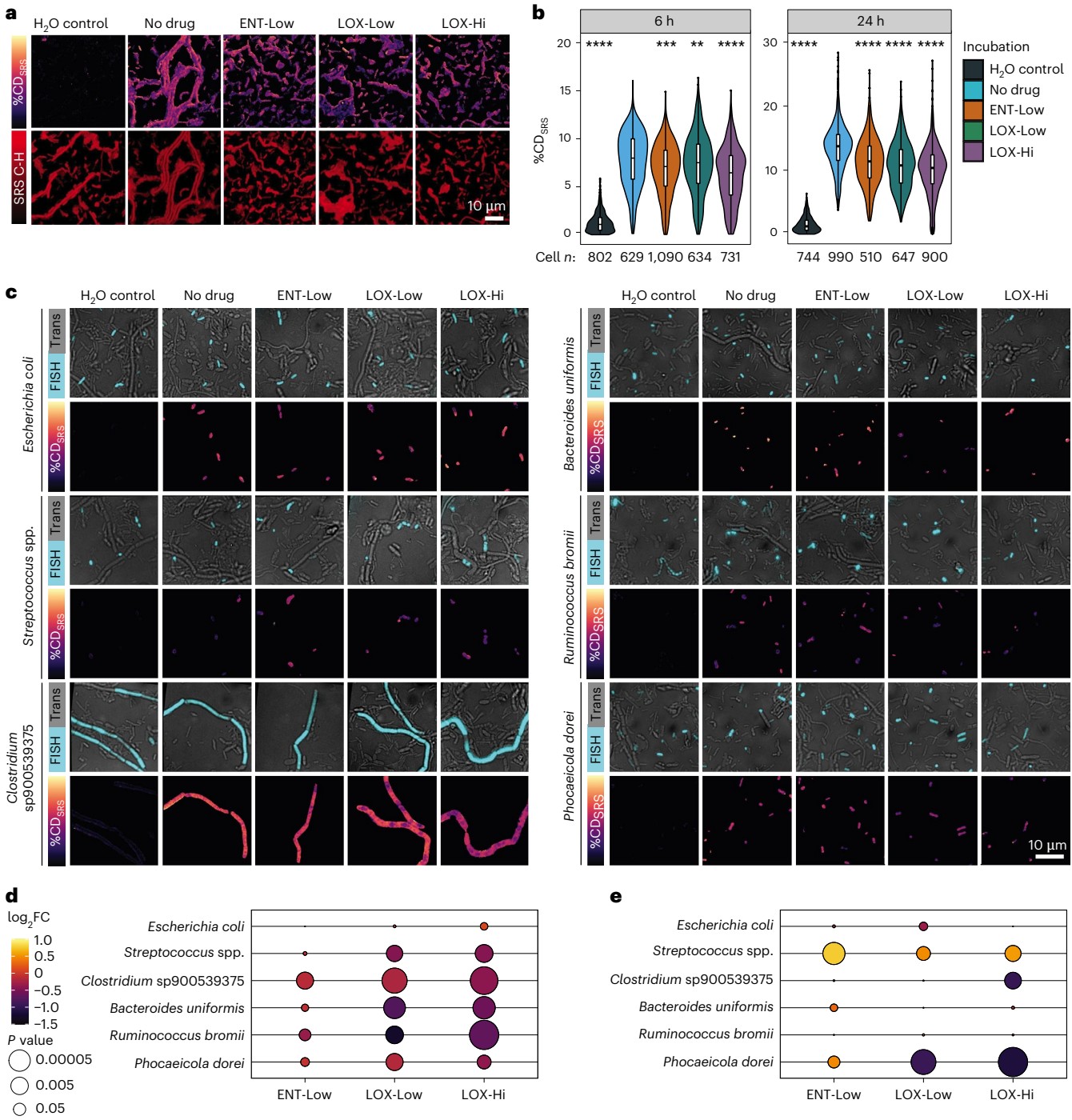

**Fig. 2 | Metabolic activity of drug-incubated single-microbiome cells measured by deuterium incorporation via SRS. a**, Representative SRS images of a mixture of 6 faecal samples incubated with the indicated drugs for 6 h, as depicted in Fig. 1. Top: %CD$_{SRS}$ distribution images. Bottom: C-H for biomass visualization (log scaled). %CD$_{SRS}$ scaling: minimum 0, maximum 20%. **b**, Single-cell %CD$_{SRS}$ values in each analysed sample (see also Supplementary Table 9). Violin plots illustrate summary statistics (median, first and third quartiles, with whiskers representing the minimum and maximum values within 1.5× the interquartile ranges from the first and third quartiles). Number of measured cells per sample are indicated on the *x* axis (cell *n*). **\*\**$P = 0.0043$, **\*\*\***$P = 5.2 × 10^{-12}$, **\*\*\*\***$P < 2 × 10^{-16}$, two-sided Wilcoxon test using 'No drug' as the reference. **c**, Drug-incubated faecal samples (at 6 h) were hybridized with fluorescently labelled oligonucleotide probes targeting *E. coli* (Ecol_268), *Streptococcus* and *Lactococcus* species (Strc_493), *Clostridium* sp900539375 (*Clostridium sensu stricto 1*, Clo_648), *Ruminococcus bromii* (Brom_2036), *B. uniformis* (Buni_1001)

and *P. dorei* (Bado_374) (Supplementary Table 10). For each targeted group, top rows show representative images obtained by overlay of transmitted light (Trans) and fluorescence intensity (FISH). Bottom rows show the corresponding SRS images (displaying %CD$_{SRS}$) for the FISH-targeted microbes (%CD$_{SRS}$ values of other microbes are not displayed for the sake of visibility). Scale bar, 10 μm. For **a** and **c**, we reproducibly detected analogous differences when measuring at least one additional batch of samples. **d**, Bubble plot denoting the fold change (FC, represented as log$_2$FC) in activity levels (calculated as %CD$_{SRS}$) for the taxa targeted by FISH and incubated with drugs relative to 'No drug' incubations at 6 h (Supplementary Table 11). **e**, Bubble plot denoting the FC in absolute abundances for the taxa targeted by FISH and incubated with drugs relative to 'No drug' incubations, as determined by DeSeq2 at 6 h (Supplementary Table 5). In **d** and **e**, *P* values were obtained using the Wald test and corrected for multiple testing using the Benjamini–Hochberg method.

## Nervous system-targeted drugs alter microbial metabolism

Next, we evaluated the effect of these drugs on microbial activity at the single-cell level. To determine drug-induced changes in microbial activity, we added heavy water ($D_2O$) as a universal metabolic tracer[28,29] to our incubations (Fig. 1a and Methods). Detection and quantification of C-D bonds from deuterium (D) incorporation from $D_2O$ by single microbial cells can be achieved using stimulated Raman scattering spectroscopy (SRS)[30,31]. We have previously successfully combined SRS with fluorescence in situ hybridization (FISH) to determine gut microbiome response to sugars with high throughput[32]. Using an optimized SRS–FISH platform that provides even higher throughput and sensitivity than the previous setup (Supplementary Text, and Extended Data Figs. 3 and 4), we measured ~8,000 individual microbiota cells after 6 and 24 h of incubation in the presence of the drugs and in controls (Fig. 2a,b). An unexpected, non-vibrational signal was detected in samples incubated with ENT-Hi, so these samples were excluded from activity measurements and the origin of this signal was further explored as detailed in the next section. As expected, in the absence of the drugs, nearly all of the analysed cells were detected as active in the incubation medium, with 90% and 98% of cells displaying %CD$_{SRS}$ values above threshold after 6 h and 24 h of incubation, respectively (Fig. 2b and Supplementary Table 9). Addition of either ENT-Low, LOX-Low or LOX-Hi resulted in a significantly reduced fraction of total active cells, as well as in a significant reduction of single-cell metabolic activity. This reduction was more pronounced for LOX-Hi, followed by LOX-Low and ENT-Low (Fig. 2b). Thus, these drugs clearly impacted microbial activity within communities, even after short incubation times and under conditions where no impacts on their overall abundances were detected (Fig. 1b).

To examine the activity of specific populations of the microbiome within the complex communities, we targeted six distinct abundant taxa of interest using SRS–FISH, which enabled us to determine the activity of individual cells from these taxa (Supplementary Table 10). Targeted taxa included organisms whose abundances were both negatively and positively impacted by drugs. Both LOX-Low and LOX-Hi incubations significantly reduced the activity of all targeted gut microbiota members except for *E. coli* (Fig. 2d). Interestingly, *Clostridium* sp900539375, *B. uniformis*, *Ruminococcus bromii* and *P. dorei* showed reduced activity in LOX treatments, but only the abundance of *P. dorei* and *Clostridium* sp900539375 was negatively impacted (Fig. 2d,e and Supplementary Table 11). LOX-Low and LOX-Hi seem to strongly inhibit *P. dorei* growth within 6 h (Fig. 1e), probably rendering most cells of this taxon undetectable by FISH, as the low ribosome content of non-active cells hinders FISH detection. We speculate that only a few drug-resistant *P. dorei* cells remained active enough to be detected by FISH, and these cells were not strongly impacted in activity (Fig. 2c,d). This observation could be explained by either phenotypic heterogeneity[33] within the *P. dorei* isogenic population, or the FISH probe used, which may target multiple *P. dorei* strains with different physiologies. A comparable decrease in activity was detected for *Streptococcus* spp., but in this case the decrease was surprisingly accompanied by a slight increase in abundance at 6 h (Fig. 2d,e). However, from 6 to 24 h, we detected a decrease of the population represented by the *Clostridium* sp900539375, *B. uniformis* and *Streptococcus lutentiensis* MAGs in the presence of LOX compared with the control (Fig. 1e,f and Extended Data Fig. 5), which could be an effect of the lower activities detected by SRS in the LOX conditions at 6 h. In summary, SRS–FISH provided unique insights into the impact of ENT and LOX on the activities of specific microbiome members, which are often masked when only considering abundance data from relatively short incubation experiments.

## Entacapone bioaccumulates in microbiota cells

Gut microbes have been shown to bioaccumulate some drugs, leading to depletion of the drug from the surrounding environment[11]. As ENT-Hi samples showed a strong, unspecific Raman signal during SRS pump-probe detection, we explored the origin of this signal and concluded that it is photothermal (PT), originating from entacapone bioaccumulation within microbial cells (Supplementary Text and Extended Data Fig. 6). By mapping the intensity of this PT signal from ENT-Hi samples and controls (Fig. 3a), we were able to show that the signal from entacapone occurred in a large fraction of microbial cells that had been incubated with the drug and washed before being fixed for analysis, but only in very few cells that had been fixed before incubation with the drug (that is, dead cells) (Fig. 3b,c and Supplementary Table 12). There was no strong correlation between entacapone accumulation and cellular activity when measured in the same cell (Extended Data Fig. 7), although the cellular activity tends to be lower for high entacapone accumulators (Fig. 3d).

To identify the main drug-accumulating taxa, we further looked at the entacapone distribution among populations targeted with the FISH probes described above (Fig. 3e). All of the targeted populations displayed entacapone signals to different intensities. While *P. dorei* and, to a lesser extent *Streptococcus* spp. and *E. coli*, seemed to be strong entacapone accumulators, only a small percentage of *R. bromii* and part of the *B. uniformis* population accumulated entacapone (Fig. 3e,f). High entacapone signals were also detected in cells not targeted by FISH (Fig. 3e,f). We further confirmed the ability of *P. dorei* to accumulate entacapone when grown in pure culture (Extended Data Fig. 8 and Supplementary Text). Interestingly, while entacapone accumulation drastically inhibited the growth of *P. dorei* as a microbiome community member and in pure culture, it did not affect *Streptococcus* growth in the community to the same extent and even showed growth promotion for *E. coli* (Fig. 3f,g). Thus, bioaccumulation of the drug correlated with growth inhibition for certain taxa, whereas others were unaffected or even stimulated in growth.

## Entacapone chelates iron and induces iron starvation

Entacapone's nitrocatechol group can act as a bidentate ligand, chelating and forming stable tris complexes with the transition metal ion ferric iron (Fe(III)), through the catecholate oxygen atoms[34]. Using an assay widely applied to detect Fe-chelating agents in solution, we confirmed entacapone's ability to chelate ferric iron (Fe(III)) (Fig. 4a). The stability constant of entacapone's association with Fe (pFe(III) = 19.3)[34] has been demonstrated to be similar to constants described for other known iron chelators such as 2,2′-bipyridyl (pFe(III) = 21.5)[35]. Entacapone was also predicted to complex ferrous iron (Fe(II)) with rather high affinity, but was not predicted to form strong complexes with any other metal cations[35]. However, we demonstrate that entacapone does not bind ferrous iron (Extended Data Fig. 9a).

Iron is a limiting nutrient in the gut and essential for the growth of most gut microbes[36]. Iron concentrations in stool are estimated to be ~60 µM (ref. 37), and the sM9 medium used here for faecal sample incubations contained similar concentrations of iron (31 µM). As the estimated concentration of entacapone in the large intestine is approximately two orders of magnitude higher (1,965 µM), we postulated that the inhibitory effect of ENT-Hi on microbial growth could be directly related to its ability to deprive gut microbes of iron via chelation, similarly to what has been documented for other Fe(II) and Fe(III) chelators[38,39]. To test this hypothesis, we grew *Bacteroides thetaiotaomicron*, a gut commensal severely impacted by Ent-Hi (log$_2$-fold change: −4.16, adjusted $P$ = 0.016, Supplementary Table 5: ASV_g1t_iba) in the presence of ENT-Hi or Fe-loaded ENT-Hi (Fig. 4b and Methods). Of note, in pre-complexation reactions, Fe(II) from FeSO$_4$ was exposed to the drug solvent DMSO, which resulted in an instantaneous oxidation to Fe(III) (Extended Data Fig. 9b). Thus, both pre-complexation of ENT-Hi with FeCl$_3$ or FeSO$_4$ resulted in Fe(III)-loaded entacapone. While ENT-Hi inhibited the growth of *B. thetaiotaomicron*, ENT-Hi pre-complexed with iron did not, enabling *B. thetaiotaomicron* to grow normally (Fig. 4b). This is because iron-saturated chelators are no longer able to complex free Fe(II) (or Fe(III)) present in the medium

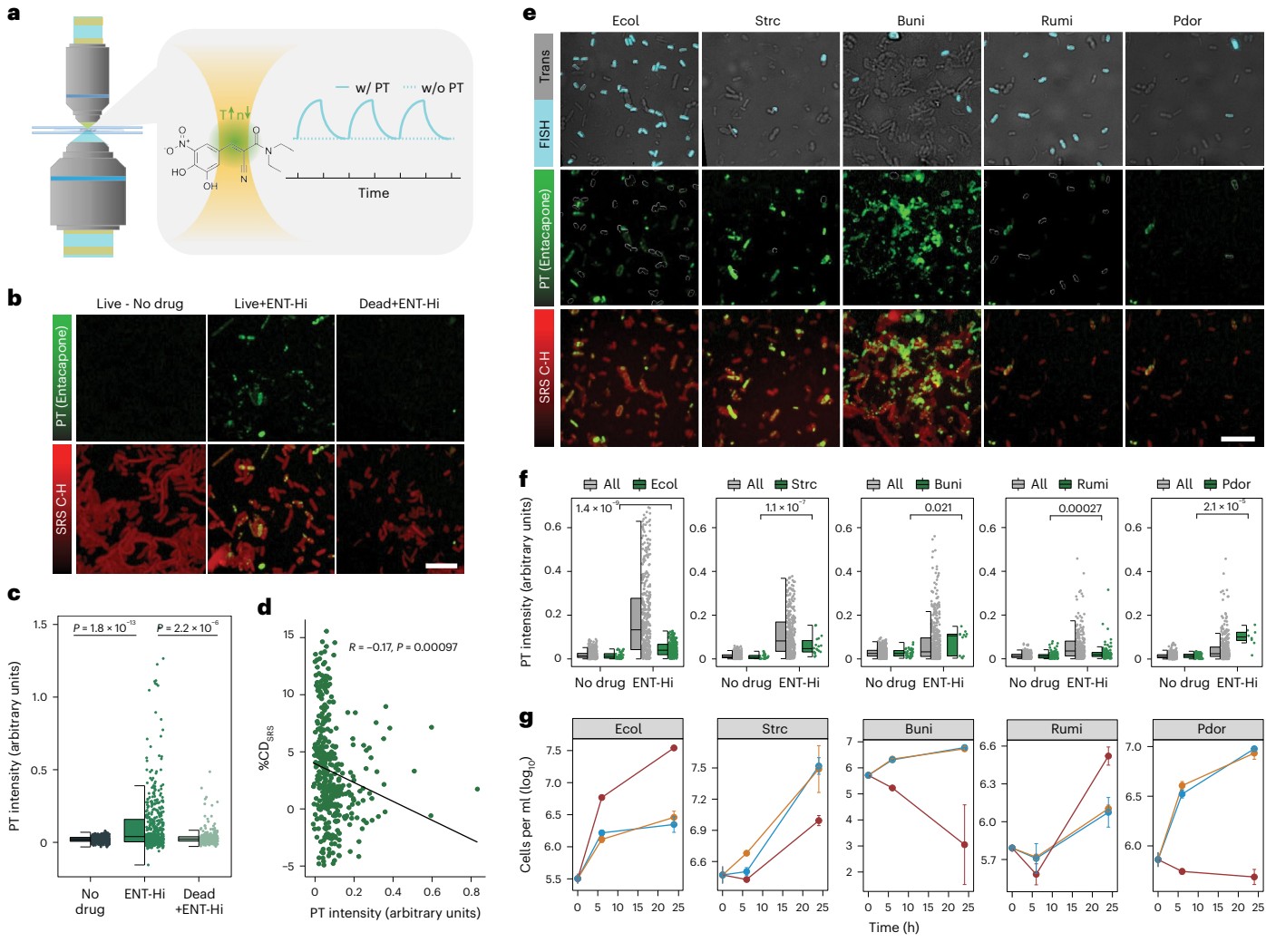

**Fig. 3 | Photothermal imaging of entacapone bioaccumulation by microbiota cells. a**, Schematic illustration of the time-dependent signal obtained from a solution of 10 mM entacapone in DMSO, with photothermal (w/ PT) and without photothermal (w/o PT) detection. **b**, Photothermal signal intensity distribution from entacapone (PT), and SRS C-H signal of live and dead (PFA fixed) microbiota cells incubated in the presence (+) or absence (−) of ENT-Hi for 6 h. Scale bar, 10 μm. **c**, Single-cell photothermal signal intensity distribution in samples shown in **b**. *P* values were determined using unpaired two-sided Wilcoxon test comparing groups ENT-Hi and No drug, or ENT-Hi and Dead+ENT-Hi (Supplementary Table 12). **d**, Correlation between activity (%CD_SRS) and PT signal intensity for microbiota cells exposed to ENT-Hi for 6 h. Pearson correlation coefficient (*R*) and two-sided *P* value are indicated. Incubations were established using a new mixture of samples from two donors in sM9. **e**, Representative images of the original mixture of 6 faecal samples incubated with ENT-Hi for 6 h followed by hybridization with FISH

probes targeting *E. coli* (Ecol), *Streptococcus* and *Lactococcus* (Strc), *Ruminococcus bromii* (Rumi), *B. uniformis* (Buni) and *P. dorei* (Pdor) (Supplementary Table 10). Top: FISH-probe signals from hybridized cells. Middle: PT signal maps from entacapone (cells targeted by FISH are shown with white contour lines). Bottom: entacapone PT signals overlayed with SRS C-H signals. Scale bar, 8 μm. In **b** and **e**, PT channel contrast is min 0 and max 1.8, and C-H signals are represented on a log scale. **f**, Single-cell photothermal signal distribution in samples shown in **c**. *P* values were determined using unpaired two-sided Wilcoxon test comparing ENT-Hi and No drug groups of targeted taxa only. **g**, Absolute abundance of taxa targeted by FISH in No drug, Ent-Low and ENT-Hi incubations (Supplementary Table 4). Data represent the mean ± s.d. of 3 incubation vials. In **c** and **f**, boxes represent the median, first and third quartile. Whiskers extend to the highest and lowest values that are within 1.5× the interquartile range.

(or intracellularly), thus enabling bacteria to access iron and grow normally[39]. Originally, we expected added iron to be mostly present in the lower oxidation state under anaerobic conditions. Surprisingly, quantification of Fe(II) in FeSO$_4$-supplemented medium revealed that the Fe(II) added was quickly oxidized to Fe(III), probably by the presence of phosphates in the medium[40], despite the anaerobic atmosphere and the presence of 0.1% (w/v) L-cystein as a reducing agent (Extended Data Fig. 9c). Supplementation of ENT-Hi alone followed by addition of increasing amounts of iron (FeSO$_4$) alleviated the inhibitory effect of ENT-Hi on *B. thetaiotaomicron* (Fig. 4b). Addition of similar amounts of cupric ions (Cu(II)) did not reverse growth inhibition by ENT-Hi (Fig. 4b), confirming that the observed effect is specific for iron.

In summary, these results demonstrate that supplementation of Fe(III) alone or of an ENT-Hi:Fe complex rescues the inhibitory effect of ENT-Hi on *B. thetaiotaomicron*, strongly indicating that entacapone drives iron limitation.

To determine whether the results above also apply to a complex microbiota, we incubated faecal samples under equivalent conditions as described in Fig. 1a. In agreement with the results obtained for *B. thetaiotaomicron* alone, supplementation of the whole microbiome with iron-loaded ENT-Hi or with ENT-Hi followed by iron resulted in a complete or near complete reversal of the inhibitory effect on microbial biomass accumulation (Fig. 4c,d and Supplementary Table 13). Supplementation of Fe(III) reverses the impact of ENT-Hi on the community

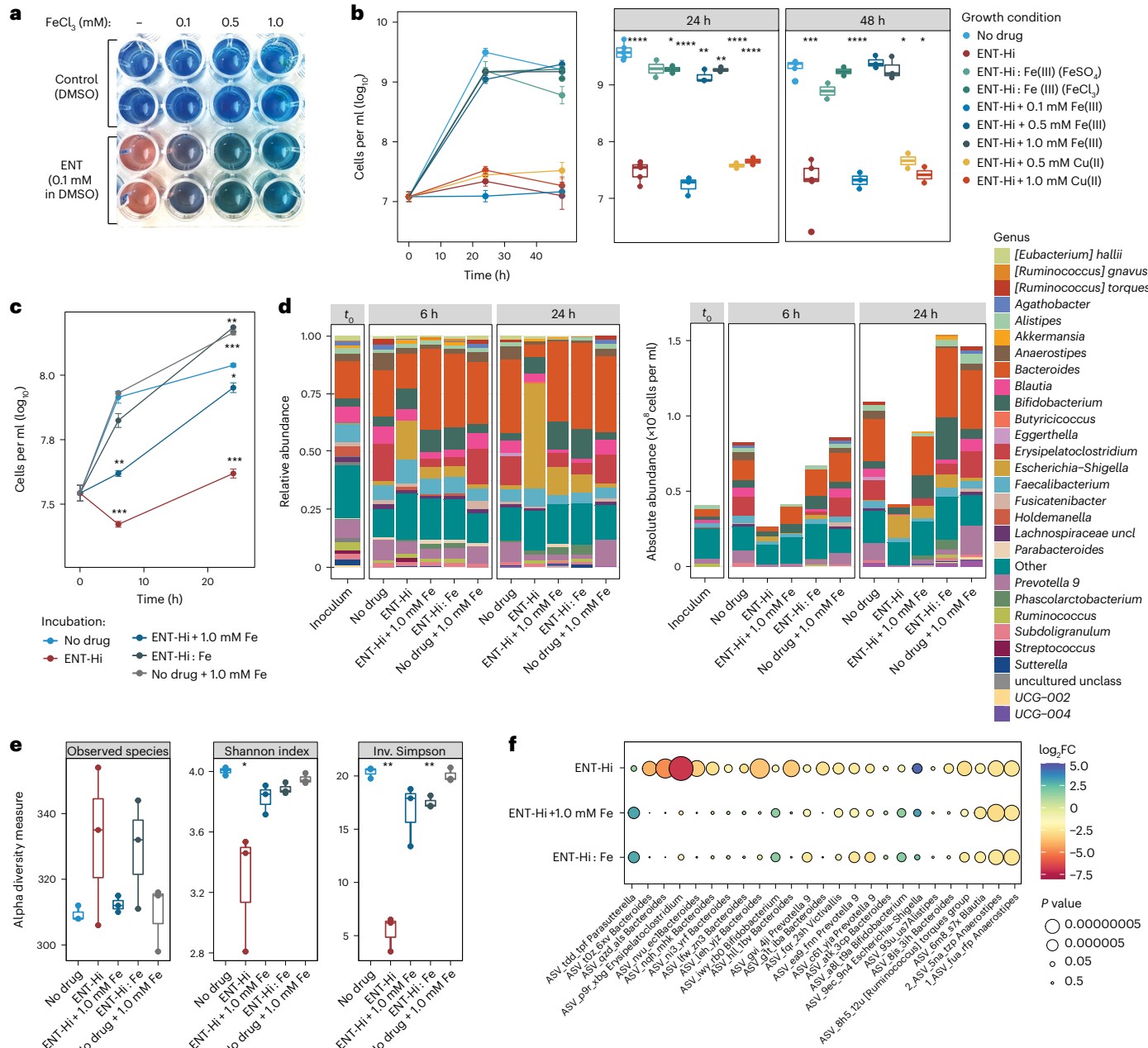

**Fig. 4 | Iron supplementation rescues the impact of entacapone on the gut microbiome. a**, Siderophore detection assay showing the change in colour in the presence of 0.1 mM entacapone. The indicator complex changed back to its original colour (blue) after addition of excess ferric iron. **b**, Growth of *B. thetaiotaomicron* in the absence of drug or in the presence of: ENT-Hi; ENT-Hi preloaded with $FeSO_4$ (ENT-Hi : Fe(III)), ENT-Hi preloaded with $FeCl_3$ (ENT-Hi : Fe(III)); or ENT-Hi supplemented with Fe(III) ($FeSO_4$) or Cu(II) ($CuCl_2$). No drug and ENT-Hi conditions: $n = 6$; all other conditions: $n = 3$ independent growths per condition. Data represent the mean ± s.e.m. The box plots in the middle and right refer to the same data as displayed on the left plot but split by incubation time. **c**, Total cell loads in faecal samples (mix from 5 donors) incubated with either ENT-Hi, $FeSO_4$ (No drug + 1.0 mM Fe), both (ENT-Hi + 1.0 mM Fe), or ENT-Hi pre-incubated with $FeSO_4$ (ENT-Hi:Fe) in sM9 medium. In **b** and **c**, Fe(II) is oxidized to Fe(III), and thus Fe(III) is present. Data represent the mean ± s.e.m.

of 3 incubation vials per condition (Supplementary Table 13). **d**, Relative (left) and absolute (right) genus abundance profiles of microbial communities incubated as described in **c** and assessed by 16S rRNA gene amplicon sequencing (Supplementary Tables 14 and 15). All genera present at relative abundances below 0.01 or absolute abundances below $1.5 \times 10^6$ cells per ml were assigned to the category 'Other'. Data are from triplicates per condition. **e**, Alpha diversity metrics in gut microbial communities at 24 h of incubation. **f**, Bubble plot denoting the fold change ($log_2FC$) in absolute abundance relative to control incubations for the 25 most abundant ASVs. *P* values (Wald test) were adjusted using the Benjamini–Hochberg method. In **b** and **e**, boxes represent the median, first and third quartiles. Whiskers extend to the highest and lowest values that are within 1.5× the interquartile range. In **b**, **c** and **e**: *$P < 0.05$, **$P < 0.01$, ***$P < 0.001$, ****$P < 0.0001$; unpaired two-sided *t*-test with 'No drug' used as the reference group. Only statistically significant differences are indicated.

alpha diversity metrics as well as on the growth of nearly all of the top 25 most affected taxa (exceptions are Firmicutes such as *Anaerostipes* and *Blautia*) (Fig. 4e,f, and Supplementary Tables 14 and 15). Iron supplementation does not only enable taxa negatively impacted

by entacapone to grow, but it also seems to restrict the accelerated expansion of organisms such as *E. coli*, prompted by ENT-Hi (Fig. 1e and Fig. 5d). Importantly, we confirmed that this effect is not due to lower cellular uptake, as ENT-Hi:Fe(III) bioaccumulates in microbiota

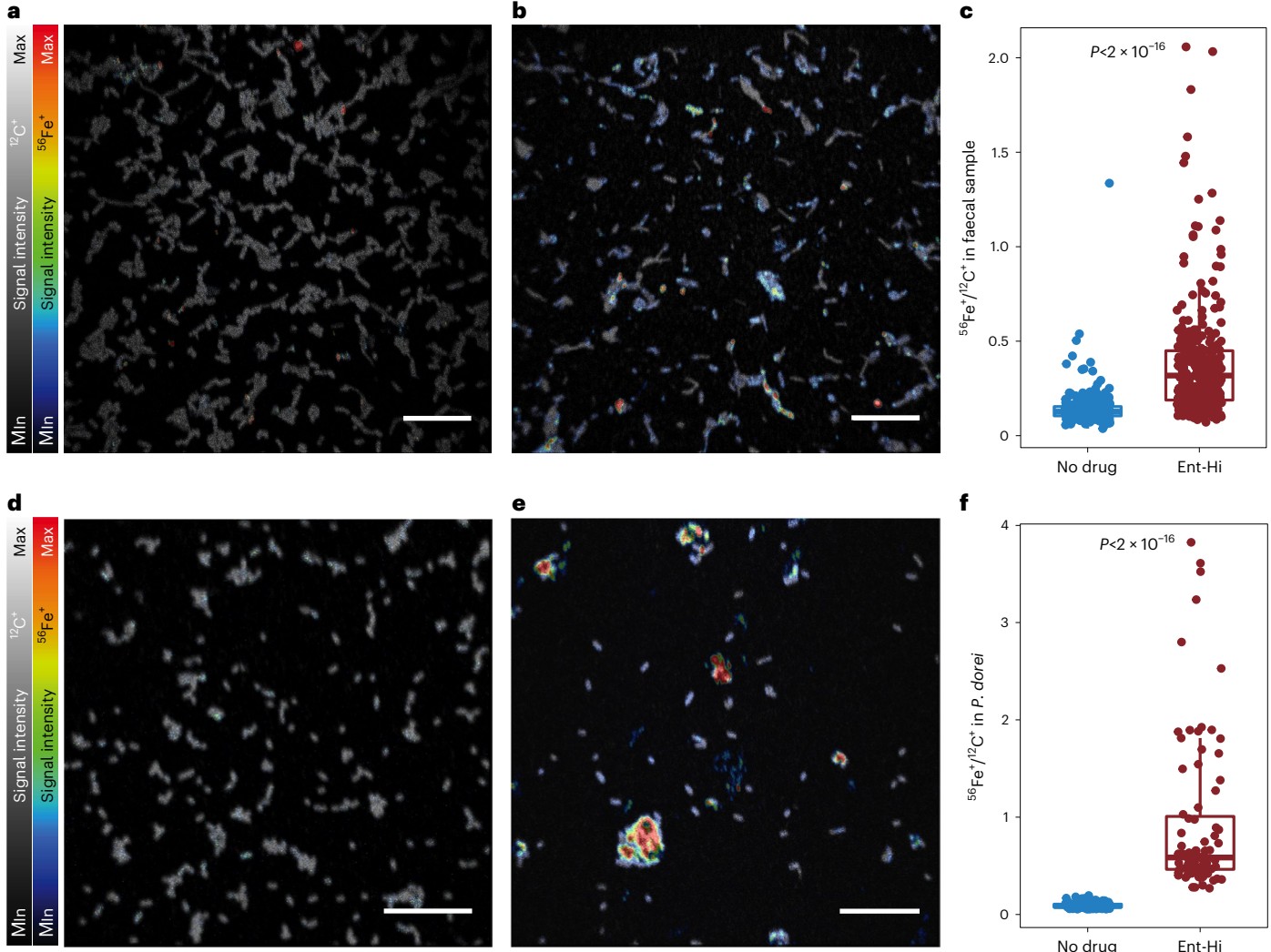

**Fig. 5 | Entacapone-exposed microbiota cells display higher iron content.**
**a,b**, NanoSIMS overlay images of the $^{56}Fe^+$ (colour scale) and $^{12}C^+$ (grey scale) signal intensities of a faecal sample incubated for 6 h without drug (**a**) or ENT-Hi (**b**) in sM9 medium. Scale bars, 10 μm. **c**, Evaluation of $^{56}Fe^+/^{12}C^+$ secondary ion signal intensity ratios in individual cells present in faecal samples. **d,e**, NanoSIMS overlay images of the $^{56}Fe^+$ (colour scale) and $^{12}C^+$ (grey scale) signal intensities in *P. dorei* cells incubated for 24 h with no drug (**d**) or ENT-Hi (**e**) in BMM medium. In the individual $^{12}C^+$ images used for merging, the minimum and maximum intensities (grey scale) are: 1–16 counts per pixel (**a**), 2–25 counts per pixel (**b**), 4–80 counts per pixel (**d**) and 0–25 counts per pixel (**e**). In the individual $^{56}Fe^+$ images used for merging, the minimum and maximum intensities (colour scale) are: 3–10 counts per pixel (**a**), 3–35 counts/ per pixel (**b**), 7–17 counts per pixel (**d**) and 4–80 counts per pixel (**e**). Scale bar, 10 μm. **f**, Evaluation of $^{56}Fe^+/^{12}C^+$ intensity ratios in individual *P. dorei* cells. Levels of significance (*P* values) displayed in **c** and **f** were calculated using unpaired two-sided Wilcoxon test comparing the groups 'Ent-Hi' and 'No drug'. Each dot represents a single cell and boxes represent the median, first and third quartiles. Whiskers extend to the highest and lowest values that are within 1.5× the interquartile range. Single-cell measurements and calculations are shown in Supplementary Table 18.

cells to the same level or higher than ENT-Hi alone (Extended Data Fig. 8e,f), thus suggesting that the ENT-Hi:Fe(III) complex behaves similarly to entacapone in terms of its ability to penetrate and accumulate in cells. Microbiota cells and the entacapone accumulator species *P. dorei* display significantly higher levels of iron when exposed to entacapone compared with non-exposed cells, strongly suggesting that ENT-Hi:Fe(III) complexes are stable and reach the cellular space (Fig. 5). Many of these cells may nevertheless starve due to an inability to release iron from the entacapone complex. All together, these results suggest that complexation of the limiting nutrient iron by entacapone is the primary mechanism causing the strong inhibitory effect of entacapone on the microbiome.

**Entacapone promotes growth of iron-scavenging *E. coli***
Next, we interrogated the capability of the few indigenous microbiome members to thrive under the iron-limiting conditions induced

by ENT-Hi. Several commensal and pathogenic Enterobacteriaceae, including *E. coli*, are known to synthesize and release siderophores that bind to Fe(III) with high affinity, enhancing their gut colonization[41]. The most dominant organism in ENT-Hi incubations is an *E. coli* strain classified as *E. coli*_D[42], for which we were able to recover a complete MAG (bin.187, Supplementary Table 8). A search for genes involved in siderophore synthesis within the *E. coli*_D MAG led us to identify an entire gene cluster coding proteins necessary for synthesis, export and import of the siderophore enterobactin (Fig. 6a). Thus, we postulate that the ability of *E. coli* to produce enterobactin enables it to grow under iron-limiting conditions induced by ENT-Hi. Indeed, after isolation of a highly similar *E. coli* strain from glycerol-preserved ENT-Hi incubations (isolate E2, see Methods), we could confirm its ability to grow in minimal medium in the presence of ENT-Hi (Fig. 6b). Similarly, ENT-Hi supplementation did not significantly impact the growth of wild-type *E. coli* K12 strain BW25113, but it significantly reduced growth of BW25113

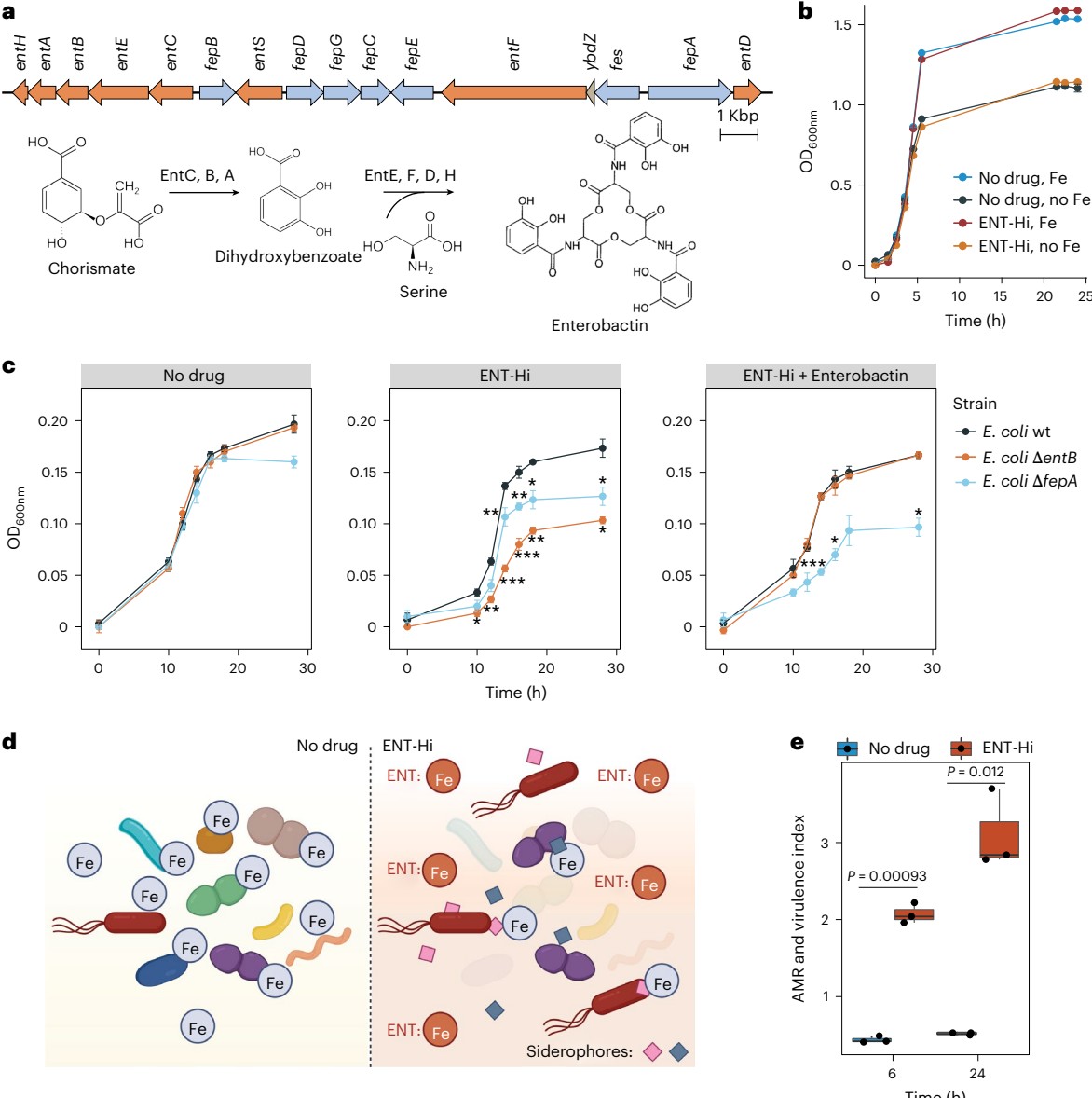

**Fig. 6 | The iron-limiting conditions induced by entacapone select for siderophore producers and drive an increase in AMR and virulence potential in the faecal microbiome. a**, Genetic organization of the enterobactin biosynthesis locus in *E. coli* D bin.187. Genes encoding enzymes involved in enterobactin biosynthesis (*entABCDEFH*) and export (*entS*) are highlighted in orange. The steps of enterobactin synthesis catalysed by the products of these genes are shown at the bottom. Genes involved in enterobactin import and iron release are highlighted in blue. **b**, Growth of the *E. coli* isolate E2, highly identical to *E. coli* D bin.187 determined by optical density measurements of cultures supplemented or not with entacapone (ENT-Hi) and/or FeSO₄ (Fe). Data represent the mean ± s.e.m. of 3 independent growths. **c**, Growth of *E. coli* K12 BW25113 wild-type (wt), *entB* (Δ*entB*) or *fepA* (Δ*fepA*) mutants determined by optical density measurements of cultures supplemented or not with ENT-Hi and/

or enterobactin. Data represent the mean ± s.e.m. of 3 independent growths (Supplementary Table 19). *$P < 0.05$; **$P < 0.01$; ***$P < 0.001$; unpaired two-sided *t*-test with '*E. coli* wt' used as the reference. **d**, Schematic representation of the working hypothesis. In the absence of entacapone (left), enough iron is available to most gut microbiome members. Under ENT-Hi conditions, entacapone complexes available iron (ENT:Fe), and only organisms able to produce and/or import siderophores (represented by diamonds) for iron scavenging are able to grow and thrive (right). Created with Biorender.com. **e**, Increase in the AMR and virulence index (Methods) in no drug controls and entacapone (ENT-Hi)-treated samples. Boxes represent the median, first and third quartiles. Whiskers extend to the highest and lowest values that are within 1.5× the interquartile range. Indicated *P* values were obtained from unpaired two-sided *t*-test comparing the groups 'Ent-Hi' and 'No drug' at each indicated timepoint.

deletion mutants encoding proteins required for enterobactin production (*entB*) or enterobactin uptake (*fepA*) (Fig. 6c and Supplementary Table 19). The *fepA* mutant grows to higher levels than the *entB* mutant in the presence of entacapone, probably due to uptake of enterobactin breakdown products in complex with iron, which may enter the cell via other receptors[43]. Supplementation of the growth medium with enterobactin rescued the entacapone-induced growth defect of *entB*, but not of the *fepA* mutant (Fig. 6c), as expected. These results

strongly support that enterobactin synthesis and/or import is required for optimal growth in the presence of entacapone, and thus, that the presence of entacapone drives iron limitation. Enterobactin chelates ferric iron with much higher affinity than entacapone (pFe(III) = 49)[44], and our results suggest that it enables *E. coli* to scavenge and acquire enough iron to sustain its growth under the iron-limiting conditions induced by entacapone (Fig. 6d). Interestingly, we could not find any genes involved in the production of known siderophores in the MAG

of the *Ruminococcus* strain thriving under ENT-Hi conditions, but we found a gene encoding an orthologue of the membrane protein YfeB required for chelated iron transport in *Yersinia pestis*[45] (Supplementary Table 16). We therefore speculate that *Ruminococcus* expands in the entacapone-supplemented medium by importing iron-loaded siderophores or other iron-chelating molecules, or by producing other high-affinity iron binding proteins[46] (Fig. 6d).

Siderophore production and iron scavenging are commonly observed in pathogenic or pathobiont strains that also tend to encode other virulence traits[41]. A screening of our MAG catalogue for the presence of virulence factors and antimicrobial resistance (AMR) genes[47] identified a diversified panel of AMR and virulence genes to be present in the retrieved gut MAGs (Supplementary Table 17), even though these samples originated from healthy individuals. ENT-Hi drives an increase in the abundance of AMR and virulence within faecal microbiomes (Fig. 6e). This increase is driven in great part by the increase in abundance of *E. coli_D*, whose genome encodes a total of 14 AMR and virulence genes, in addition to at least one siderophore production cluster, thus suggesting a pathogenic potential of this organism. While we cannot rule out the possibility that the increased AMR and virulence index is due to improved genomic characterization of *E. coli* compared with other taxa, these results reveal that ENT-Hi promotes the growth of iron scavenging organisms with an associated pathogenic potential.

## Discussion

Gaining a deeper insight into the interactions between drugs and the gut microbiome is essential for revealing and predicting how the microbiome might influence the availability, efficacy and toxicity of pharmaceuticals. Here we demonstrated that loxapine succinate and entacapone cause major shifts in microbial communities at physiologically relevant concentrations, inhibiting the growth of many taxa while stimulating others. Using single-cell activity measurements, we further revealed fine changes in the activity of specific community members even at low drug concentrations that were not captured by sequencing (Figs. 1e and 2d, and Extended Data Fig. 5). This is a major advantage of our SRS–FISH approach as it captures drug-induced changes in short incubation times, which are ideal for ex vivo systems such as the one used here. Importantly, our results show that loxapine succinate exerts an effect on a broader range of taxa in the context of the community than its effects on microbes grown in pure culture[1]. This highlights the necessity of using complex microbial communities for a better assessment of drug–microbiota interactions. One way by which some taxa may sensitize or protect others to a particular drug is through chemical conversion or accumulation of the drug[9]. However, previous studies have shown that the gut microbiota neither significantly bioaccumulates nor transforms loxapine succinate[9]. Thus, we presume that cross-sensitization to loxapine is probably due to drug-induced changes in microbial metabolites that are involved in interspecies interactions. These differences may, however, not be entirely related to cross-sensitization, but instead due to variations in growth conditions or in the genomic content of strains present in the faecal sample compared with the type strains tested in previous studies[1], which may limit the conclusions from our study.

Entacapone has been shown to be metabolized by gut microbial species[9], mostly by means of nitroreduction. Here we demonstrate that entacapone is also bioaccumulated, which probably results in depletion of entacapone in the surrounding environment over time, explaining the slight alleviation of entacapone's inhibitory effect between 6 and 24 h of incubation (Fig. 1b). In addition, we further show that entacapone bioaccumulation does not necessarily lead to growth inhibition (Fig. 3f,g), similar to what has been described before for other bioaccumulated drugs such as duloxetine[11]. It remains to be determined whether entacapone bioaccumulation can be linked to its biotransformation. Despite its bioaccumulation and extensive conversion[9], we demonstrate that entacapone exerts a strong effect on

the gut community due to its ability to complex iron via the catechol group, which is not affected by nitroreduction.

Iron is an essential enzyme cofactor in most bacteria[36]. Ferric iron complexation by entacapone led to a decrease in biomass accumulation and in the abundance of most of the top abundant taxa detected in our samples, an effect that was reversed by supplementation of iron (Fig. 4c–f). Most iron is expected to enter the large intestine in Fe(III) form, as the increase in pH in the duodenum favours the oxidation of Fe(II) in the presence of oxygen[48]. In fact, ferric iron is an important source of iron for the gut microbiota[49], and we expect Fe(III) chelation by entacapone to impact microbiome homeostasis in the colon. Taxa not found to be significantly impacted by entacapone were presumably able to grow by releasing iron from this complex, or by relying on intracellularly accumulated iron or on high-affinity surface-associated iron transporters that are common in bacteria[46]. Among stimulated taxa, the siderophore-producing *E. coli*_D strain present in our incubations greatly benefitted from entacapone's presence, but only in the context of the community, as entacapone supplementation to an isolate alone did not cause any significant boost in growth (Fig. 6b). Thus, taxa stimulated by entacapone probably acquired iron via the mechanisms mentioned above, or via siderophores, and expanded in the community at the cost of nutrients released by dead cells or supplied via cross-feeding from other species. An increased abundance of Enterobacteriaceae was previously reported for Parkinson's disease (PD) patients taking entacapone[50]. An increase in Enterobacteriaceae may also help to explain diarrhoea experienced by some patients taking entacapone. Our results are consistent with these findings, although they are based on short-term incubations of drugs with microbiomes of healthy individuals, which is a limitation of our study. To better understand the long-term effects of entacapone on the microbiome, it would be important to follow a cohort of PD patients before and during entacapone treatment. Entacapone is always prescribed in combination with levodopa to inhibit its off-site metabolism[24], and such a cohort would also enable investigating the impact of this drug combination on the microbiome.

The expansion of an organism with siderophore synthesis, AMR and virulence potential in the presence of entacapone is concerning. As most successful gut pathogens tend to encode siderophore production systems, it would be important to determine whether entacapone treatment increases the likelihood of intestinal infections, similar to what has been observed in patients taking PPIs[18]. As oral iron supplementation reduces levodopa and entacapone absorption in the small intestine[51,52], we hypothesize that the effect of entacapone on the microbiome could be circumvented in the future by the oral administration of iron between entacapone doses or, alternatively, by delivering iron to the colon of patients taking the drug. If administered in appropriate amounts, this proposed adjuvant therapy could help preserve microbiome homeostasis for patients taking entacapone and/or other catechol-containing drugs that might reach relevant concentrations in the large intestine. Colon-targeted adsorbents, such as DAV132, could also be tested, as these have shown the ability to sequester some drugs[53]. Our results advance our understanding of the impact of antipsychotic and antidyskinetic drug therapies on microbiome homeostasis and their mechanisms of action, and can direct future optimization of such therapies.

## Methods

### Human faecal sample collection

The study protocol was approved by the University of Vienna Ethics Committee (reference no. 00161) and the University of Southampton Ethics and Research Governance Office (reference no. 78743). All meta(data) were 100% anonymized and compliant with the University's regulations. Human faecal samples were collected from 9 healthy adult individuals (3 males and 6 females between 22 and 39 years old) who had not received antibiotics in the previous 3 months and had no

history of digestive disease. All study participants provided written informed consent and self-sampled using an adhesive paper-based faeces catcher (FecesCatcher, Tag Hemi) and a sterile polypropylene tube with the attached sampling spoon (Sarstedt). The participants did not receive any compensation to participate in this study.

## Drug concentrations used for incubations

We used two drug concentrations (low and high) of each drug for our incubations. The low drug concentration (ENT-Low or LOX-Low: 20 µM) was previously used in a screening aimed at determining drug effects on pure culture isolates, while the high concentrations (ENT-Hi: 1,965 µM, LOX-Hi: 100 µM) were based on estimated colon concentrations for each drug and were included to better reflect the exposure of gut microbes to these drugs in the large intestine[1]. The colon concentration estimated for entacapone is 1,965 µM (ref. [1]), based on a typical oral dose of 200 mg, while for loxapine no estimate was available. We predicted that loxapine would reach similar colon concentrations as its chemical and therapeutic analogues amoxapine and clozapine, prescribed at similar doses (10–20 mg daily) and estimated to reach colon concentrations of 138 µM and 153 µM, respectively[1]. Using these values as a reference, we predicted that loxapine succinate should be present in the colon at concentrations of at least 100 µM and chose it as the LOX-Hi concentration. Both drugs are found in their parent form in human plasma samples[54,55]. Some drug metabolites have also been shown to be present, but these are largely products of liver metabolism[56], which is an indication of low levels of chemical transformation of these drugs in the upper gastrointestinal tract before absorption.

## Ex vivo gut microbiome incubations with drugs

For data shown in Figs. 1, 2 and 3, samples from 6 donors (2 males, 4 females, between 25 and 39 years old) were transferred into an anaerobic tent (Coy Laboratory Products) within 30 min after sampling, and all sample manipulation and incubations were performed under anaerobic conditions (5% $H_2$, 10% $CO_2$, 85% $N_2$). Each sample was suspended in M9 mineral medium supplemented with 0.5 mg ml$^{-1}$ D-glucose (Merck), 0.5% (v/v) vitamin solution (DSMZ Medium 461) and trace minerals, herein referred to as sM9. Samples were suspended in sM9 to yield a 0.05 g ml$^{-1}$ faecal slurry. At this point, one aliquot of each sample was collected, pelleted and stored at −80 °C for metagenomic analysis. The homogenate was left to settle for 10 min, and the supernatant (devoid of any large faecal particles) was transferred into a different flask where supernatants from the 6 different donors were combined. This combined sample was further diluted 1:10 in sM9 medium (as described above) or in supplemented brain heart infusion (BHI) medium containing either 0% or 55% $D_2O$ (99.9% atom % (at%) D, Merck) for a final 0% (control) or 50% $D_2O$ in the incubation medium (Fig. 1a). Supplemented BHI medium consisted of 37 g l$^{-1}$ BHI broth (Oxoid), 5 g l$^{-1}$ yeast extract (Oxoid), 1 g l$^{-1}$ L-cysteine (Merck) and 1 g l$^{-1}$ NaHCO$_3$ (Carl Roth). Incubation tubes were supplemented with dimethylsulfoxide (DMSO, Merck), entacapone (Prestwick Chemicals) or loxapine succinate (Prestwick Chemicals) predissolved in DMSO. The final concentration was 2% (w/v) DMSO in all vials (except for the $H_2O$ control, where water was added instead of DMSO). After short incubation times (6 and 24 h) under anaerobic conditions at 37 °C, samples were collected as detailed below and processed to determine: (1) changes in the total microbial loads; (2) microbial community profile dynamics based on 16S rRNA gene amplicon sequencing; (3) reconstructed microbial genomes based on long-read metagenomics and (4) single-cell microbial activity changes via tracing of the incorporation of deuterium from isotopically labelled heavy water ($D_2O$) into single cells of microbiota by chemical imaging based on SRS[32] (Fig. 1a).

At time 0 and after an incubation time of 6 or 24 h at 37 °C under anaerobic conditions, two sample aliquots from each incubation and from controls were collected by centrifugation. One aliquot was washed with 1× PBS and then fixed in 3% paraformaldehyde (PFA) solution for 2 h at 4 °C. Samples were finally washed twice with 1 ml PBS and stored in PBS:ethanol (50% v/v) at −20 °C until further use. The second pelleted aliquot was stored at −20 °C until further processing. A third aliquot was collected into sealed anaerobic vials containing 40% glycerol (Carl Roth) in PBS for a final 50% (v/v) cell suspension in 20% glycerol and stored at −80 °C until further use. In addition, amendment of ENT-Hi to 3 individual faecal samples (2 females and 1 male, average age: 26.3 years) and sample processing were carried out as described above, except that samples were not mixed before incubation (Extended Data Fig. 2).

For iron rescue experiments, fresh faecal samples received from 5 out of the 6 individuals that participated in the initial drug supplementation experiment (except for 1 male that was travelling at the time of the experiment) were collected and processed as described above. Donors were between 25 and 38 years old. To establish appropriate controls for imaging of entacapone bioaccumulation by microbiota cells, an aliquot of the freshly prepared 0.05 g ml$^{-1}$ faecal slurry was immediately fixed with either 3% PFA solution or ethanol (50% v/v) at 4 °C for 2 h. Fixed faecal samples were washed with 1× PBS as described above and incubations with fixed samples and entacapone or entacapone:iron (see below) were conducted in parallel with incubations using live samples. Incubation vials were then supplemented with 2% (v/v) DMSO with or without 1,965 µM entacapone, in the presence or absence of supplemented iron (1 mM FeSO$_4$, Merck; Fig. 5). Additional incubation vials (triplicates) were treated with entacapone pre-complexed with iron: briefly, entacapone and iron (FeSO$_4$ or FeCl$_3$) powder were mixed and resuspended in 2 ml DMSO, yielding a final concentration of 1,965 µM entacapone and 1 mM FeSO$_4$ (or FeCl$_3$), and stored overnight under anaerobic conditions. The next day, 120 µl of monobasic sodium phosphate (0.5 M) was added and the samples were mixed well. After 20 min, samples were centrifuged at 14,000 g for 5 min to remove unbound iron precipitated by the addition of sodium phosphate[57]. The supernatant containing the iron-complexed entacapone was collected into a new Eppendorf tube and supplemented to the faecal incubation vials. Incubations were sampled as described above.

## Cell counts from ex vivo microbiome incubations

Microbial loads in faecal incubation vials were determined using flow cytometry and counting beads as detailed below. Samples preserved in glycerol were diluted 200–800 times in 1× PBS (Supplementary Table 1). To remove any additional debris from the faecal incubations, samples were transferred into a flow cytometry tube by passing the sample through a snap cap containing a 35-µm-pore-size nylon mesh. Next, 500 µl of the microbial cell suspension was stained with the nucleic acid dye SYTO 9 (Thermo Fisher, 0.5 µM in DMSO) for 15 min in the dark. The flow cytometry analysis of the microbial cells present in the suspension was performed using a BD FACSMelody cell sorter (BD Biosciences), equipped with a BD FACSChorus software v.3.0 (BD Biosciences). Briefly, background signals from the instrument and the buffer solution (PBS) were identified using the operational parameters forward scatter (FSC) and side scatter (SSC). Microbial cells were then displayed with the same settings in a scatterplot using the forward scatter (FSC) and side scatter (SSC), and pre-gated on the basis of the presence of SYTO 9 signals (Extended Data Fig. 10). Singlets discrimination was performed. Absolute counting beads (CountBright, Thermo Fisher) added to each sample were used to determine the number of cells per ml of culture by following manufacturer instructions. Fluorescence signals were detected using the blue (488 nm, staining with SYTO 9 and CountBright beads) and yellow-green (561 nm, CountBright beads only) optical lasers. The gated fluorescence signal events were evaluated on the forward–sideways density plot to exclude remaining background events and obtain an accurate microbial cell count. Microsoft Excel v.16.87 was used for data sorting (Supplementary Table 1). Instrument and gating settings were identical for all samples

(fixed staining–gating strategy), and gating strategy is exemplified in Extended Data Fig. 10.

## Nucleic acid isolation and 16S rRNA gene sequencing

Pellets of microbiome incubation samples were resuspended in 600 µl of lysis solution RL (InnuPREP DNA/RNA mini kit, Analytik Jena) and subjected to bead beating for 30 s at 6.5 m s$^{-1}$ in lysis matrix E (MP Biomedicals) tubes. After pelleting cell debris for 10 min at 8,000 g, supernatants were transferred into the InnuPREP DNA/RNA mini kit spin filter tubes (Analytik Jena), and DNA and RNA were extracted according to manufacturer protocol. Amplification of bacterial and archaeal 16S rRNA genes from DNA extracts was performed with a two-step barcoding approach (UDB-H12)[58].

In the first-step PCR, the primers 515F[59] (5′-GTGYCAGCMGCC GCGGTAA-3′) and 806R[60] (5′-GGACTACNVGGGTWTCTAAT-3′), including a 5′-head sequence for 2-step PCR barcoding, were used. PCRs, barcoding, library preparation and Illumina MiSeq sequencing were performed by the Joint Microbiome Facility (Vienna, Austria) under project numbers JMF-2208-05 and JMF-2103-29. First-step PCRs were performed in triplicate (12.5 µl vol per reaction) with the following conditions: 1× DreamTaq buffer (Thermo Fisher), 2 mM MgCl$_2$ (Thermo Fisher), 0.2 mM dNTP mix (Thermo Fisher), 0.2 µM of forward and reverse primer each, 0.08 mg ml$^{-1}$ bovine serum albumin (Thermo Fisher), 0.02 U DreamTaq polymerase (Thermo Fisher) and 0.5 µl of DNA template. Conditions for thermal cycling were: 95 °C for 3 min, followed by 30 cycles of 30 s at 95 °C, 30 s at 52 °C and 50 s at 72 °C, and finally 10 min at 72 °C. Triplicates were combined for barcoding (with 8 PCR cycles). Barcoded samples were purified and normalized over a SequalPrep Normalization Plate kit (Invitrogen) using a Biomek NXP Span-8 pipetting robot (Beckman Coulter), and pooled and concentrated on PCR purification columns (Analytik Jena). Indexed sequencing libraries were prepared with the Illumina TruSeq Nano kit according to manufacturer instructions[61] and sequenced in paired-end mode (2× 300 bp, v3 chemistry) on an Illumina MiSeq system following manufacturer instructions. The workflow systematically included four negative controls (PCR blanks, that is, PCR-grade water as template) for each 90 samples sequenced. The 16S rRNA gene sequences were deposited in the NCBI Sequence Read Archive (SRA) as BioProject Accession PRJNA1033532.

## Analysis of 16S rRNA gene amplicon sequences

Amplicon pools were extracted from the raw sequencing data using the FASTQ workflow in BaseSpace (Illumina) with default parameters[58]. Demultiplexing was performed with the Python package demultiplex v.1.2.1 (Laros JFJ, github.com/jfjlaros/demultiplex) allowing one mismatch for barcodes and two mismatches for linkers and primers. DADA2 (ref. 62) R package v.1.16.0 (https://www.r-project.org/, R 4.0.2) was used for demultiplexing amplicon sequencing variants (ASVs) using a previously described standard protocol[63]. FASTQ reads were trimmed at 150 nt with allowed expected errors of 2. Taxonomy was assigned to 16S rRNA gene sequences on the basis of the SILVA taxonomy[64] (release 138) using the DADA2 classifier.

Samples were analysed using the vegan (v.2.5-.6; https://CRAN.R-project.org/package=vegan) and phyloseq[65] (v.1.30.0) packages in R (https://www.r-project.org/, R 4.0.2). For samples subjected to different drug treatments, sequencing in parallel with two extraction controls (without adding faecal samples) yielded 10 (control 1) and 189 reads (control 2). In control 1, 9 of the 10 reads were assigned to Cyanobacteria or chloroplast and were not detected in the samples. Likewise, in control 2, more than 90% of the reads originated from either Cyanobacteria ASVs or a single Comamonadaceae (Aquabacterium) not detected in any of the samples. These ASVs were removed from analysis. The remaining negative control reads (control 1: 1 read, control 2: 8 reads) were assigned to taxa typically found in the gut that were also detected in the samples and were therefore retained for subsequent

analyses. We assumed these low number of reads to originate from a low level of cross-contamination that can occur when multiple samples are handled in parallel. After quality filtering and removal of contaminant sequences, a total of 1,132 ASVs were retained. The average read number per sample was 11,176 ± 3,087 high-quality sequences and sample coverage was above 98% (Supplementary Table 2). For alpha and beta diversity analysis, sequence libraries were rarefied to 4,681 reads per sample. For samples referring to the entacapone and iron supplementation experiment, sequencing in parallel with two extraction controls yielded 2 (control 1) and 134 reads (control 2). After quality filtering and removal of contaminant sequences (using the rationale described above), a total of 716 ASVs were retained. The average read number per sample was 18,733 ± 4,427 high-quality sequences and the sample coverage was above 99% (Supplementary Table 2). For alpha diversity analysis, sequence libraries were rarefied to 9,729 reads per sample. For quantitative microbiome analyses, relative abundances of each taxon in a sample were calculated after correcting for the different number of copies of the 16S rRNA gene, according to rrnDB (v.5.7). For this correction, we classified ASVs using DADA2 and the RDP[66] taxonomy 18, release 11.5 (https://doi.org/10.5281/zenodo.4310151), by applying default parameters. These corrected relative abundances were then multiplied by the total microbial loads obtained from flow cytometry (Supplementary Table 1), yielding the total abundance of each taxon per sample.

DESeq2 (v.1.26.0)[67] implemented in phyloseq was used to identify significant differences in ASV abundances between drug treatments. Only ASVs that had in total ≥10 reads (relative abundance microbial profile) or 5.0 × 10$^5$ reads (quantitative microbial profile) were considered for comparisons by DESeq2 analyses. All statistical analysis of microbiome data was carried out in R (R 4.0.2). The applied significance tests and obtained P values are referred to in the main text and figure legends.

## Long-read sequencing

DNA for long-read sequencing was isolated using the DNeasy PowerSoil Pro kit (Qiagen), according to manufacturer instructions. A pool of 6 DNA extracts was prepared for sequencing using the ligation sequencing kit (SQK-LSK112, Oxford Nanopore) following manufacturer protocol. The DNA was sequenced on a Promethion P24 sequencing device (Oxford Nanopore) on an R10.4 flow cell (FLO-PRO112, Oxford Nanopore). The DNA sequencing was carried out using Minknow (v.21.10.8, Oxford Nanopore).

## Shotgun metagenomic sequencing

The same 6 samples were individually sequenced in an Illumina Novaseq 6000 platform by the Joint Microbiome Facility (Vienna, Austria) under project number JMF-2110-04. The Illumina reads were trimmed using cutadapt (v.3.1)[68]. Illumina reads were mapped to the assemblies using Minimap2 (v.2.17)[69].

## Metagenomic analysis

The Nanopore reads were assembled using flye[70] (v.2.9-b1768) with '–nano-hq', polished three times with Minimap2 (v.2.17)[69] and Racon (v.1.4.3)[71], followed by two rounds of polishing with Medaka (v.1.4.4, github.com/nanoporetech/medaka). Illumina and Nanopore reads (Supplementary Table 7) were mapped to the assemblies using Minimap2 (v.2.17) and read mappings were converted using SAMtools (v.1.12)[72]. Read coverage and automatic binning was performed using MetaBAT2 (v.2.15)[73]. Contigs labelled as circular by the assembler were extracted as independent bins before the automated binning process. The quality of the recovered MAGs was checked using QUAST (v.5.0.2)[74] and CheckM (v.1.1.1)[75], and genomes were classified using GTDBtk (v.1.5.1)[42]. rRNA genes were detected using Barrnap (v.0.9, https://github.com/tseemann/barrnap) and transfer (t)RNA genes were detected using trnascan (v.2.0.6)[76]. MAGs with completeness >90% but where barrnap did not pick up a 5S rRNA gene were checked

for 5S rRNA genes using INFERNAL (v.1.1.3)[77]. MAGs were searched for iron-related genes and gene neighbourhood protein orthologues using FeGenie (v.1.2)[78].

All MAGs were searched for AMR and virulence genes using AMRFinderPlus (v.3.10.21)[47], which identified genes encoding resistance to, among others, beta-lactams, tetracyclines, macrolides and aminoglycosides, as well as more general antimicrobial resistance genes such as efflux pumps and virulence genes (Supplementary Table 17). The AMR and virulence index (Fig. 6e) was calculated as follows: the total copies of AMR and virulence genes found to be present in each MAG were multiplied by the absolute abundance of the MAG (abundance of the ASV matching the 16S rRNA gene of the MAG) in the sample. The same was repeated for all MAGs for which AMRFinderPlus identified AMR or virulence genes and by summing these, we were able to predict the total number of copies of AMR and virulence genes for each sample, per ml of culture. The resulting values were then normalized to the total biomass per ml of each sample to obtain an AMR and virulence index per sample.

### FISH probe design and optimization
Phylogenetic analysis and FISH probe design were performed using the software ARB (v.7.0)[79]. By analysis of the 16S and 23S rRNA gene, phylogenetic trees were calculated with IQ-TREE (v.1.6.12) using the RAxML GTR algorithm with 1,000 bootstraps in ARB[80]. For abundant groups, 4 FISH probes were designed for this study and 2 additional published probes were used (Supplementary Table 10). The probes were validated in silico with mathFISH to test the in silico hybridization efficiency of target and non-target sequences[81,82]. The number of non-target sequences was assessed using the probe match function in ARB and the mismatch analysis function in mathFISH. All probes were purchased from biomers (Biomers.net) and were double labelled with indocarbocyanine (Cy3) or sulfo-cyanine5 (Cy5) fluorochromes.

Pure cultures of *Escherichia coli* K12, *P. dorei* 175 (DSM 17855) and *B. thetaiotaomicron* VPI-5482 (DSM 2079) were grown in supplemented BHI until the mid-exponential phase and collected by centrifugation. Pure cultures were fixed for 2 h by addition of 3 volumes of 4% (w/v) PFA solution at 4 °C. After washing once with PBS, cells were stored in a 1:1 mixture of PBS and 96% (v/v) ethanol at −20 °C. Where pure cultures were not available, fixed faecal samples with a high relative abundance (as determined by amplicon sequencing) of the specific target taxon were used.

To evaluate probe dissociation profiles, cells obtained from fixed pure cultures or faecal incubation samples (Supplementary Table 10) were spotted onto microscopy slides (Paul Marienfeld). FISH was performed as described before[81], with 3 or 5 h hybridization to obtain fluorescence signals with sufficient intensity. The optimal hybridization formamide concentration was found using formamide dissociation curves, obtained by applying a formamide concentration series in the range of 0–70% in 5% increments[83]. After a stringent washing step and counterstaining using 4′,6-diamidino-2-phenylindole, samples were visualized using a Leica Thunder epifluorescence microscope with an APO ×100/1.40 Leica oil immersion objective and the Leica Application Suite X software (LAS X 5.1.0). Probe EUB338 (ref. 84), which is complementary to a region of the 16S rRNA conserved in many members of the domain Bacteria, was used as a positive control, and a nonsense NON-EUB probe was applied to samples as a negative control for binding. Images for inferring probe dissociation profiles were recorded using the same microscopy settings and exposure times. The probe dissociation profiles were determined on the basis of the mean fluorescence signal intensities of at least 100 probe-labelled cells and evaluated using the ImageJ software (v.1.53t). From the calculated average values, a curve was plotted and the respective value right before a decline on each curve was defined as the optimal formamide concentration.

### FISH in solution
Fixed cells (100 µl) were pelleted at 14,000 g for 10 min, resuspended in 100 µl 96% analytical grade ethanol and incubated for 1 min at room temperature for dehydration. Subsequently, the samples were centrifuged at 14,000 g for 5 min, the ethanol was removed and the cell pellet was air dried. For SRS–FISH analysis, cells were hybridized in solution (100 µl) for 3 h at 46 °C. The hybridization buffer consisted of 900 mM NaCl, 20 mM Tris-HCl, 1 mM EDTA and 0.01% SDS, and contained 100 ng of the respective fluorescently labelled oligonucleotide as well as the required formamide concentration to obtain stringent conditions (Supplementary Table 10). After hybridization, samples were immediately transferred into a centrifuge with a rotor preheated at 46 °C and centrifuged at 14,000 g for 15 min at the maximum allowed temperature (40 °C) to minimize unspecific probe binding. Samples were washed in a buffer of appropriate stringency for 15 min at 48 °C, and cells were centrifuged for 15 min at 14,000 g and finally resuspended in 20 µl of PBS. Cells (5 µl) were spotted on poly-L-lysine-coated cover glasses no. 1.5H (170 µm ± 5 µm thickness, Paul Marienfeld) and allowed to dry overnight at 4 °C under protection from light. Salt precipitates were removed by dipping the coverslips twice in ice-cold Milli-Q water and the coverslips allowed to dry at room temperature under protection from light.

### Picosecond SRS and microscopy
In complex microbial communities, all metabolically active cells will incorporate deuterium (D) from $D_2O$ present in the medium into their biomass during synthesis of new macromolecules[29]. The newly formed carbon-deuterium (C-D) bonds can then be used as a read-out of microbial activity. SRS efficiently excites the Raman active vibrational modes coherently with two synchronized ultrafast lasers and was employed to determine the presence of C-D bonds and thus, microbial activity. In the SRS set up employed, an 80-MHz pulsed laser (InSight DeepSee+, Spectra-Physics) emitting two synchronized femtosecond beams was used (Extended Data Fig. 3a). One beam was tunable in wavelength from 680 nm to 1,300 nm, while the other beam had a fixed wavelength of 1,040 nm. The time delay between single pulses of the two beams is adjustable by a motorized delay line on the 1,040 nm beam. To implement the picosecond (ps) SRS (Extended Data Fig. 3), the fixed beam (termed Stokes beam) was intensity modulated at 2.5 MHz by an acousto-optic modulator (1205c, Isomet Corporation) and co-aligned with the tunable beam (termed pump beam) by a dichroic mirror (DMLP1000, Thorlabs). Both beams were chirped by SF57 rods to 2-ps pulse width and directed towards the lab-built upright microscope frame. Then, a 4-focal system and a flip mirror conjugated a pair of galvo mirrors to the back aperture of a ×60 water objective (UPlanApo ×60W, 1.2 NA, Olympus) or a ×100 1.49 NA oil objective (UAPON 100XOTIRF, Olympus), allowing the collinear pump and Stokes beams to raster-scan the sample via synchronized movement of the galvo mirrors. A 1.4 numerical aperture oil condenser (Aplanat Achromat 1.4, Olympus) collected the output beams, which were then reflected by a flip mirror and filtered by a short-pass filter (DMSP950, Thorlabs). Finally, the filtered-out pump beam was focused onto a silicon photodiode connected to a resonant amplifier effective at a resonant frequency of ~2.5 MHz. The output alternative current signal was further amplified by a lock-in amplifier (UHFLI, Zurich Instrument) at the frequency and in phase (x channel detection) with the modulation. The output direct current signal was recorded for normalization. A data acquisition card (PCIe-6363, National Instruments) collected the output signal for image generation.

To perform widefield fluorescence imaging for FISH visualization of the identical sample areas analysed by SRS and photothermal imaging, two flip mirrors were flipped off (Extended Data Fig. 3a). A halogen lamp (12V100WHAL, Olympus) provided Kohler illumination of the sample from the condenser side. Then, the objective and the tube lens conjugated the sample plane to the camera (CS505CU, Thorlabs).

To enable imaging of various fluorophores, different excitation and emission filter sets were inserted between the lamp and condenser, and in front of the camera. For Cy3 imaging, two 530/10 nm bandpass filters (FBH530-10, Thorlabs) were used as excitation filters and two 570/20 nm bandpass filters (ET570/20x, Chroma) were used as emission filters. For Cy5 imaging, two 640 nm bandpass filters (FBH640-10, Thorlabs) were used as excitation filters and two 670/20 nm bandpass filters (ET670/50 m, Chroma) were used as emission filters.

### Photothermal imaging of entacapone accumulation

By utilizing the multiphoton absorption of entacapone at 10 mM, we detected the photothermal signal originating from optical absorption to generate entacapone distribution maps. A concentrated solution of entacapone was needed for a test and we chose 10 mM to provide a strong signal (Fig. 3a). The experimental setup was identical to picosecond SRS, but with detection of the lock-in signal by the $y$ channel, which exhibits a π/2 phase delay relative to the intensity modulation by the acousto-optic modulator (Supplementary Text and Extended Data Fig. 6b). With this orthogonal phase detection, the interference of the photothermal signal with the signals emerging from cross-phase modulation and SRS was minimized (Extended Data Fig. 6d).

### Image acquisition and processing

Samples were prepared by drying fixed bacterial cells spotted onto poly-L-lysine-coated coverslips (VistaVision cover glasses, No. 1, VWR) in a 4 °C refrigerator and subsequent dipping into water three times to dissolve precipitates from the growth media. Then the bacteria were immersed in 5 µl of water and sandwiched by another coverslip with a 0.11-mm-thick spacer in between. To acquire deuterium incorporation profiles of microbiome members labelled by FISH probes, widefield fluorescence was performed first. For different fluorophores, corresponding excitation and emission filters were applied. The signal and colour gain of the camera were set to 5. Then the exposure time was adjusted to 0.5–5 s depending on the fluorescence signal intensity. The widefield transmission image was acquired by minimizing the condenser aperture and removing the filters.

To acquire the deuterium incorporation profile of the FISH-visualized cells, two flip mirrors were inserted into the beam path to guide the pump and Stokes lasers to the sample. Three SRS images, specific for Raman active vibrational modes of C-D bonds, carbon-hydrogen (C-H) bonds as well as the off-resonance background signal were recorded by tuning the wavelength of the pump beam to 849 nm, 796 nm and 830 nm, respectively. These wavelength values correspond to spectral wavenumbers of 2,163 cm$^{-1}$ (C-D), 2,947 cm$^{-1}$ (C-H) and 2,433 cm$^{-1}$ (silent region). Signal intensities were accumulated over increments of 20 cm$^{-1}$. Images were acquired sequentially within the identical field of view of 32 × 32 µm$^2$ with a raster step size of 106.8 nm. The per-pixel dwell time was set to 10 µs and, depending on the signal intensity level, 1-10 image cycles were recorded to achieve a signal-to-noise ratio (SNR) of >5 for single bacterial cells in the C-H spectral region.

For acquisition of entacapone distribution maps, the pump laser was tuned to 849 nm and the signal detection was switched to the $y$ channel of the lock-in amplifier. All images were recorded utilizing the identical scanning parameters as applied for SRS.

To process the image data sets: first, the illumination patterns were corrected for both widefield images (FISH) and point-scan images (SRS and entacapone distribution). Then, the widefield images and point-scan images were co-localized via a calibrated projective transform matrix. The fluorescence images were utilized to generate a single-cell mask for inference of the single-cell activity and the drug accumulation level. The single-cell activity is expressed as %CD$_{SRS}$ = $(I_{CD}-I_{off}) / (I_{CD} + I_{CH}-2I_{off})$, where the symbols $I_{CD}$, $I_{CH}$ and $I_{off}$ refer to the SRS signal intensities detected at the spectral positions assigned to C-D bonds, C-H bonds and the silent region (off-resonance background).

All intensity values were normalized to the direct current intensity level detected at the photodiode. The relative entacapone accumulation level is expressed as the signal intensity level detected in the $y$ channel of the lock-in amplifier. Intensity outliers (>mean ± 2s.d.) observed in the SRS signals and the photothermal signal of the human gut microbiome samples were rejected from the single-cell masks. This intensity threshold was set after testing and validation with independent samples[32], as food residues can be distinguished on the basis of an irregular shape compared with cells, and by a stronger signal due to other absorption processes that will gradually decrease due to the degradation of the absorption component. All imaging data analysis was performed with CellProfiler v.4.2.6 and Matlab R2023a.

### Bleaching of entacapone photothermal signal

Entacapone has a contribution to the lock-in amplifier $x$ channel measurement that affects accurate %CD$_{SRS}$ measurements. To measure the levels of entacapone signal and activity (%CD$_{SRS}$) in the same cell, we implemented a protocol to bleach the photothermal signal of entacapone that enables subsequent accurate activity measurements (%CD$_{SRS}$). After measuring the entacapone photothermal signal, cells in the same field of view were bleached by continuous laser scanning with the same power used for imaging (power on sample: 30 mW pump and 120 mW Stokes; dwell time: 10 µs; 800 frames scanned in 720 s). This resulted in a sharp drop of signal in the $y$ channel, which plateaued at 500 s, indicating a drop in entacapone signal to levels below the limit of detection (Extended Data Fig. 7a). This bleaching resulted in a smaller but detectable drop in intensity in the $x$ channel that reached a plateau at around the same time as the $y$ channel (500 s; Extended Data Fig. 7b). SRS measured in the $x$ channel signal in cells that were not exposed to entacapone remained constant throughout the entire bleaching process, indicating that the drop in the $x$ channel in ENT-Hi cells is indeed driven by the entacapone bleaching (Extended Data Fig. 7c). After bleaching, the SRS intensity became independent of entacapone levels and could then be used to determine the levels of activity per cell (%CD$_{SRS}$). Correlation between PT at %CD$_{SRS}$ was done in a mix of faecal samples (from 2 donors, 1 male and 1 female, age range 22–39 years) incubated with Ent-Hi and 50% D$_2$O in sM9 medium.

### Nanoscale secondary ion mass spectrometry (NanoSIMS) sample preparation and analysis

After incubation, cell fractions of faecal samples and *P. dorei* were washed once in PBS, fixed in a 3% PFA solution at 4 °C for 2 h and then stored in a suspension of PBS and ethanol (1:1) at 4 °C. For sample preparation, each suspension was diluted in 150 µl ultrapure water and homogenized in a sonication bath (Sonorex-Super_RK-31, Bandelin) at 35 kHz twice for 30 s to separate single cells from cellular aggregates. Cells were then collected and concentrated by filtration on gold coated (film thickness 150 nm, obtained by sputter deposition) polycarbonate filters (GTTP type, 0.2 µm pore size, Millipore) to a density of 200–300 cells per 65 × 65 µm$^2$. To remove PBS precipitates and residual ethanol, filters were washed with ultrapure water and air dried after spotting of the cell suspensions. The density and distribution of single cells was inspected by light microscopy utilizing air objectives. Appropriate regions for NanoSIMS analysis were marked by a laser microdissection microscope (Leica LMD 7000). Selected filter regions were attached to antimony-doped silicon wafer platelets (7.1 × 7.1 mm, Active Business) serving as sample carriers.

NanoSIMS measurements were conducted on a NanoSIMS 50L ion microprobe (Cameca) equipped with a Hyperion RF-Plasma O$^-$ ion source (Oregon Physics) at the Large-Instrument Facility for Environmental and Isotope Mass Spectrometry at the University of Vienna. Before data acquisition, analysis areas were presputtered by rastering of a high-intensity (500 pA beam current), defocused O$^-$ ion beam to an O$^-$ ion fluence of 2.6 × 10$^{16}$ ions cm$^{-2}$. Data were acquired as multilayer image stacks by repeated scanning of a finely focused O$^-$ primary ion

beam (~100 nm probe size at 12 pA beam current) over areas between $50 \times 50\ \mu m^2$ and $65 \times 65\ \mu m^2$ at $512 \times 512$ pixel image resolution and a primary ion beam dwell time of 5 ms pixel$^{-1}$. The detectors of the multicollection assembly were positioned for simultaneous detection of $^{12}C^+$, $^{23}Na^+$, $^{31}P^+$, $^{40}Ca^+$ and $^{56}Fe^+$ secondary ions. The mass spectrometer was tuned to achieve a mass resolving power of >8,000 (according to Cameca's definition) for detection of $^{56}Fe^+$ secondary ions.

NanoSIMS images were generated and analysed with the Open-MIMS[85] plugin v.3.0.5 in the image processing package Fiji (1.54g)[86]. All images were autotracked for compensation of primary ion beam and/or sample stage drift, and secondary ion signal intensities were corrected for detector dead time. For visualization of the iron distribution pattern between cells from individual samples, overlay images of the secondary ion signal intensities of $^{12}C^+$ (shown on a grey scale) and $^{56}Fe^+$ (shown on a rainbow colour scale) were generated (Fig. 5). Region of interest (ROI) specific numerical data evaluation was applied to display the relative iron content of single cells via normalization of the $^{56}Fe^+$ signal intensity to the respective $^{12}C^+$ signal intensity (Supplementary Table 18). ROIs were defined utilizing carbon as a reference element indicating biomass, visualized by the $^{12}C^+$ secondary ion signal intensity distribution images. The presence of single cells was confirmed via their morphological appearance in the $^{23}Na^+$ and $^{40}Ca^+$ secondary ion maps.

### Siderophore assay

The iron binding capacity of entacapone was tested using the SideroTec Assay (Accuplex Diagnostics), a colorimetric test for the detection of siderophores, following manufacturer instructions. Wells were read photographically and on a microplate reader at 630 nm wavelength (Multiskan GO microplate spectrophotometer, Thermo Fisher).

### Measurement of Fe(II) using ferrozine

The ferrozine chromogenic method, in which ferrozine reacts with free divalent Fe to form a stable magenta complex species with a maximum absorbance at 562 nm (ref. [87]), was used to determine the ability of entacapone to complex Fe(II) and to follow the presence of Fe(II) in incubations with entacapone. All solutions were prepared in degassed solvents or buffers to prevent spontaneous iron oxidation. To determine the ability of entacapone to complex Fe(II), FeSO$_4$·7H$_2$O (Carl Roth) was dissolved in distilled water to produce a 183.3 µM Fe(II) working solution. Ferrozine (BLD Pharmatech) was dissolved in distilled water to produce a 246.3 µM ferrozine working solution. Na$_2$H$_2$EDTA (Carl Roth) was dissolved in Tris-HCl buffer pH 7.5 to produce a 50 mM stock solution, and entacapone was dissolved in DMSO/water (1:1) to produce a 2 mM solution. Both Na$_2$H$_2$EDTA and entacapone were further diluted to the final concentrations shown in Extended Data Fig. 9a. A volume of 50 µl of Fe(II) working solution was then added to a 96-well plate, followed by addition of 50 µl of a chelator solution (diluted in water as needed). Ferrozine working solution (100 µl) was then added and the absorption was measured at 526 nm with a Tecan plate reader after 10 min of incubation. To determine the ability of pure DMSO to oxidize iron, FeSO$_4$·7H$_2$O (Carl Roth) was dissolved in pure DMSO (Merck) or water to produce 50 µM Fe(II) working solutions. Each solution was aliquoted into a 96-well plate and an equal volume of a 0.2 M ferrozine solution (and respective dilutions) was added. Absorption was measured at 526 nm with a Tecan plate reader after 10 min of incubation. Finally, to determine the oxidation of Fe(II) in sM9 media, a 1 mM FeSO$_4$ stock solution was prepared by dissolving FeSO$_4$·7H$_2$O in sM9 media or water. After 1, 5, 10, 15 and 30 min of incubation at room temperature, 50 µl was added to a 96-well plate and diluted with 50 µl water. Ferrozine solution (100 µl, 2 mM) was added and the absorption was measured at 526 nm with a microplate reader (Tecan). For the immediate timepoint, 48.75 µl of sM9 media was added to a 96-well plate. Subsequently, 1.25 µl of a 40 mM FeSO$_4$ solution, 50 µl degassed water and 100 µl of a 2 mM ferrozine solution were added and the absorption measured at 526 nm. A calibration curve was established using FeSO$_4$ stock solutions (1, 0.5, 0.25, 0.125, 0.0625 and 0.03125 mM). For this, 50 µl of each FeSO$_4$ stock solution was added to a 96-well plate and diluted with 50 µl water. Ferrozine solution (100 µl, 2 mM) was added to each well and absorption was measured at 526 nm with a Tecan plate reader after 10 min of incubation at room temperature.

### Isolation and sequencing of an *E. coli* isolate

*Escherichia coli* isolate E2 was isolated from glycerol-preserved faecal samples (mix of samples from 6 donors, 4 females and 2 males aged 25–39 years; ethics reference no. 00161) after incubation in sM9 medium supplemented with 1,965 µM entacapone for 48 h under anaerobic conditions. An aliquot of the glycerol stock was serially diluted in PBS, and 10$^{-4}$ and 10$^{-5}$ dilutions were plated in BHI agar. Isolated colonies were restreaked 3 times to purity and submitted to colony PCR using the 16S rRNA gene primers 616V (5′-AGA GTT TGA TYM TGG CTC AG-3′) and 1492R (5′-GGT TAC CTT GTT ACG ACT T-3′). Single colonies were picked with an inoculation loop, resuspended in 50 µl of nuclease-free water and boiled at 95 °C for 10 min to lyse the cells and release cell contents. After a short spin to pellet cell debris, 2 µl of the supernatant was added to a PCR reaction mix (final concentrations; Green 1× Dream*Taq* buffer, dNTPs 0.2 mM, BSA 0.2 mg ml$^{-1}$, *Taq* polymerase 0.05 U µl$^{-1}$, primers 1 µM) prepared in a final volume of 50 µl per reaction. The amplification cycles were as follows: initial denaturation at 95 °C for 3 min, followed by 30 cycles at 95 °C for 30 s, 52 °C for 30 s, 72 °C for 1.5 min and a final elongation at 72 °C for 10 min. PCR products were visualized on 1.5% agarose gel electrophoresis, and subsequently cleaned and concentrated on columns (innuPREP PCRpure kit, Analytik Jena) according to manufacturer instructions. Concentrations were measured using Nanodrop and samples were sent for Sanger sequencing to Microsynth (Vienna, Austria). Sequencing results were analysed using BLASTn[88] against the 16S rRNA gene sequences retrieved by metagenomics. The near full-length 16S rRNA gene sequences of isolate E2 obtained were 99% identical to the 16S rRNA copies of *E. coli* D bin.187 and 100% identical to ASV_9ec_9n4.

### Growth of microbial pure cultures

*Bacteroides thetaiotaomicron* (DSM 2079) cells were grown anaerobically (85% N$_2$, 10% CO$_2$, 5% H$_2$) at 37 °C in *Bacteroides* minimal medium (BMM) containing 27.5 µM of iron (FeSO$_4$, Merck)[89]. *Bacteroides* and *Phocaeicola* (formerly *Bacteroides*) species do not grow well in sM9 medium when in isolation. Therefore, we used BMM, a defined medium tailored for *Bacteroides* spp.[89] for which the concentration of iron is known and can be controlled. To test the effect of iron on rescuing the inhibitory effect of entacapone on *B. thetaiotaomicron* growth, entacapone-treated (1,965 µM) cultures were supplemented with 100 µM, 500 µM or 1 mM of iron. Growth rescue was observed when cultures were supplemented with either 500 µM FeSO$_4$ (Fe(II)) or FeCl$_3$ (Fe(III)), and for all subsequent experiments, only FeSO$_4$ was used. Cu(II)SO$_4$ (500 µM or 1 mM, Merck) or entacapone preloaded with iron (1,965 µM, prepared as described above) were also supplemented to *B. thetaiotaomicron* cultures. Fe-loaded entacapone was obtained by addition of 1 mM of Fe(II) (FeSO$_4$) or Fe(III) (FeCl$_3$) to ENT-Hi, followed by removal of any excess iron cations via phosphate precipitation[57]. Samples for total cell counts were taken at 0, 24 and 48 h of growth under anaerobic conditions. *Phocaeicola dorei* (DSM 17855) was grown in BMM in the presence or absence of 1,965 µM entacapone. After 24 and 48 h of growth, an aliquot was collected and fixed with PFA as described above, and stored at −20 °C in PBS:ethanol 50% (v/v) until further analyses. To test whether the growth of the *E. coli* faecal isolate E2 was dependent on the presence of iron, the isolate was grown in MOPS medium with or without 10 µM FeSO$_4$ in the presence or absence of entacapone for 24 h at 37 °C. MOPS medium was chosen as it enables the reduction of free-iron concentration to growth-limiting levels

compared with media containing M9 salts[90] (such as sM9). Wild-type *E. coli* K12 strain BW25113, *entB* mutant and *fepA* mutants (Keio collection strains OEC5042, OEC4987-213605695 and OEC4987-200829240, respectively) were purchased from Horizon discovery (https://horizondiscovery.com). Isolated colonies of BW25113 wild-type and mutant strains for *entB* (encoding the enterobactin synthase component B) and *fepA* (encoding a ferric enterobactin Ton-B-dependent outer membrane transporter) were grown anaerobically in MOPS medium with no additional iron. To avoid the transfer of any iron from the pre-inoculum, pre-inoculum cells were washed thoroughly in MOPS medium without iron before inoculation. Enterobactin (Merck) was dissolved in DMSO (2 mM) and supplemented to the medium to a final concentration of 2 μM, as needed.

### Determination of total cell loads in pure cultures

To determine microbial cell loads in pure culture, samples were collected at 0, 24 or 48 h of growth and stained (either undiluted or after a dilution of 10 times in 1× PBS) with the QUANTOM Total Cell Staining kit (Logos Biosystems). Total cell counts were determined using a QUANTOM Tx Microbial Cell Counter (Logos Biosystems) following manufacturer instructions.

### Study design and statistical analyses

No statistical methods were used to predetermine sample sizes, but our sample sizes are similar to those reported in previous publications[7,32]. Data distribution was assumed to be normal but this was not formally tested. For microcosm experiments, faecal samples were thoroughly mixed before allocation to microcosms, and allocations to microcosms were performed randomly. For flow cytometry, cell counts, SRS and photothermal measurements and analyses, both sample preparation and sample measurements were randomized. Preparation of samples for sequencing was randomized. For Nano-SIMS, we first analysed the entacapone-treated samples followed by the untreated controls to confirm that the observed Fe distribution patterns were not affected by measurement artefacts. Blinding during the microcosm and pure-culture experiments was not possible, as different samples had to receive a different treatment/drug. Sample preparation for sequencing was carried out by an independent investigator in a blinded manner. Initial steps of sequencing data processing were performed blindly. Bioinformatic analyses of sequencing data were performed using automated software but were not blinded to identify samples for direct comparisons. For all other experiments or assays, the investigators were not formally blinded because of the high demand for certain samples and timepoints, which required appropriate sample selection.

### Reporting summary

Further information on research design is available in the Nature Portfolio Reporting Summary linked to this article.

## Data availability

The 16S rRNA gene sequencing data, metagenomics data and retrieved MAGs have been deposited in the National Center for Biotechnology Information (NCBI) Sequence Read Archive under BioProject number PRJNA1033532. Databases used were: the Genome Taxonomy Database (GTDB) v.0.1.3 (https://gtdb.ecogenomic.org/); RDP taxonomy 18, release 11.5 (https://doi.org/10.5281/zenodo.4310151); Short Read Archive (https://www.ncbi.nlm.nih.gov/sra); SILVA taxonomy (release 138) using the DADA2 classifier (https://zenodo.org/records/4587955); and the SILVA 119 SSU NR99 database (https://www.arb-silva.de/download/arb-files/).

## Code availability

CellProfiler pipelines and Matlab codes have been deposited in GitHub[91].

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

## Acknowledgements

The research reported in this manuscript was funded by NIH Awards R35GM136223 (to J.-X.C.), R01EB032391 and R01AI141439 (to J.-X.C.) and supported by the Boston University Micro and Nano Imaging Facility and the Office of the Director, NIH Award S10OD024993. The content is solely the responsibility of the authors and does not necessarily represent the official views of the NIH. Funding for the presented research was also provided via the Young Independent Research Group Grant ZK-57 (to F.C.P.) of the Austrian Science Fund (FWF) and the FWF-Wittgensteinaward Z383-B (to M.W.), as well as the Austrian Science Fund (FWF)-funded Cluster of Excellence 'Microbiomes drive Planetary Health' (10.55776; COE 7; D.B., T.B., M.W.). The project was also supported by pilot funding from the Institute for Life Sciences, University of Southampton. We thank J. Schwarz, G. Kohl, P. Pjevac and J. S. Silva from the Joint Microbiome Facility of the Medical University of Vienna and the University of Vienna for assisting with amplicon and metagenomic sequencing, as well as repositing of sequencing data. We thank L. Guantai for assistance in setting up drug incubations.

## Author contributions

F.C.P., X.G., J.-X.C. and M.W. designed the study. F.C.P, X.G., J.M.K., K.M., J.B.S., D. Seki and M.D. performed the experiments with input from T.B., Y.Z., D.B. and A.S. F.C.P, J.M.K., R.H.K, B.H. and K.W. analysed sequencing data and performed bioinformatic analysis. A.S. and S.I. conducted NanoSIMS analysis and D. Szamosvari investigated iron binding and redox states. F.C.P. and X.G. wrote the manuscript with input from all co-authors, and all authors read and approved the final manuscript.

## Competing interests

The authors declare no competing interests.

## Additional information

**Extended data** is available for this paper at https://doi.org/10.1038/s41564-024-01853-0.

**Correspondence and requests for materials** should be addressed to Fátima C. Pereira, Ji-Xin Cheng or Michael Wagner.

[1]Centre for Microbiology and Environmental Systems Science, Department of Microbiology and Ecosystem Science, University of Vienna, Vienna, Austria. [2]School of Biological Sciences, University of Southampton, Southampton, UK. [3]Department of Electrical and Computer Engineering, Boston University, Boston, MA, USA. [4]Joint Microbiome Facility of the Medical University of Vienna and the University of Vienna, Vienna, Austria. [5]Department of Biological Chemistry, Faculty of Chemistry, University of Vienna, Vienna, Austria. [6]Department of Chemistry, Boston University, Boston, MA, USA. [7]Department of Laboratory Medicine, Medical University of Vienna, Vienna, Austria. [8]Department of Biomedical Engineering, Photonics Center, Boston University, Boston, MA, USA. [9]Department of Chemistry and Bioscience, Aalborg University, Aalborg, Denmark. [10]Present address: School of Biological Sciences, University of Portsmouth, Portsmouth, UK. [11]These authors contributed equally: Fátima C. Pereira, Xiaowei Ge. ✉e-mail: f.c.pereira@soton.ac.uk; jxcheng@bu.edu; michael.wagner@univie.ac.at

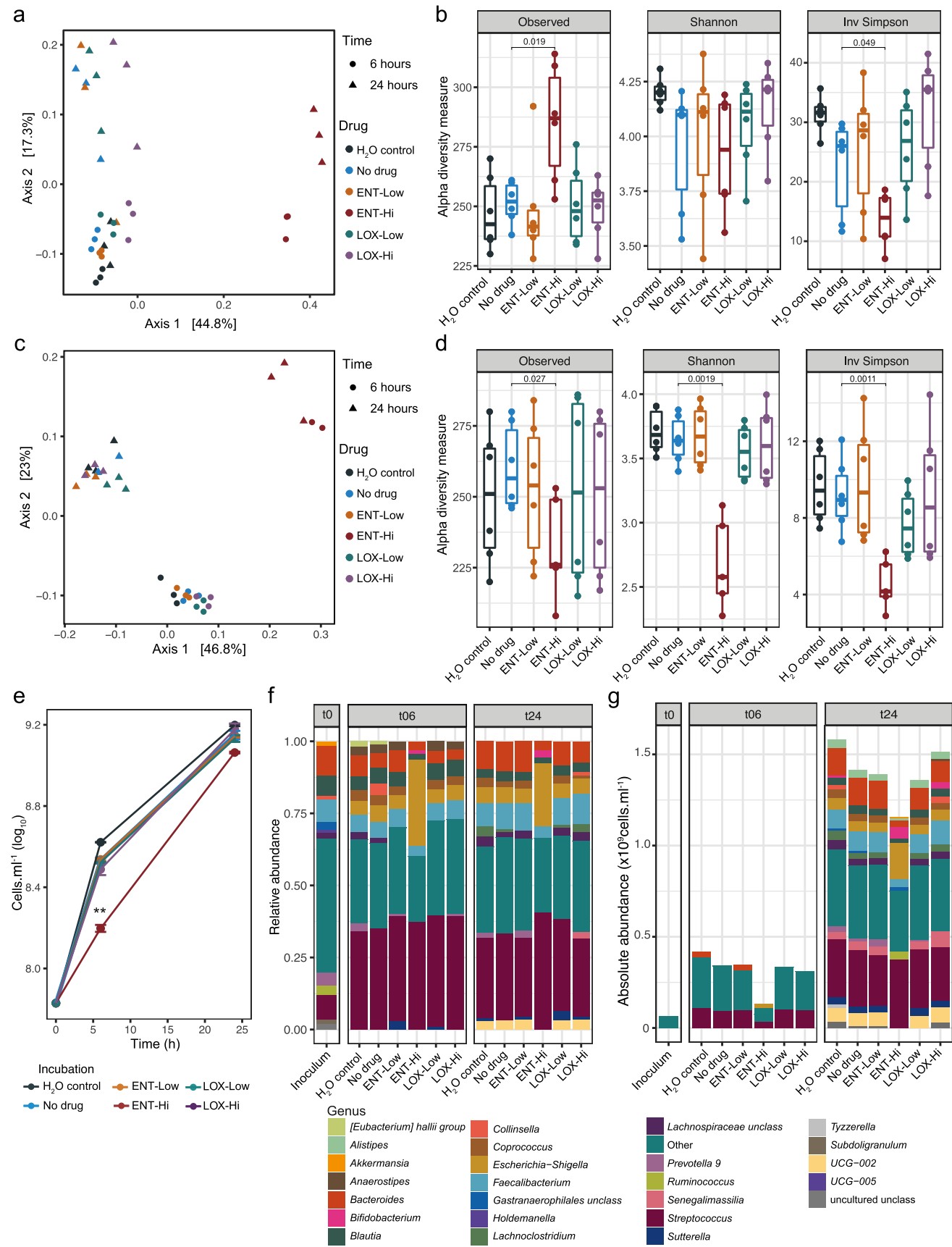

**Extended Data Fig. 1 | See next page for caption.**

**Extended Data Fig. 1 | Relative and absolute abundance profiles of faecal samples resuspended in sM9 or BHI and incubated with drugs. a**. Ordination plot of Bray–Curtis distances of microbial communities (ASV-level composition) in faecal samples resuspended in sM9 medium (**a**) or BHI medium (**c**) and incubated in the presence of drugs ($n = 3$ incubation vials per condition). Samples are coloured according to the drug amended and shaped according to the time of incubation. Please note that in (**a**) at T6 ENT-Hi condition two of the data points show overlapping positions in the ordination plot. sM9 medium: PERMANOVA = 0.001, $R^2$ = 0.64. BHI medium: PERMANOVA = 0.001, $R^2$ = 0.45. **b**. Alpha diversity metrics (number of Observed ASVs, Shannon Index, and Inverse Simpson Index) of faecal samples resuspended in sM9 medium (**b**) or BHI medium (**d**) and incubated in the presence of drugs. Each data point represents a replicate and samples from both 6 and 24 h of incubation are shown. Significant differences (p-values) determined by two-sided Wilcoxon testing are indicated,

with "No drug" used as a reference group. Boxes represent the median, first and third quartile. Whiskers extend to the highest and lowest values that are within one and a half times the interquartile range. **e**. Total cell loads in faecal samples resuspended in BHI medium and incubated with drugs, as assessed by flow cytometry (Supplementary Table 1). **p = 0.002; unpaired two-sided t-test with "No drug" used as a reference group. Data points represent the mean value +/- SEM from three incubation vials per condition. **f, g**. Relative (**f**) and absolute (**g**) genus abundance profiles of microbial communities originating from faecal samples resuspended in BHI medium and incubated in the presence of drugs at T0 and after 6 or 24 h of incubation. Each bar represents the mean from triplicate incubations. All genera present at relative abundances below 0.025 or absolute abundances below $2.5 \times 10^7$ cells.ml$^{-1}$ were assigned into the category "Other" ("unclass": unclassified). Complete relative and absolute abundance profiles can be found in Supplementary Tables 3 and 4.

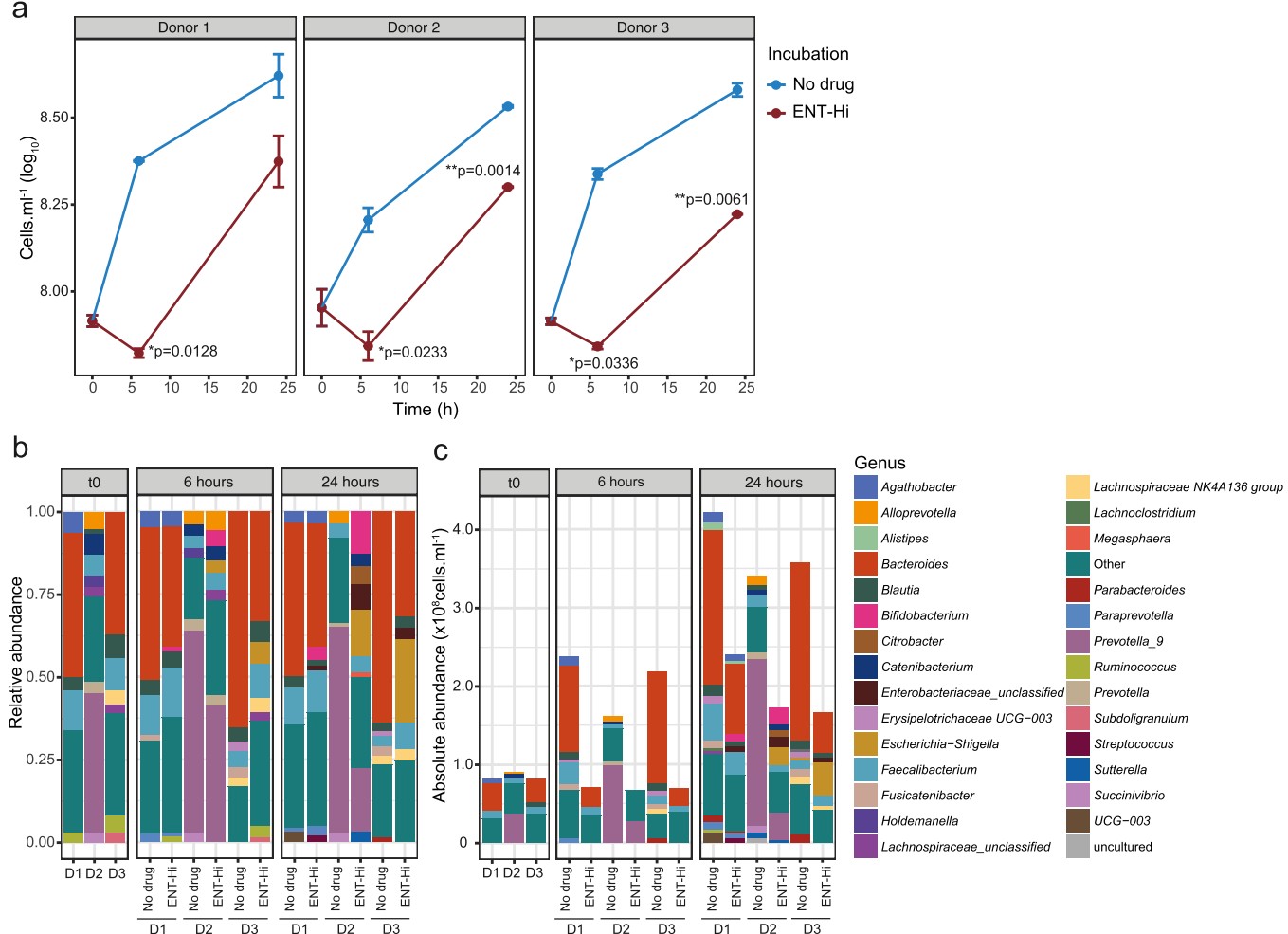

**Extended Data Fig. 2 | Relative and absolute abundance profiles of individual faecal samples incubated with entacapone. a**. Total microbial cell loads in incubations of samples from three different donors (compared to samples used in main text, for example, Fig. 1) with entacapone (ENT-Hi) and controls (No drug - DMSO only), at the start of the incubation (time=0 h, t0) and after 6 and 24 h in sM9 medium. Total cell loads were assessed by flow cytometry. Indicated P values are from an unpaired two-sided t-test with "No drug" as a reference group. Data points represent the mean value +/- SEM from three incubation vials per condition. **b, c**. Relative (**b**) and absolute (**c**) genus abundance profiles of faecal microbial communities (Donor 1: D1, Donor 2: D2 and Donor 3: D3) incubated with ENT-Hi and assessed by 16S rRNA gene amplicon sequencing. The community composition at 0 h (inoculum) is also shown. All genera present at a relative abundance below 0.025 or absolute abundance below $5 \times 10^6$ cells. ml-1 were assigned to the category "Other". Each bar represents the mean from triplicate incubations.

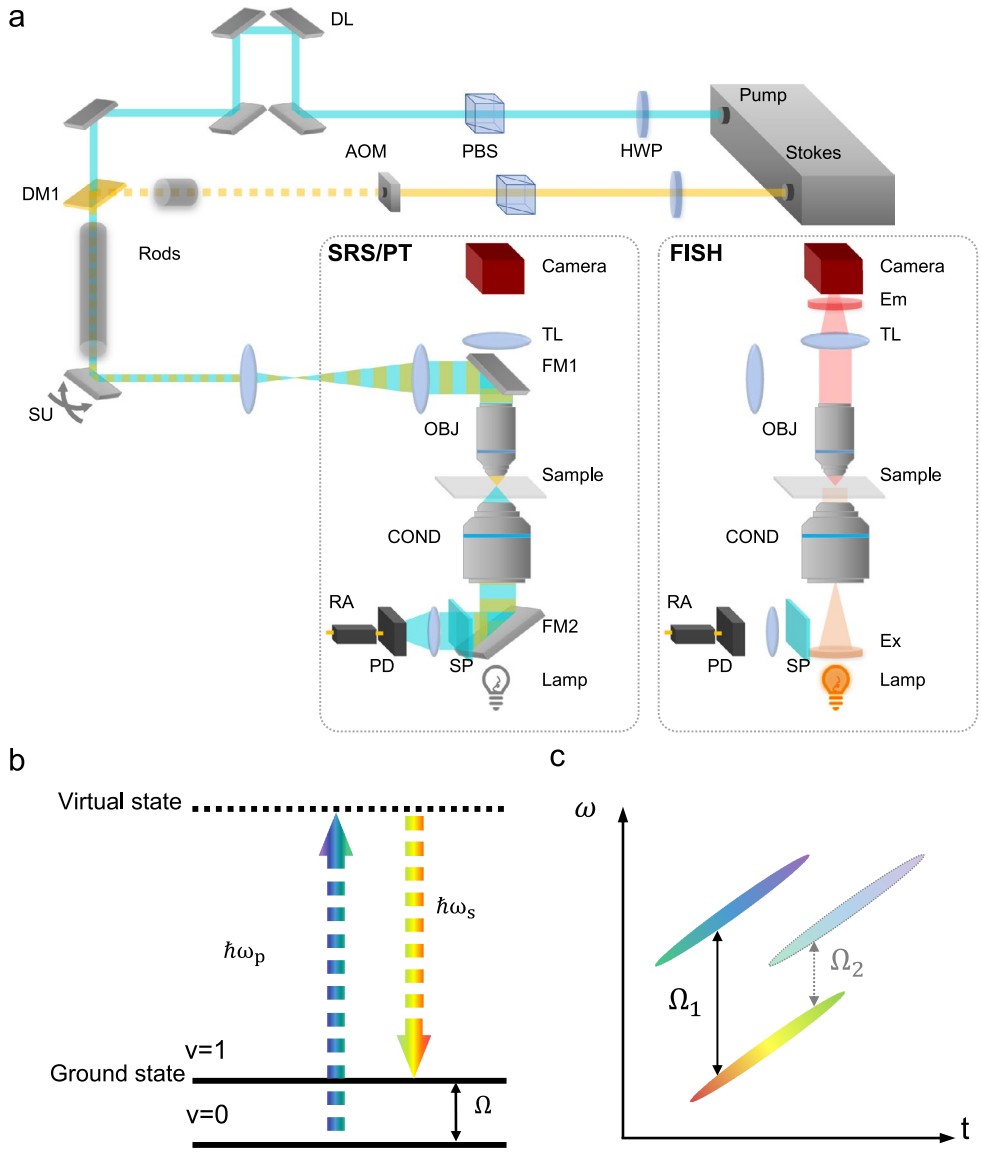

**Extended Data Fig. 3 | Pump probe microscopy combined with widefield fluorescence imaging for correlative chemical and fluorescence *in situ* hybridization imaging. a**. Experimental setup. Left dashed box: laser point scanning mode for SRS/PT. DL: delay line; HWP: half waveplate; PBS: polarized beam splitter; AOM: acousto-optic modulator; DM: dichroic mirror; SM: scanning unit; FM: flip mirror; OBJ: objective; COND: condenser; SP: short pass filter;

PD: photodiode; RA: resonant amplifier; Ex: excitation filter; Em: emission filter; TL: tube lens. **b**. Stimulated Raman scattering energy diagram. $\omega_p$: pump beam frequency; $\omega_S$: Stokes beam frequency; $\hbar$: Planck's constant; $\Omega$: energy of the vibrational mode; v: vibrational level. **c**. Picosecond stimulated Raman scattering by spectral focusing.

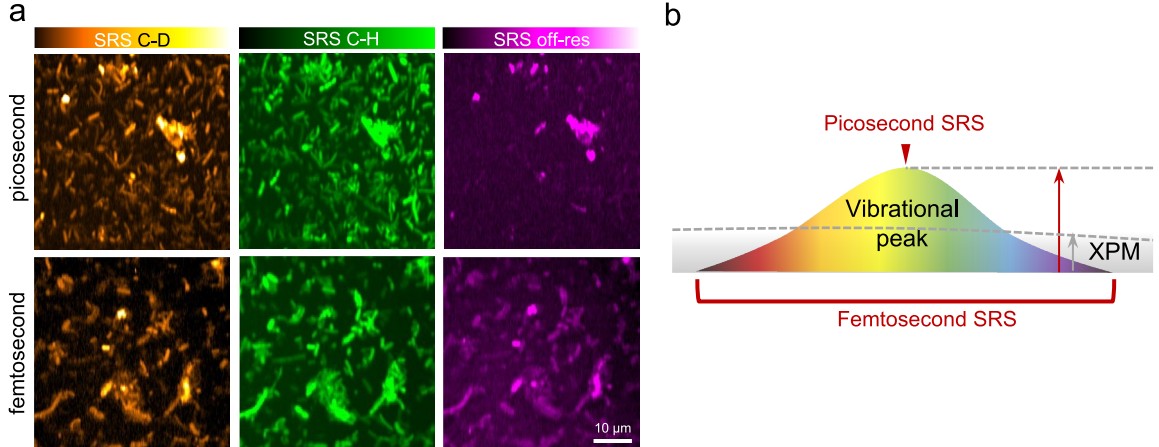

**Extended Data Fig. 4 | Sensitive mapping of microbial activity by picosecond SRS. a**. Picosecond SRS alleviates cross-phase modulation (XPM) and provides higher sensitivity for measuring the activity of microbial cells via deuterium incorporation with a laser power below the cell-damaging threshold. Picosecond SRS pump power: 30 mW, Stokes power: 120 mW. Femtosecond SRS: pump power: 15 mW, Stokes power: 70 mW. We reproducibly detected analogous differences when measuring at least one additional batch of samples. **b**. Picosecond and femtosecond SRS probing difference in the spectral domain. For picosecond SRS, the targeting bandwidth is 20 cm$^{-1}$, which is much narrower than the femtosecond SRS which covers more than 200 cm$^{-1}$. Thus, the ratio between the signal (vibrational feature) and the background (cross phase modulation, XPM) is improved in picosecond SRS.

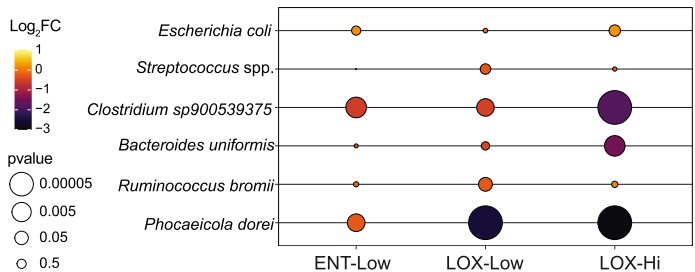

**Extended Data Fig. 5 | Change in abundance of key taxa at 24 h of incubation.** Bubble plot denoting the fold change in absolute abundances for the taxa targeted by FISH and incubated with drugs relative to "No drug" incubations, as determined by DeSeq2, at 24 h of incubation in sM9. Original data is from triplicates per condition. P values were attained using the Wald test and corrected for multiple testing using the Benjamini and Hochberg method.

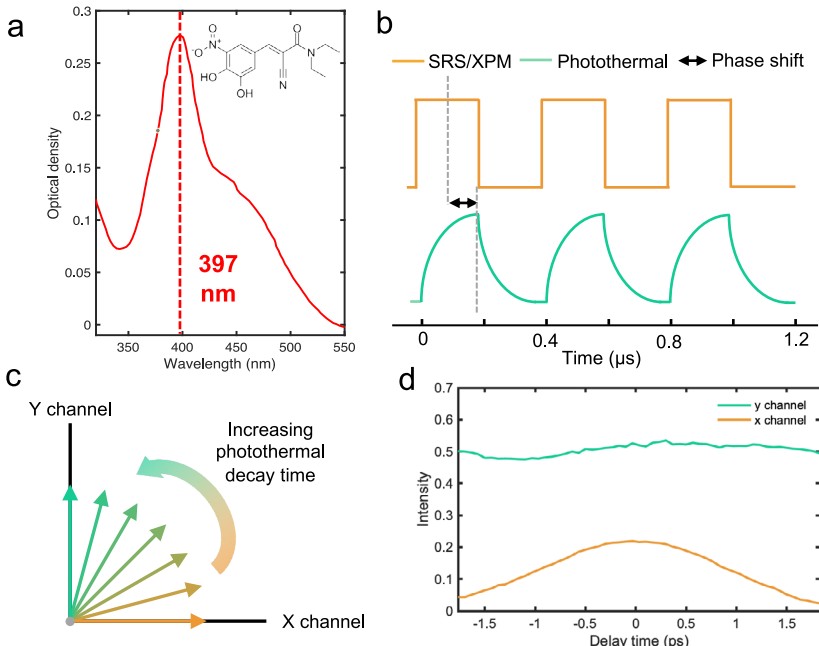

**Extended Data Fig. 6 | Entacapone photothermal signal retrieved from modulation transfer detection. a**. 10 mM Entacapone DMSO solution (under acidic pH conditions) measured by UV-VIS spectroscopy. **b**. Photothermal signal time trace exhibits a π/2 phase delay relative to the intensity modulation by the AOM and instantaneous signals, such as SRS and XPM. **c**. Pump probe signal phase upon increasing photothermal decay time. **d**. Entacapone accumulation in microbiota cells via pump probe detection with delay in time and x/y channel lock-in amplifier signal detection. As photothermal is not sensitive to picosecond level delay, the y channel showed a constant strong photothermal signal from the drug, while in the x-channel, the instantaneous cross phase modulation background showed up as a cross-correlation profile of two the pump and probe laser beams.

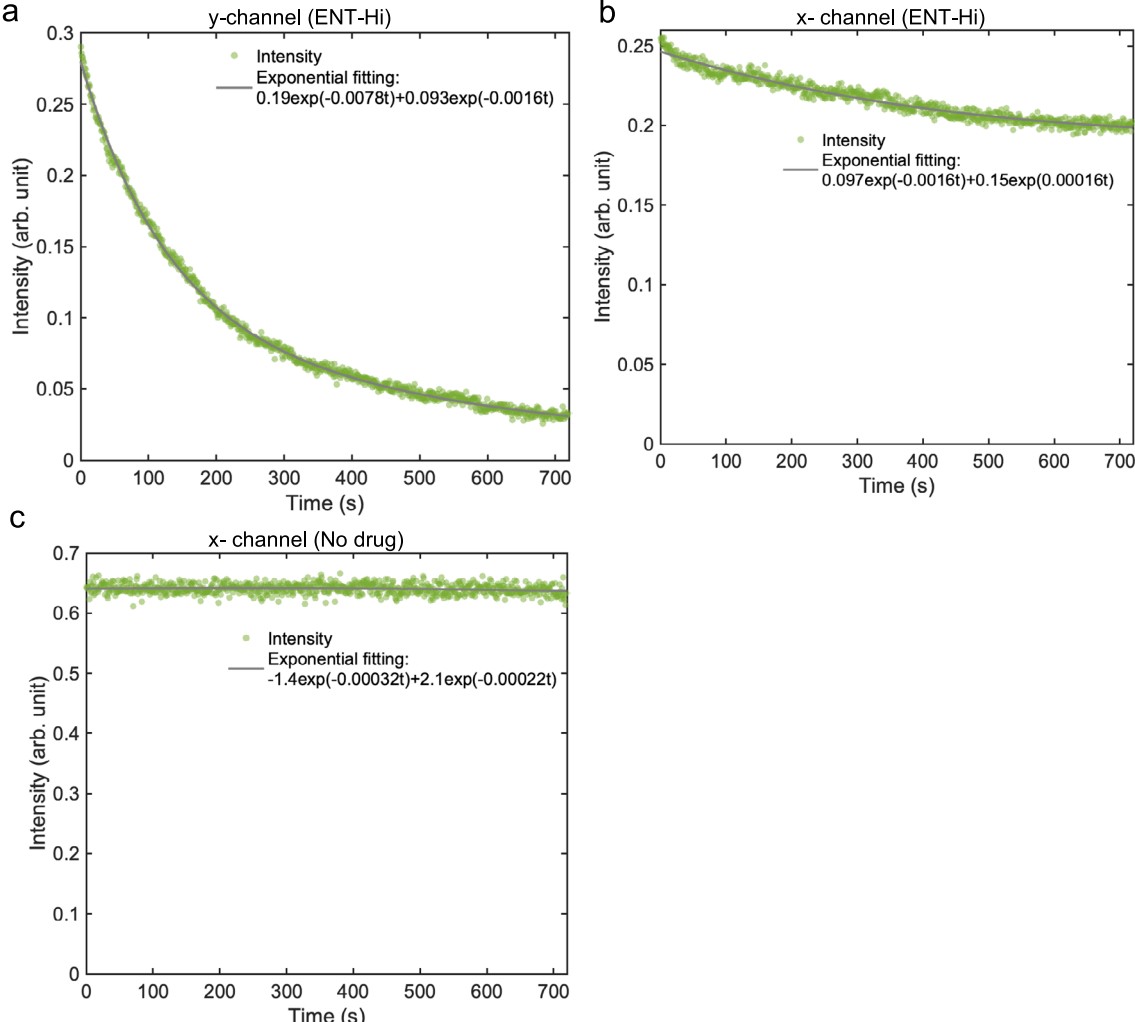

**Extended Data Fig. 7 | Recorded signal intensity of the lock-in amplifier x and y channels over time in faecal samples subjected to continuous bleaching of the entacapone photothermal signal. a**. Signal intensity of the lock-in amplifier y channel (corresponding to photothermal signal) in samples incubated with ENT-HI and $D_2O$. **b**. Signal intensity of the x channel (corresponding to SRS signal and its background) in samples incubated with ENT-HI and $D_2O$. **c**. Signal intensity of the SRS x channel (corresponding to SRS signal and its background) in samples incubated with no drug, but with $D_2O$. Each dot represents the average signal intensity of 20 microbiota cells.

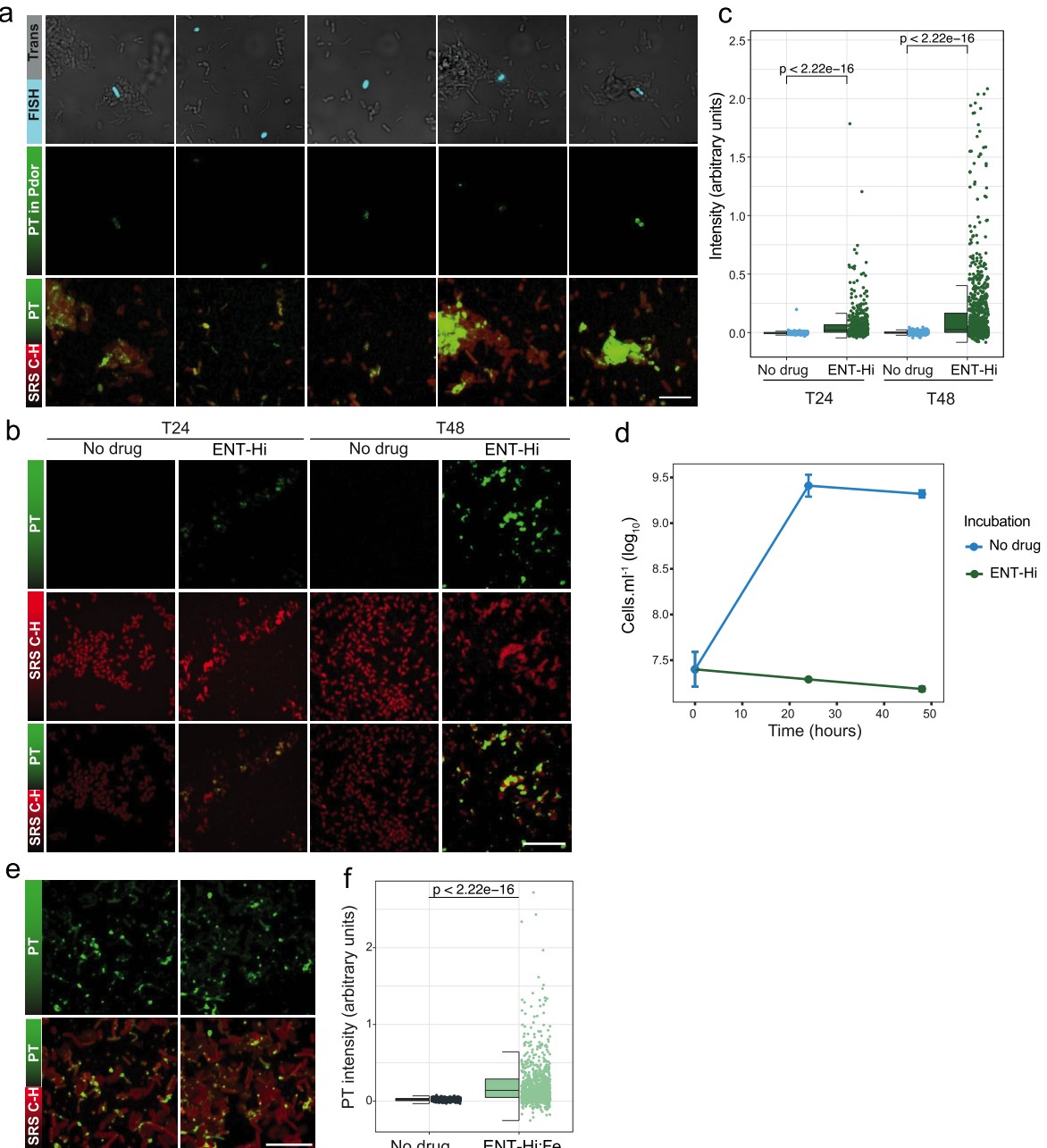

**Extended Data Fig. 8 | Photothermal imaging of entacapone accumulation by *Phocaeicola dorei* and faecal samples. a**. All columns show representative images of faecal samples incubated with ENT-Hi for 6 h followed by hybridization with a fluorescently-labelled oligonucleotide probe targeting *Phocaeicola dorei* (Pdor) (Supplementary Table 10). Top panel: fluorescence signals from hybridized cells (FISH). Middle panel: Photothermal signal intensity from entacapone (PT) in Pdor cells only (PT signals of other cells were removed to enhance clarity). Bottom panel: PT signals overlayed with the SRS C-H signals. PT channel contrast: min 0 max 1.8. C-H channel is represented on a log scale. We reproducibly detected analogous differences when measuring at least one additional batch of samples. **b**. Representative images of a pure culture of *Phocaeicola dorei* incubated anaerobically with entacapone for 24 or 48 h. Photothermal signal from entacapone (PT) in *P. dorei* cells is shown in the top panel, SRS C-H signal in the middle panel, and an overlay of both signals is shown in the bottom panel. **c**. Single-cell specific photothermal signal intensity detected in samples shown in *b*. p-values were determined by the unpaired two-sided t-test comparing the groups "Ent-Hi:Fe" and "No drug". **d**. Growth (assessed by microbial cell counts) of *P. dorei* in the presence or absence of entacapone. Data points represent the mean abundance values +/- SEM of three independent growths per condition. **e**. Accumulation of entacapone pre-complexed with Fe(III) by faecal samples anaerobically incubated for 6 h with ENT-Hi pre-complexed with $FeSO_4$ (ENT-Hi:Fe). Photothermal signal from entacapone (PT) is shown on top and the overlay with the SRS C-H biomass signal is shown in the bottom panel. **f**. Single-cell specific photothermal signal intensity in samples shown in *e*. p-value was determined with the unpaired two-sided Wilcoxon test comparing the groups "Ent-Hi:Fe" and "No drug". In *c* and *f*, boxes represent median, first, and third quartile. Whiskers extend to the highest and lowest values that are within one and a half times the interquartile range.

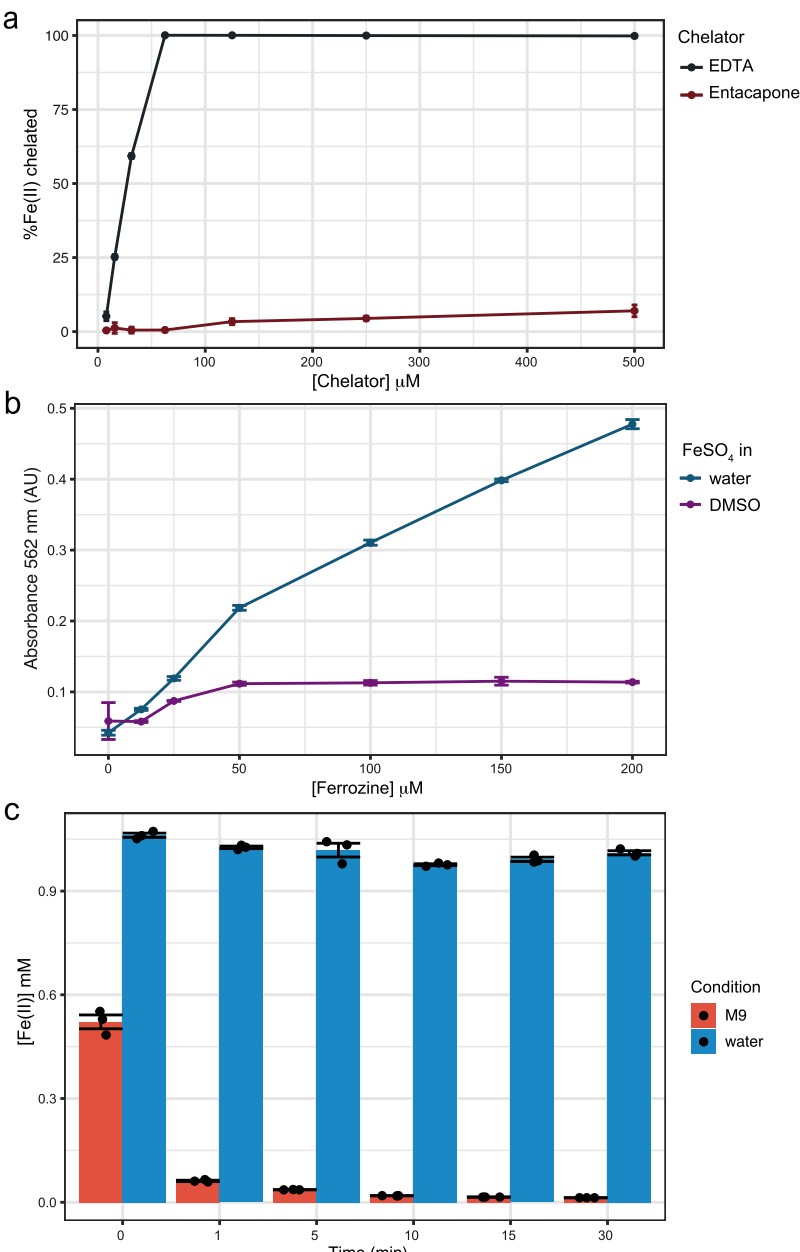

**Extended Data Fig. 9 | Determination of Fe(II) concentrations by spectrophotometric measurements of the Fe(II)–ferrozine complex.** The ferrozine reagent reacts with divalent Fe to form a stable magenta complex species with a maximum absorbance at 562 nm. **a.** Percentage of total Fe(II) in anaerobic solutions of FeSO$_4$ containing increasing concentrations of Na$_2$H$_2$EDTA (a known Fe(II) chelator) or entacapone, as determined using the ferrozine method. **b.** Absorbance at 562 nm, indicating the presence of ferrozine-Fe(II) complexes in solutions of FeSO$_4$ in degassed DMSO (100%) or degassed double distilled water, following reaction with increasing concentrations of ferrozine. **c.** Fe(II) concentrations in water or sM9 solutions of FeSO$_4$ (1 mM FeSO$_4$). FeSO$_4$ was dissolved in degassed water or sM9 and incubated for 0, 1, 5, 10, 15 and 30 min, as indicated, before the ferrozine reagent was added. Concentrations of Fe(II) were determined using a calibration curve obtained by measuring absorbance at 562 nm in solutions of known Fe(II) concentration (see Methods) supplemented with 2 mM ferrozine. In $a,b,c$ error bars denote standard deviation from triplicates per condition. Data are presented as mean values from three independent experiments +/- SD.

a

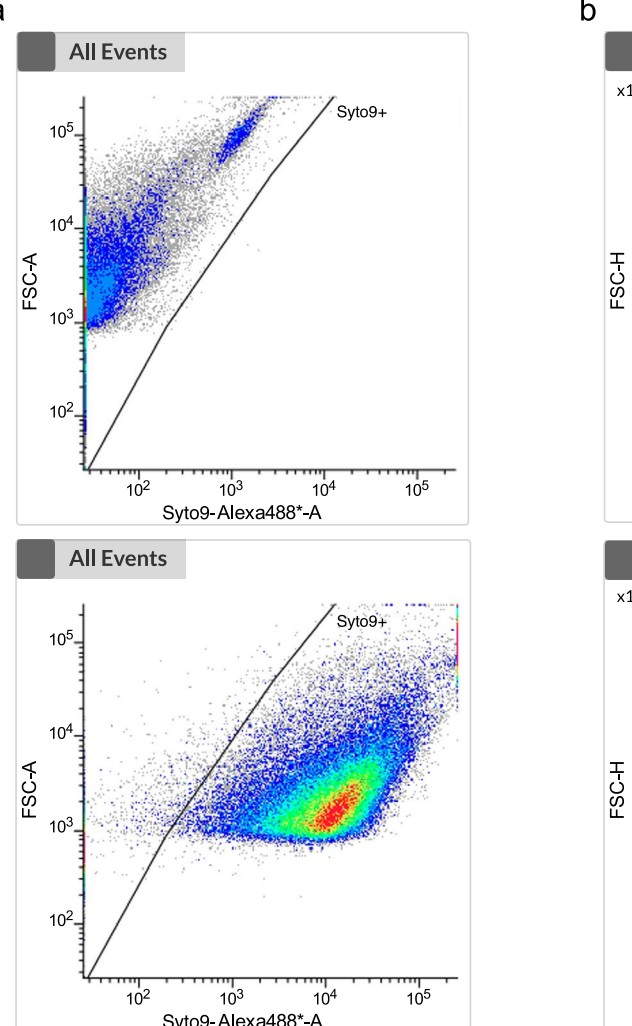

b

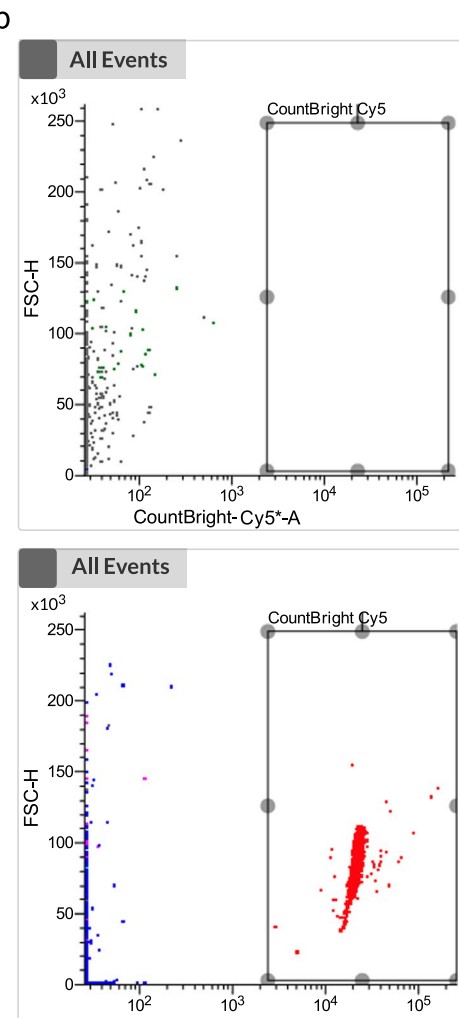

**Extended Data Fig. 10 | Flow cytometry gating strategy to determine total cell counts.** A fixed staining–gating strategy was used. **a**. Gating was performed on the Syto9-Alexa488-A channel to discriminate between debris or background and microbial events stained with SYTO™9. Gating was established by comparing colour density acquisition plots of the sample unstained (top panel) with plots from the same sample stained with SYTO™9 (bottom panel). Each panel reflects over 100,000 registered events. **b**. An additional gating was performed on the CountBright-Cy5-A channel to count in parallel the number of CountBright™ beads (added to the sample). Gating was established by comparing acquisition plots of the sample without the addition of beads (top panel) with plots from the same sample with CountBright™ beads added

(bottom panel). Events pseudocolored red represent CountBright™ bead events that fall within the FSC-H/CountBright-Cy5-A gate. The number of cell events were defined as events registered in the FSC-A/Syto9-Alexa488-A gating area, excluding number of bead events observed in the FSC-H/CountBright-Cy5-A gate (as CountBright™ beads also display Alexa488-A signal). Concentration of cells in a sample was calculated as: number of cell events divided by the number of bead events, and then multiplied by the known concentration of beads in the sample, according to the assigned bead count of the lot. At least 2,000 bead events were recorded per sample, for accurate counting. The acquisition rate did not exceed 2,000 events per second.

Prof. Michael Wagner
Prof. Ji-Xin Cheng

# Reporting Summary

## Statistics

For all statistical analyses, confirm that the following items are present in the figure legend, table legend, main text, or Methods section.

| n/a | Confirmed | |
|---|---|---|
| ☐ | ☒ | The exact sample size ($n$) for each experimental group/condition, given as a discrete number and unit of measurement |
| ☐ | ☒ | A statement on whether measurements were taken from distinct samples or whether the same sample was measured repeatedly |
| ☐ | ☒ | The statistical test(s) used AND whether they are one- or two-sided *Only common tests should be described solely by name; describe more complex techniques in the Methods section.* |
| ☐ | ☒ | A description of all covariates tested |
| ☐ | ☒ | A description of any assumptions or corrections, such as tests of normality and adjustment for multiple comparisons |
| ☐ | ☒ | A full description of the statistical parameters including central tendency (e.g. means) or other basic estimates (e.g. regression coefficient) AND variation (e.g. standard deviation) or associated estimates of uncertainty (e.g. confidence intervals) |
| ☐ | ☒ | For null hypothesis testing, the test statistic (e.g. $F$, $t$, $r$) with confidence intervals, effect sizes, degrees of freedom and $P$ value noted *Give P values as exact values whenever suitable.* |
| ☒ | ☐ | For Bayesian analysis, information on the choice of priors and Markov chain Monte Carlo settings |
| ☒ | ☐ | For hierarchical and complex designs, identification of the appropriate level for tests and full reporting of outcomes |
| ☐ | ☒ | Estimates of effect sizes (e.g. Cohen's $d$, Pearson's $r$), indicating how they were calculated |

*Our web collection on statistics for biologists contains articles on many of the points above.*

## Software and code

Policy information about availability of computer code

| Data collection | BD FACSChorus™ v3.0; Minknow v. 21.10.8, Oxford Nanopore Technologies; Leica Application Suite X (LAS X 5.1.0). |
|---|---|
| Data analysis | ARB v. 7.0; R v4.0.2; R packages: DADA2 v1.16.0, vegan v2.5-.6, phyloseq v1.30.0, DESeq2 v1.26.0; Demultiplex FASTA/FASTQ v1.2.1; cutadapt v. 3.1; Minimap2 v. 2.17; flye v. 2.9-b1768; Racon v. 1.4.3; Medaka v. 1.4.4; SAMtools v. 1.12; MetaBAT2 v. 2.15; QUAST v. 5.0.2; CheckM v. 1.1.1; GTDBtk v. 1.5.1; Barrnap v. 0.9; trnascan v. 2.0.6; INFERNAL v. 1.1.3; AMRFinderPlus v.3.10.21; IQTREE v 1.6.12; rrnDB (version 5.7); ImageJ software v 1.53t; FeGenie (v.1.2); Cell Profiler v4.2.6; Matlab R2023a, Fiji 1.54g with OpenMIMS plug in v3.0.5. In addition, CellProfiler pipelines and Matlab codes have been deposited in GitHub (https://github.com/buchenglab/srs-fish-drugs) |

For manuscripts utilizing custom algorithms or software that are central to the research but not yet described in published literature, software must be made available to editors and reviewers. We strongly encourage code deposition in a community repository (e.g. GitHub). See the Nature Portfolio guidelines for submitting code & software for further information.

## Data

Policy information about availability of data

All manuscripts must include a data availability statement. This statement should provide the following information, where applicable:

- Accession codes, unique identifiers, or web links for publicly available datasets
- A description of any restrictions on data availability
- For clinical datasets or third party data, please ensure that the statement adheres to our policy

16S rRNA gene sequencing data, metagenomics data and retrieved MAGs have been deposited in the National Center for Biotechnology Information (NCBI) Sequence Read Archive under BioProject number PRJNA1033532.

Databases used were: Genome Taxonomy Database (GTDB) v.0.1.3 (https://gtdb.ecogenomic.org/); RDP taxonomy 18, release 11.5 (https://doi.org/10.5281/zenodo.4310151); Short Read Archive (https://www.ncbi.nlm.nih.gov/sra); SILVA taxonomy (release 138) using the DADA2 classifier (https://zenodo.org/records/4587955); SILVA 119 SSU NR99 database (https://www.arb-silva.de/download/arb-files/).

# Research involving human participants, their data, or biological material

Policy information about studies with human participants or human data. See also policy information about sex, gender (identity/presentation), and sexual orientation and race, ethnicity and racism.

| | |
|---|---|
| Reporting on sex and gender | This study included participants of both sexes: 3 males and 6 females. In most of the analyses performed, samples from both sexes were combined (i.e. pooled), and therefore we could not perform analyses with sex as a factor. In experiments where samples were analysed separately (long-read sequencing of faecal samples), there was insufficient statistical power to perform analyses with sex as a factor. |
| Reporting on race, ethnicity, or other socially relevant groupings | N.a. |
| Population characteristics | Study participants (between 22 and 39 years old, average age 32.4 years old) had not received antibiotics in the prior 3 months and had no history of digestive disease. |
| Recruitment | Study participants were researchers of the University of Vienna and University of Southampton. Participants were recruited via email. Participants worked in the same building and were chosen based on their availability to provide a fresh faecal sample. All study participants provided written informed consent. Participants did not receive any compensation from participating in this study. Participants were all young adults, which can be considered a sampling bias. |
| Ethics oversight | University of Vienna Ethics Committee (reference #00161). University of Southampton Ethics and Research Governance Office (reference #78743). |

Note that full information on the approval of the study protocol must also be provided in the manuscript.

# Field-specific reporting

Please select the one below that is the best fit for your research. If you are not sure, read the appropriate sections before making your selection.

☒ Life sciences  ☐ Behavioural & social sciences  ☐ Ecological, evolutionary & environmental sciences

For a reference copy of the document with all sections, see nature.com/documents/nr-reporting-summary-flat.pdf

# Life sciences study design

All studies must disclose on these points even when the disclosure is negative.

| | |
|---|---|
| Sample size | For microcosms experiments, faecal samples form 6 individuals were pooled and used per incubation, and each condition was run in triplicates. All triplicates were sequenced and analysed. Our rationale for mixing samples from 6 healthy donors was to assess the effect of drugs on a microbial community with greater microbial diversity than that of a single individual. For feasibility reasons (availability of freshly collected samples on the same day), this number was restricted to six donors. No statistical methods were used to pre-determine sample sizes, but our sample sizes are similar to those reported in previous publications. For SRS analyses, the number of cells analyzed was chosen based on feasibility. |
| Data exclusions | No other data was excluded from analyses, except for:<br>-amplicon sequencing variants (ASVs, from 16S rRNA gene amplicon sequencing) assigned to Cyanobacteria or chloroplast were removed based on pre-established criteria: these ASVs were detected in negative controls but were not present in any of the faecal samples, and were therefore removed from analyses;<br>-stimulated Raman scattering (SRS) analyses: during image acquisition and processing, outlier cells displaying abnormal intensity (>mean±2 standard deviations) were rejected from the single cell masks. This intensity threshold was set after test and validation with independent samples, as food residues can be distinguished based on an irregular shape compared to cells, and by a stronger signal due to other absorption processes, that will gradually decrease due to the degradation of the absorption component. |

| Replication | Microcosm experiments with drugs were replicated (two times in total) and the results were successfully reproduced. Entacapone and iron rescue experiments using pure culture experiments were successfully reproduced one time. All other experiments were replicated at least one time. NanoSIMS analysis were only performed once based on feasibility, and was deemed successful because the iron levels in entacapone-exposed cells were very highly significant compared to control cells. For SRS measurements, we reproducibly detected analogous differences between treatments when sample analyses were repeated. |
| --- | --- |
| Randomization | For microcosms experiments, faecal samples were thoroughly mixed prior to allocation to microcosms, and allocations to microcosms were performed randomly. For flow cytometry, cell counts and SRS/photothermal analyses, both sample preparation and sample measurements were randomized. Preparation of samples for sequencing was randomized. For NanoSIMS, we first analysed the entacapone treated samples followed by the untreated controls to confirm that the observed Fe distribution patterns were not affected by measurement artifacts. |
| Blinding | Blinding during the microscosm and pure culture experiments was not possible, as different samples had to receive a different treatment/drug. Sample preparation for sequencing was carried out by an independent investigator in a blinded manner. Initial steps of sequencing data processing were performed blindly. Bioinformatic analyses of sequencing data was performed with automated software, but were not performed blindly, as we needed to know which samples to compare. For all other experiments or assays, the investigators were not formally blinded because of the high demand for certain sample/group samples at certain time points, which required appropriate selection of some of the samples. |

# Reporting for specific materials, systems and methods

We require information from authors about some types of materials, experimental systems and methods used in many studies. Here, indicate whether each material, system or method listed is relevant to your study. If you are not sure if a list item applies to your research, read the appropriate section before selecting a response.

## Materials & experimental systems

| n/a | Involved in the study |
| --- | --- |
| ☒ | ☐ Antibodies |
| ☒ | ☐ Eukaryotic cell lines |
| ☒ | ☐ Palaeontology and archaeology |
| ☒ | ☐ Animals and other organisms |
| ☒ | ☐ Clinical data |
| ☒ | ☐ Dual use research of concern |
| ☒ | ☐ Plants |

## Methods

| n/a | Involved in the study |
| --- | --- |
| ☒ | ☐ ChIP-seq |
| ☐ | ☒ Flow cytometry |
| ☒ | ☐ MRI-based neuroimaging |

## Plants

| Seed stocks | N.a. |
| --- | --- |
| Novel plant genotypes | N.a. |
| Authentication | N.a. |

## Flow Cytometry

### Plots

Confirm that:

☒ The axis labels state the marker and fluorochrome used (e.g. CD4-FITC).

☒ The axis scales are clearly visible. Include numbers along axes only for bottom left plot of group (a 'group' is an analysis of identical markers).

☒ All plots are contour plots with outliers or pseudocolor plots.

☒ A numerical value for number of cells or percentage (with statistics) is provided.

### Methodology

| Sample preparation | Faecal samples from six donors (two males, four females) were transferred into an anaerobic tent (Coy Laboratory Products, USA) within 30 min after sampling, and all sample manipulation and incubations were performed under anaerobic conditions (5% $H_2$, 10% $CO_2$, 85% $N_2$). Each sample was suspended in M9 mineral medium supplemented with 0.5 mg.mL-1 D-glucose |
| --- | --- |

(Merck), 0.5% v/v of vitamin solution (DSMZ Medium 461) and trace minerals, herein referred to as sM9. Samples were suspended in sM9 to yield a 0.05 g.mL-1 faecal slurry. At this point one aliquot of each sample was collected, pelleted, and stored at -80°C for metagenomic analysis. The homogenate was left to settle for 10 minutes, and the supernatant (devoid of any large faecal particles) was transferred into a new flask, where supernatants from the six different donors were combined. This combined sample was further diluted 1:10 in sM9 medium (as described above) or in supplemented Brain Heart Infusion (BHI) medium containing either 0% or 55% D2O (99.9% atom % (at%) D; Merck) for final 0% (control) or 50% D2O in incubation medium. Supplemented BHI medium consisted of 37 g.L-1 of brain heart infusion broth (Oxoid), 5 g.L-1 yeast extract (Oxoid), 1 g.L-1 L-cysteine (Merck) and 1 g.L-1 NaHCO3 (Carl Roth GmbH, Germany). Incubation tubes were supplemented with dimethylsulfoxide (DMSO, from Merck), entacapone (Prestwick Chemicals) or loxapine succinate (Prestwick Chemicals) pre-dissolved in DMSO. The final concentration was 2% w/v of DMSO in all vials (except for the H2O control, where water was added instead of DMSO). A subset of vials was supplemented with 20 μM or 1965 μM entacapone (ENT-Low and ENT-Hi, respectively) and another subset was supplemented with 20 μM or 100 μM loxapine succinate (LOX-Low and LOX-Hi, respectively).  At time 0, and after an incubation time of 6 or 24 hours at 37°C under anaerobic conditions, two sample aliquots from each incubation and controls were collected by centrifugation. One aliquot was washed with 1× PBS and then fixed in 3% paraformaldehyde solution for 2 h at 4°C. Samples were finally washed two times with 1 ml of PBS and stored in PBS:Ethanol (50% v/v) at -20°C until further use. The second pelleted aliquot was stored at -20°C until further processing. A third aliquot was collected into sealed anaerobic vials containing 40% glycerol (Carl Roth GmbH, Germany) in PBS for a final cell 50% v/v cell suspension in 20% glycerol and stored at -80°C until further use. In addition, amendment of ENT-Hi to three individual faecal samples (two females and one male) and sample processing were carried as described above, except that samples were not mixed prior to incubation.

Samples preserved in glycerol were diluted 200 to 800 times in 1x PBS. To remove any additional debris from the faecal incubations, samples were transferred into a flow cytometry tube by passing the sample through a snap cap containing a 35 μm pore size nylon mesh. Next, 500 μL of the microbial cell suspension was stained with the nucleic acid dye SYTO™ 9 (Thermo Fisher Scientific, 0.5 μM in DMSO) for 15 min in the dark. The flow cytometry analysis of the microbial cells present in the suspension was performed using a BD FACSMelody™ (BD Biosciences), equipped with a BD FACSChorus™ software v 3.0 (BD Biosciences).

| | |
|---|---|
| Instrument | BD FACSMelody™ (BD Biosciences) |
| Software | Collection: BD FACSChorus™ software v 3.0 (BD Biosciences).<br>Analyses and visualisation: Microsoft Excel v 16.87 was used for data sorting and R v 4.0.2 was used for data visualization. |
| Cell population abundance | No cells were sorted. |
| Gating strategy | Briefly, background signals from the instrument and the buffer solution (PBS) were identified using the operational parameters forward scatter (FSC) and side scatter (SSC). Microbial cells were then displayed using the same settings in a scatter plot using the forward scatter (FSC) and side scatter (SSC) and pre-gated based on the presence of SYTO™ 9 signals. Singlets discrimination was performed. Absolute counting beads (CountBrightTM, ThermoFisher Scientific) added to each sample were used to determine the number of cells per mL of culture by following the manufacturer's instructions. Fluorescence signals were detected using the blue (488 nm – staining with SYTO™ 9 and CountBright™ beads) and yellow-green (561 nm - CountBrightTM beads only) optical lasers. The gated fluorescence signal events were evaluated on the forward–sideways density plot, to exclude remaining background events and to obtain an accurate microbial cell count. Instrument and gating settings were identical for all samples (fixed staining-gating strategy). |

☒ Tick this box to confirm that a figure exemplifying the gating strategy is provided in the Supplementary Information.

