## [Peer Review File · Nature Microbiology]

The Parkinson's disease drug entacapone disrupts gut microbiome homeostasis via iron sequestration.

Corresponding Author: Dr Fátima Pereira

Version 0:

Reviewer comments:

Reviewer #1

(Remarks to the Author)

The study by Pereira et al., focused on the impact of two Parkinson-targeting drugs on the gut microbiome of healthy young individuals. Using a wide array of methodological approaches comprising microbiome profiling, metagenomics, stable isotope probing, and chemical imaging coupled to ex vivo physiological experiments the authors aimed at unveiling the mechanisms by which the drugs entacapone and loxapine impact the gut microbiome and how these drugs affect microbial abundance, function, and diversity.

The findings are novel, the results well-structured and clearly described. The data shows that microbial metabolism of key members is strongly impacted by these drugs even at lower concentrations. The mechanism by which the drug entacapone affects the microbiome is shown to be related to the depletion of iron leading to selection of microorganisms with iron-scavenging capabilities.

The obtained data sets are valid and statistically relevant. Quality and quantity of data and its presentation is excellent. The results are fully supported by the data. The effort that the authors have invested in the study and obtaining the large data sets is commendable. The study's findings are novel with potential implications for human health.

The study results are relevant for microbial ecology of the gut, also potentially relevant for medical and pharmaceutical fields where considerations must be made for drugs treatments.

The conclusions are supported by the data to a high extent. There are a few speculative conclusions that this reviewer thinks are circumstantial (e.g. specific comment 9).

Overall, the study is novel and opens new perspectives into gut microbiome research, nevertheless it represents a snapshot into healthy individuals microbiome impacted when in direct contact to unmetabolized drugs targeting Parkinson disses.

Conceptually the study would have profited from inclusion of samples collected from individuals that suffer of Parkinson and are at different stages of treatment, with one of these drugs.

This reviewer has the following major concerns related to biological replication and significance of the data.

1. The significance of the results in the context of under treatment Parkinson patients: regular treatment involves the oral administration of the drugs which pass through the whole digestive tract and possibly suffer several chemical transformations until it reaches the gut microbiome. The current study shows the effect of these drugs added in a mixture as active substances dissolved in DMSO directly applied to the enriched fecal microbial samples obtained from healthy young individuals.
2. Biological replication was cancelled out by mixing the samples from the 6 individuals.
3. The relevance of the study with only healthy individuals included. In a medical study this group will be considered a "control" group. The target group usually includes individuals that suffer of Parkinson and are at different stages of treatment, with these drugs. Following few such patients from before they are taking the drugs and at different time intervals into their treatment would reflect the impact of these drugs metabolized by the body on the microbiome of diseased patients. How the effect of the drugs on gut microbiomes of healthy young subjects can relate to the effect on gut microbiomes of sick Parkinson subjects, which generally are much older?
4. What is the significance of the observed effect during a relatively short incubation (up to 24h) time when compared to the long-term treatments that Parkinson patients usually experience?

More specific comments:

1. are the high and low conc of the 2 drugs used here to follow the effect on the gut microbiome relevant for daily dosage the Parkinson patients usually take?
2. Lines 202-204: how the cross-sensitization to loxapine in the context of community should work?
3. Lines 223-225: authors should specify distinct short-term effect.
4. Lines 271-274: is there an explanation or speculation that authors can offer for these results?
5. Lines 275-276: do the authors suggest a lag response between activity and abundance?
6. Fig 3 b, d why the bioaccumulation of entacapone takes place only in random cells and not all cells show this accumulation? Are those cells showing bioaccumulation also highly enriched in D2O, e.g. highly active?
7. Lines 343-344: Is there any explanation for that?

8. 454-456: this speculation seems rather doubtful. How the Ruminococcus should import the siderophores from E.coli exactly? Is there a precedent that can be cited?
9. 474-476: the fact that the samples come from healthy individuals does not necessarily mean these have not been exposed throughout their life to antibiotics and pathogenic bacteria that will enrich their gut microbiome with AB resistance & virulence. The conclusion based on that seems rather circumstantial. The fact that at least one individual have an E.coli isolate resistant to AB that is able to overcome iron limitation does not mean the drug selects for such strains. Without biological replication this may be considered an overstatement.
10. 543-553: how lowering of entacapone conc locally by bioaccumulation (which seems to be a feature of many genera) will impact the activity/abundance of the other members of the microbiome?

Reviewer #2

(Remarks to the Author)

In their study, Pereira et al. explore the effects of two drugs targeting the nervous system on the signature and metabolic activity of the human gut microbiome ex vivo. Employing 16S rRNA sequencing, FACS-based cell counting, and long-read metagenomics, the researchers assess the drugs' impact on abundances across six healthy donors. Through stable isotope probing and single-cell chemical imaging, they pinpoint metabolic adaptations to drug exposure. Specifically, for entacapone, an antiparkinson's drug primarily excreted in feces, the authors demonstrate its bioaccumulation by certain taxa. Crucially, the authors elucidate the growth inhibitory effect of entacapone, revealing its ability to sequester iron, thereby limiting microbial growth. Whether these mechanisms are also at play in the intestines of patients remains to be established.

For me personally, the "mechanism of action" of entacapone, the inhibition of bacterial growth through iron binding, is the eye-opening, surprising novelty of this study.

The study is presented in a clear and accessible manner, with figures aiding comprehension. The methods employed are cutting-edge and apt for addressing the research questions, and the conclusions are mostly substantiated by the presented data. The results hold immediate interest for the microbiology community and extend to pharmacologists studying nervous system-targeting compounds.

While the study is generally suitable for publication in Nature Microbiology, several points should be addressed or clarified:

Major Concerns:

- Fig. 1 c/d. I assume that the data plotted here are the combined results across all six donors and all three replicates. How have these results been combined? Is this the mean, median or something else shown here? Is the observation described here true for each individual or were there differences between individuals? The same question applies to Figures 2, 3 and 4.
- The idea that E. coli_D enterobactin production allows it to grow under iron-limiting conditions induced by ENT-Hi is intriguing, but has not been convincingly demonstrated. So far, there is only an association between the presence of the enterobactin biosynthetic gene cluster and a bloom in the presence of high entacapone concentrations, not a causal relationship. The fact that the E.coli_D isolate can grow on iron-free minimal medium and is unaffected by ENT-Hi is not sufficient proof. It should be shown that in the absence of the enterobactin biosynthetic gene cluster (natural isolate or genetically engineered knockout in any E. coli strain), growth and/or metabolic activity/iron requirements are altered in the presence of entacapone.
- Line 500 - 508: "Our results show that loxapine succinate exert an effect on a broader range of taxa in the context of the community than its effects on microbes grown in isolation". One explanation, as the authors mention, is indeed cross-sensitisation to the drug in the community context. However, other possibilities include media differences (sM9/BHI in this study vs. mGAM in Maier et al., 2018) or differences between natural isolates at higher taxonomic levels, as in this study here, compared to type strains (in Maier et al., 2018). As there are alternative explanations for their observations, the authors should also discuss / mention these possibilities.

Minor Concerns:

- Figure 2b: Please add the number of cells analysed per condition. Was this done for all six donors?
- Line 313, figure legend 3a: It is unclear to me why the concentration here is 10 mM. Please either explain or delete.
- Line 318: Please correct: Single cell photothermal signal intensity distribution in samples shown in (b).
- Figure 4: Which medium was used for the faecal incubations? BHI or sM9?
- Figure 4b: How many independent replicates were tested? What does the error bar represent? SD, SE,...?
- Line 387: The given concentration of iron (31 μ M) refers to which medium?
- Figure 5d: This could be due to biases in knowledge and databases, with better annotation for Enterobacteriaceae compared to other taxa. This should be discussed/mentioned in the text.
- Line 547: over time
- Line 956: Why are monoculture growth assays performed in a different medium than faecal incubation assays? Would an increase in E. coli growth be observed with ENT-Hi in sM9 or BHI?

Reviewer #3

(Remarks to the Author)

In this manuscript the authors evaluate the impact of two clinically relevant nervous system targeting drugs on commensal microbes. The authors report that these drugs affect microbiome composition and abundance, and these experiments were performed across two distinct media increasing the rigor of these observations. Through the application of an innovative SRS-FISH assay the authors determined nervous system-targeting drugs alter microbial metabolism without impacting microbial abundance. Using photothermal imaging approaches, the authors report that entacapone bioaccumulates in microbial cells. Finally, the authors report that entacapone chelates iron and induces iron starvation in whole microbiome populations and

promotes growth of iron-scavenging *E. coli*. The authors put forward an exciting and innovative model but the presented data fall short of conclusively determining the mechanism by which entacapone inhibits microbial growth. I have the following suggestions to strengthen the conclusions and improve the readability of the manuscript.

1. The inclusion of the CAS assay is informative to evaluate the ability of entacapone to bind ferric iron. However, due to the focus of the paper on the gut microbiota, the binding of entacapone to ferrous iron should be evaluated, the predicted affinity is not convincing without experimental data to support this prediction.
2. The presented data do not distinguish between the possibility that entacapone inhibits microbial growth through iron sequestration vs. the possibility that iron binding to entacapone inhibits its antibacterial activity in a manner independent of microbial iron starvation. In this regard, the manuscript falls short of convincingly supporting the proposed model. To clearly show that entacapone inhibits microbial growth through nutrient iron restriction, a series of additional experiments are required. First, the authors should test whether entacapone exposure induces a gene expression profile in microbes of interest that is consistent with iron starvation. Second, the authors should measure iron levels within entacapone exposed bacteria. Finally, the authors should test if microbial strains defective in siderophore production (siderophore synthesis mutants) are more susceptible to entacapone than isogenic wildtype bacteria.
3. If the authors convincingly show that entacapone inhibits microbial growth through iron restriction, they should determine if entacapone bioaccumulation within microbes occurs through microbial siderophore receptors as this would significantly increase the impact of the work.

Decision Letter:

22nd January 2024

Dear Dr Pereira,

Thank you for your patience while your manuscript "The Parkinson's drug entacapone disrupts gut microbiome homeostasis via iron sequestration" was under peer-review at Nature Microbiology. It has now been seen by 3 referees, whose expertise and comments you will find at the end of this email. Although they find your work of some potential interest, they have raised a number of concerns that will need to be addressed before we can consider publication of the work in Nature Microbiology.

In this case, we are asking you to address the concerns by reviewers #2 and #3 experimentally. It will be particularly important to strengthen the mechanistic insights, including but not limited to strengthening the evidence that entacapone acts via iron sequestration (as mentioned by reviewer #3) and performing additional experiments in *E. coli* mutants to investigate the link between iron and enterobactin (as mentioned by reviewer #2). We are willing to overrule the request by reviewer #1 to include additional samples from PD patients and are therefore asking you to address the comments by reviewer #1 by textual edits and explanations.

Should further experimental data allow you to address these criticisms, we would be happy to look at a revised manuscript.

Please include a data availability statement as a separate section after Methods but before references, under the heading "Data Availability". This section should inform readers about the availability of the data used to support the conclusions of your study. This information includes accession codes to public repositories (data banks for protein, DNA or RNA sequences, microarray, proteomics data etc...), references to source data published alongside the paper, unique identifiers such as URLs to data repository entries, or data set DOIs, and any other statement about data availability. At a minimum, you should include the following statement: "The data that support the findings of this study are available from the corresponding author upon request", mentioning any restrictions on availability. If DOIs are provided, we also strongly encourage including these in the Reference list (authors, title, publisher (repository name), identifier, year). For more guidance on how to write this section please see: <http://www.nature.com/authors/policies/data/data-availability-statements-data-citations.pdf>

* If you have not done so already we suggest that you begin to revise your manuscript so that it conforms to our Article format

instructions at <http://www.nature.com/nmicrobiol/info/final-submission>. Refer also to any guidelines provided in this letter.

When submitting the revised version of your manuscript, please pay close attention to our [href="https://www.nature.com/nature-portfolio/editorial-policies/image-integrity">Digital Image Integrity Guidelines.](https://www.nature.com/nature-portfolio/editorial-policies/image-integrity) and to the following points below:

Link Redacted

Note: This url links to your confidential homepage and associated information about manuscripts you may have submitted or be reviewing for us. If you wish to forward this e-mail to co-authors, please delete this link to your homepage first.

Nature Microbiology is committed to improving transparency in authorship. As part of our efforts in this direction, we are now requesting that all authors identified as 'corresponding author' on published papers create and link their Open Researcher and Contributor Identifier (ORCID) with their account on the Manuscript Tracking System (MTS), prior to acceptance. This applies to primary research papers only. ORCID helps the scientific community achieve unambiguous attribution of all scholarly contributions. You can create and link your ORCID from the home page of the MTS by clicking on 'Modify my Springer Nature account'. For more information please visit www.springernature.com/orcid.

If you wish to submit a suitably revised manuscript we would hope to receive it within 6 months. If you cannot send it within this time, please let us know. We will be happy to consider your revision, even if a similar study has been accepted for publication at Nature Microbiology or published elsewhere (up to a maximum of 6 months).

Yours sincerely,

Consulting Editor
Nature Microbiology

Reviewer Expertise:

Referee #1: imaging, microbes
Referee #2: drug metabolism
Referee #3: iron metabolism

Reviewer Comments:

Reviewer #1 (Remarks to the Author):

The study by Pereira et al., focused on the impact of two Parkinson-targeting drugs on the gut microbiome of healthy young individuals. Using a wide array of methodological approaches comprising microbiome profiling, metagenomics, stable isotope probing, and chemical imaging coupled to ex vivo physiological experiments the authors aimed at unveiling the mechanisms by which the drugs entacapone and loxapine impact the gut microbiome and how these drugs affect microbial abundance, function, and diversity.

The findings are novel, the results well-structured and clearly described. The data shows that microbial metabolism of key members is strongly impacted by these drugs even at lower concentrations. The mechanism by which the drug entacapone affects the microbiome is shown to be related to the depletion of iron leading to selection of microorganisms with iron-scavenging capabilities.

The obtained data sets are valid and statistically relevant. Quality and quantity of data and its presentation is excellent. The results are fully supported by the data. The effort that the authors have invested in the study and obtaining the large data sets is commendable. The study's findings are novel with potential implications for human health.

The study results are relevant for microbial ecology of the gut, also potentially relevant for medical and pharmaceutical fields where considerations must be made for drugs treatments.

The conclusions are supported by the data to a high extent. There are a few speculative conclusions that this reviewer thinks are circumstantial (e.g. specific comment 9).

Overall, the study is novel and opens new perspectives into gut microbiome research, nevertheless it represents a snapshot into healthy individuals microbiome impacted when in direct contact to unmetabolized drugs targeting Parkinson diseases.

Conceptually the study would have profited from inclusion of samples collected from individuals that suffer of Parkinson and are at different stages of treatment, with one of these drugs.

This reviewer has the following major concerns related to biological replication and significance of the data.

1. The significance of the results in the context of under treatment Parkinson patients: regular treatment involves the oral administration of the drugs which pass through the whole digestive tract and possibly suffer several chemical transformations until it reaches the gut microbiome. The current study shows the effect of these drugs added in a mixture as active substances dissolved in DMSO directly applied to the enriched fecal microbial samples obtained from healthy young individuals.
2. Biological replication was cancelled out by mixing the samples from the 6 individuals.
3. The relevance of the study with only healthy individuals included. In a medical study this group will be considered a "control" group. The target group usually includes individuals that suffer of Parkinson and are at different stages of treatment, with these drugs. Following few such patients from before they are taking the drugs and at different time intervals into their treatment would reflect the impact of these drugs metabolized by the body on the microbiome of diseased patients. How the effect of the drugs on gut microbiomes of healthy young subjects can relate to the effect on gut microbiomes of sick Parkinson subjects, which generally are much older?
4. What is the significance of the observed effect during a relatively short incubation (up to 24h) time when compared to the long-term treatments that Parkinson patients usually experience?

More specific comments:

1. are the high and low conc of the 2 drugs used here to follow the effect on the gut microbiome relevant for daily dosage the Parkinson patients usually take?
2. Lines 202-204: how the cross-sensitization to loxapine in the context of community should work?
3. Lines 223-225: authors should specify distinct short-term effect.
4. Lines 271-274: is there an explanation or speculation that authors can offer for these results?
5. Lines 275-276: do the authors suggest a lag response between activity and abundance?
6. Fig 3 b, d why the bioaccumulation of entacapone takes place only in random cells and not all cells show this accumulation? Are those cells showing bioaccumulation also highly enriched in D₂O, e.g. highly active?
7. Lines 343-344: Is there any explanation for that?
8. 454-456: this speculation seems rather doubtful. How the Ruminococcus should import the siderophores from E.coli exactly? Is there a precedent that can be cited?
9. 474-476: the fact that the samples come from healthy individuals does not necessarily mean these have not been exposed throughout their life to antibiotics and pathogenic bacteria that will enrich their gut microbiome with AB resistance & virulence. The conclusion based on that seems rather circumstantial. The fact that at least one individual have an E.coli isolate resistant to AB that is able to overcome iron limitation does not mean the drug selects for such strains. Without biological replication this may be considered an overstatement.
10. 543-553: how lowering of entacapone conc locally by bioaccumulation (which seems to be a feature of many genera) will impact the activity/abundance of the other members of the microbiome?

Reviewer #2 (Remarks to the Author):

In their study, Pereira et al. explore the effects of two drugs targeting the nervous system on the signature and metabolic activity of the human gut microbiome *ex vivo*. Employing 16S rRNA sequencing, FACS-based cell counting, and long-read metagenomics, the researchers assess the drugs' impact on abundances across six healthy donors. Through stable isotope probing and single-cell chemical imaging, they pinpoint metabolic adaptations to drug exposure. Specifically, for entacapone, an antiparkinson's drug primarily excreted in feces, the authors demonstrate its bioaccumulation by certain taxa. Crucially, the authors elucidate the growth inhibitory effect of entacapone, revealing its ability to sequester iron, thereby limiting microbial growth. Whether these mechanisms are also at play in the intestines of patients remains to be established.

For me personally, the "mechanism of action" of entacapone, the inhibition of bacterial growth through iron binding, is the eye-opening, surprising novelty of this study.

The study is presented in a clear and accessible manner, with figures aiding comprehension. The methods employed are cutting-edge and apt for addressing the research questions, and the conclusions are mostly substantiated by the presented data. The results hold immediate interest for the microbiology community and extend to pharmacologists studying nervous system-targeting compounds.

While the study is generally suitable for publication in Nature Microbiology, several points should be addressed or clarified:

Major Concerns:

- Fig. 1 c/d. I assume that the data plotted here are the combined results across all six donors and all three replicates. How have these results been combined? Is this the mean, median or something else shown here? Is the observation described here true for each individual or were there differences between individuals? The same question applies to Figures 2, 3 and 4.
- The idea that E. coli_D enterobactin production allows it to grow under iron-limiting conditions induced by ENT-Hi is intriguing, but has not been convincingly demonstrated. So far, there is only an association between the presence of the enterobactin biosynthetic gene cluster and a bloom in the presence of high entacapone concentrations, not a causal relationship. The fact that the E.coli_D isolate can grow on iron-free minimal medium and is unaffected by ENT-Hi is not sufficient proof. It should be

shown that in the absence of the enterobactin biosynthetic gene cluster (natural isolate or genetically engineered knockout in any *E. coli* strain), growth and/or metabolic activity/iron requirements are altered in the presence of entacapone.

- Line 500 - 508: "Our results show that loxapine succinate exert an effect on a broader range of taxa in the context of the community than its effects on microbes grown in isolation". One explanation, as the authors mention, is indeed cross-sensitisation to the drug in the community context. However, other possibilities include media differences (sM9/BHI in this study vs. mGAM in Maier et al., 2018) or differences between natural isolates at higher taxonomic levels, as in this study here, compared to type strains (in Maier et al., 2018). As there are alternative explanations for their observations, the authors should also discuss / mention these possibilities.

Minor Concerns:

- Figure 2b: Please add the number of cells analysed per condition. Was this done for all six donors?
- Line 313, figure legend 3a: It is unclear to me why the concentration here is 10 mM. Please either explain or delete.
- Line 318: Please correct: Single cell photothermal signal intensity distribution in samples shown in (b).
- Figure 4: Which medium was used for the faecal incubations? BHI or sM9?
- Figure 4b: How many independent replicates were tested? What does the error bar represent? SD, SE,...?
- Line 387: The given concentration of iron (31 μ M) refers to which medium?
- Figure 5d: This could be due to biases in knowledge and databases, with better annotation for Enterobacteriaceae compared to other taxa. This should be discussed/mentioned in the text.
- Line 547: over time
- Line 956: Why are monoculture growth assays performed in a different medium than faecal incubation assays? Would an increase in *E. coli* growth be observed with ENT-Hi in sM9 or BHI?

Reviewer #3 (Remarks to the Author):

In this manuscript the authors evaluate the impact of two clinically relevant nervous system targeting drugs on commensal microbes. The authors report that these drugs affect microbiome composition and abundance, and these experiments were performed across two distinct media increasing the rigor of these observations. Through the application of an innovative SRS-FISH assay the authors determined nervous system-targeting drugs alter microbial metabolism without impacting microbial abundance. Using photothermal imaging approaches, the authors report that entacapone bioaccumulates in microbial cells. Finally, the authors report that entacapone chelates iron and induces iron starvation in whole microbiome populations and promotes growth of iron-scavenging *E. coli*. The authors put forward an exciting and innovative model but the presented data fall short of conclusively determining the mechanism by which entacapone inhibits microbial growth. I have the following suggestions to strengthen the conclusions and improve the readability of the manuscript.

1. The inclusion of the CAS assay is informative to evaluate the ability of entacapone to bind ferric iron. However, due to the focus of the paper on the gut microbiota, the binding of entacapone to ferrous iron should be evaluated, the predicted affinity is not convincing without experimental data to support this prediction.
2. The presented data do not distinguish between the possibility that entacapone inhibits microbial growth through iron sequestration vs. the possibility that iron binding to entacapone inhibits its antibacterial activity in a manner independent of microbial iron starvation. In this regard, the manuscript falls short of convincingly supporting the proposed model. To clearly show that entacapone inhibits microbial growth through nutrient iron restriction, a series of additional experiments are required. First, the authors should test whether entacapone exposure induces a gene expression profile in microbes of interest that is consistent with iron starvation. Second, the authors should measure iron levels within entacapone exposed bacteria. Finally, the authors should test if microbial strains defective in siderophore production (siderophore synthesis mutants) are more susceptible to entacapone than isogenic wildtype bacteria.
3. If the authors convincingly show that entacapone inhibits microbial growth through iron restriction, they should determine if entacapone bioaccumulation within microbes occurs through microbial siderophore receptors as this would significantly increase the impact of the work.

Version 1:

Reviewer comments:

Reviewer #1

(Remarks to the Author)

Upon careful review of this revised version of the manuscript by Pereira et al., I consider that all my major concerns regarding i) the biotransformation of the entacapone and loxapine that may occur post-drug absorption; ii) biological replication iii) relevance of the target group considering that only healthy individuals were considered in the study; IV) significance considering the short timeframe of maximum 24h have been addressed and answered, either by plausible explanations supported by references or by addition of supplementary results and profs.

I consider the study excellently suited for the publication in Nature Microbiology. I would like to acknowledge the authors efforts to improve the initial work by considering in detail all reviewers comments and by carried out several additional experiments during revision.

Reviewer #2

(Remarks to the Author)

The authors have done an exceptional job addressing my feedback. The revised manuscript is well-suited for publication in Nature Microbiology, and I eagerly await its publication.

Reviewer #3

(Remarks to the Author)

The authors performed a significant amount of new experimentation to address my prior concerns. I think the inclusion of these new data strengthens the conclusions and supports the authors' model. I have no further suggestions for improvement. This is an exciting paper that will be of broad significance.

Decision Letter:

Our ref: NMICROBIOL-23113287A

13th August 2024

Dear Dr. Pereira,

Thank you for submitting your revised manuscript "The Parkinson's drug entacapone disrupts gut microbiome homeostasis via iron sequestration." (NMICROBIOL-23113287A). It has now been seen by the original referees and their comments are below. The reviewers find that the paper has improved in revision, and therefore we'll be happy in principle to publish it in Nature Microbiology, pending minor revisions to satisfy the referees' final requests and to comply with our editorial and formatting guidelines.

We are now performing detailed checks on your paper and will send you a checklist detailing our editorial and formatting requirements in a few weeks. Please do not upload the final materials and make any revisions until you receive this additional information from us.

Thank you again for your interest in Nature Microbiology Please do not hesitate to contact me if you have any questions.

Sincerely,

Emily

Emily White, PhD
Chief Editor
Nature Microbiology

4 Crinan Street, London, UK, N1 9XW
+44 207 418 5601
emily.white@nature.com

orcid.org/0000-0002-2314-5718

Reviewer #1 (Remarks to the Author):

Upon careful review of this revised version of the manuscript by Pereira et al., I consider that all my major concerns regarding i) the biotransformation of the entacapone and loxapine that may occur post-drug absorption; ii) biological replication iii) relevance of the target group considering that only healthy individuals were considered in the study; IV) significance considering the short timeframe of maximum 24h have been addressed and answered, either by plausible explanations supported by references or by addition of supplementary results and profs.

I consider the study excellently suited for the publication in Nature Microbiology. I would like to acknowledge the authors efforts to improve the initial work by considering in detail all reviewers comments and by carried out several additional experiments during revision.

Reviewer #2 (Remarks to the Author):

The authors have done an exceptional job addressing my feedback. The revised manuscript is well-suited for publication in Nature Microbiology, and I eagerly await its publication.

Reviewer #3 (Remarks to the Author):

The authors performed a significant amount of new experimentation to address my prior concerns. I think the inclusion of these new data strengthens the conclusions and supports the authors' model. I have no further suggestions for improvement. This is an exciting paper that will be of broad significance.

Version 2:

Decision Letter:

10th October 2024

Dear Dr Pereira,

I am pleased to accept your Article "The Parkinson's disease drug entacapone disrupts gut microbiome homeostasis via iron sequestration." for publication in Nature Microbiology. Thank you for having chosen to submit your work to us and many congratulations.

Please note that *Nature Microbiology* is a Transformative Journal (TJ). Authors may publish their research with us through the traditional subscription access route or make their paper immediately open access through payment of an article-processing charge (APC). Authors will not be required to make a final decision about access to their article until it has been accepted. [Find out more about Transformative Journals](https://www.springernature.com/gp/open-research/transformative-journals)

Authors may need to take specific actions to achieve [compliance](https://www.springernature.com/gp/open-research/funding/policy-compliance-faqs) with funder and institutional open access mandates. If your research is supported by a funder that requires immediate open access (e.g. according to [Plan S principles](https://www.springernature.com/gp/open-research/plan-s-compliance)) then you should select the gold OA route, and we will direct you to the compliant route where possible. For authors selecting the subscription publication route, the journal's standard licensing terms will need to be accepted, including [self-archiving policies](https://www.nature.com/nature-portfolio/editorial-policies/self-archiving-and-license-to-publish). Those licensing terms will supersede any other terms that the author or any third party may assert apply to any version of the manuscript.

With kind regards,

Nature Microbiology

P.S. Click on the following link if you would like to recommend Nature Microbiology to your librarian
<http://www.nature.com/subscriptions/recommend.html#forms>

** Visit the Springer Nature Editorial and Publishing website at http://editorial-jobs.springernature.com?utm_source=ejP_NMicro_email&utm_medium=ejP_NMicro_email&utm_campaign=ejp_NMicro for more information about our career opportunities. If you have any questions please click [here](mailto:editorial.publishing.jobs@springernature.com). **

Open Access This Peer Review File is licensed under a Creative Commons Attribution 4.0 International License, which permits use, sharing, adaptation, distribution and reproduction in any medium or format, as long as you give appropriate credit to the original author(s) and the source, provide a link to the Creative Commons license, and indicate if changes were made. In cases where reviewers are anonymous, credit should be given to 'Anonymous Referee' and the source. The images or other third party material in this Peer Review File are included in the article's Creative Commons license, unless indicated otherwise in a credit line to the material. If material is not included in the article's Creative Commons license and your intended use is not permitted by statutory regulation or exceeds the permitted use, you will need to obtain permission directly from the copyright holder.

Response to Reviewers

We thank all reviewers for their valuable comments, which we used to improve our manuscript. To address the points of the reviewers we have performed a large set of additional experiments comprising for example (i) faecal incubations with entacapone from additional donors, (ii) SRS-based activity measurement of entacapone-accumulating microbes, (iii) chemical experiments to identify whether Fe(II) and/or Fe(III) is complexed by entacapone, (iv) NanoSIMS-analyses of iron-accumulation in faecal microbiome members and (v) *E. coli* mutant analyses. These extensive additional experiments were performed with the help of four new co-authors.

Below, we have copied the reviewers' comments verbatim and provide point-by-point responses and a description of the modifications that we have made to the manuscript. All answers to reviewers' comments are shown in blue. Furthermore, all modifications of the manuscript's original text are highlighted in yellow in the revised manuscript version.

Reviewer Expertise:

Referee #1: imaging, microbes

Referee #2: drug metabolism

Referee #3: iron metabolism

Reviewer Comments:

Reviewer #1 (Remarks to the Author):

The study by Pereira et al., focused on the impact of two Parkinson-targeting drugs on the gut microbiome of healthy young individuals. Using a wide array of methodological approaches comprising microbiome profiling, metagenomics, stable isotope probing, and chemical imaging coupled to ex vivo physiological experiments the authors aimed at unveiling the mechanisms by which the drugs entacapone and loxapine impact the gut microbiome and how these drugs affect microbial abundance, function, and diversity.

The findings are novel, the results well-structured and clearly described. The data shows that microbial metabolism of key members is strongly impacted by these drugs even at lower concentrations. The mechanism by which the drug entacapone affects the microbiome is shown to be related to the depletion of iron leading to selection of microorganisms with iron-scavenging capabilities.

The obtained data sets are valid and statistically relevant. Quality and quantity of data and its presentation is excellent. The results are fully supported by the data. The effort that the authors have invested in the study and obtaining the large data sets is commendable. The study's findings are novel with potential implications for human health.

The study results are relevant for microbial ecology of the gut, also potentially relevant for medical and pharmaceutical fields where considerations must be made for drugs treatments.

The conclusions are supported by the data to a high extent. There are a few speculative conclusions that this reviewer thinks are circumstantial (e.g. specific comment 9).

Overall, the study is novel and opens new perspectives into gut microbiome research, nevertheless it represents a snapshot into healthy individuals microbiome impacted when in direct contact to unmetabolized drugs targeting Parkinson disses. Conceptually the study would have profited from inclusion of samples collected from individuals that suffer of Parkinson and are at different stages of treatment, with one of these drugs.

We thank the reviewer for the positive feedback, and the suggestions on how to improve the manuscript. We fully agree that analysing samples from Parkinson's and schizophrenia patients would provide additional insights on how these drugs affect the gut microbiome, but this would require at least a year of additional work (ethics, cohort building, actual analysis) and we feel that this is beyond the scope of this study.

This reviewer has the following major concerns related to biological replication and significance of the data.

We provide clarifications to all the specific points raised by the reviewer below partly based on new experimental data that are included in the revised manuscript.

1. The significance of the results in the context of under treatment Parkinson patients: regular treatment involves the oral administration of the drugs which pass through the whole digestive tract and possibly suffer several chemical transformations until it reaches the gut microbiome. The current study shows the

effect of these drugs added in a mixture as active substances dissolved in DMSO directly applied to the enriched faecal microbial samples obtained from healthy young individuals.

Thank you for this point and we understand the concern of the reviewer. However, it should be noted that we did not add these drugs in a mixture, each drug (loxapine or entacapone) was separately supplemented to the faecal slurry.

A crucial step in the development of safe and effective drugs is a careful assessment of a drug's ADMET (absorption, distribution, metabolism, excretion, toxicity) properties. In the case of orally delivered drugs, this includes an evaluation of various physicochemical properties (e.g. lipophilicity, acid stability) of the drug in the gastrointestinal tract, as these dramatically affect the amount of drug absorbed and its efficacy (Hua, 2020). The fact that entacapone and loxapine have been approved for clinical use strongly indicates that their pharmacokinetics profile is favourable to its effect. Indeed, orally delivered entacapone is rapidly absorbed in the small intestine and the parental form accumulates in the plasma in a dose-dependent manner, reaching concentrations up to $1.2 \mu\text{g}\cdot\text{ml}^{-1}$ one hour after oral administration (Keränen *et al.*, 1994). The (Z)-isomer of entacapone is the only metabolite found in addition to entacapone in human plasma (Wikberg *et al.*, 1993; Keränen *et al.*, 1994). Isomerization of entacapone occurs after the intake of entacapone since the entacapone tablets contain less than 0.2 % of the (Z)-isomer. However, the (Z)-isomer accounts for only about 5 % of the total area under the curve of both isomers in plasma, indicating that most entacapone is absorbed in the small intestine in its parental form. Loxapine reaches a peak plasma concentration of $16\text{-}18 \text{ ng}\cdot\text{ml}^{-1}$ two hours after oral delivery (Simpson *et al.*, 1978). The hydroxylated metabolite 8-OH-loxapine, and to a lower extent the N-demethylated metabolite amoxapine are also found in plasma at concentrations of approximately 30 and $2 \text{ ng}\cdot\text{ml}^{-1}$, respectively (Simpson *et al.*, 1978). Hepatic cytochrome P450 enzymes are responsible for the oxidative metabolism of loxapine to 8-hydroxyloxapine and amoxapine (Luo *et al.*, 2011), indicating that this biotransformation occurs post-drug absorption, *i.e.*, outside of the gastrointestinal tract.

Thus, both drugs can be found in its parent form in plasma following absorption in the small intestine. Metabolites of these drugs are found in plasma, but at very low levels (entacapone) or are the product of liver metabolism (loxapine) - and thus result from metabolism/transformations outside of the GI tract, indicating low level of chemical transformation in the upper gastrointestinal tract. However, it is possible that some degree of drug transformation occurs *in vivo* before reaching the large intestine, due to interactions with microbes or dietary compounds. For instance, microbially mediated nitroreduction of entacapone has been reported (Zimmerman *et al.*, 2019). Of note, nitroreduction is not expected to alter the entacapone catechol moiety necessary for iron chelation. The presence and extent of chemical or microbial transformations between the small and large intestines are challenging to replicate in an *in vitro* study like ours. To accurately determine these transformations, access to human small intestinal communities, such as ileostomy effluents, would be necessary, though currently, this remains limited.

This point is now mentioned in results section the revised manuscript, please see lines 140-144 or below.

"Both drugs are found in their parent form in human plasma samples^{30,31}. Some drug metabolites have also been shown to be present, but these are largely products of liver metabolism³², which is an indication of low levels of chemical transformation of these drugs in the upper gastrointestinal tract prior to absorption."

2. Biological replication was cancelled out by mixing the samples from the 6 individuals.

Our rationale for mixing samples from 6 healthy donors was to assess the effect of drugs on a microbial community with greater microbial diversity than that of a single individual. First, we would like to highlight that the effect of both drugs on microbial growth and community composition was consistent when using two different media: sM9 and BHI (Figures 1 and Extended Data Fig. 1). We have further demonstrated replication of the effect of entacapone when we tested if replenishing iron levels could rescue the effect of entacapone, as shown in Figure 4. For this experiment of Figure 4, we also used a mix of faecal samples, but the experiment was conducted independently (with new samples) of the original experiment in Figure 1. We did observe a decrease in the abundance of key microbial taxa such as *Bacteroides* or *Clostridium* species upon entacapone treatment and an expansion of *Escherichia* spp, in agreement with results shown in Fig.1 and Extended Data Fig. 1.

To confirm that the observed effect was not only present when the community was a combination of samples from different donors, we performed additional experiments assessing the effect of supplementing entacapone to individual samples from different donors, more precisely in three donors that did not take part in the initial experiments. Results demonstrate that entacapone supplementation significantly reduces microbial loads in samples from all three donors tested individually (Extended Data Fig.2a, please see also below). These results further confirm that entacapone dramatically remodels the microbiome community of

all three donors and promotes an expansion of *Escherichia-Shigella* and other *Enterobacteriaceae* species, similarly to what we show in Fig.1 and Fig.4. This expansion is subtle in Donor 1, but quite pronounced in Donors 2 and 3. These results are shown in Extended Data Fig.2 and Supplementary Table 6 of the revised manuscript.

Extended Data Fig. 2. Relative and absolute abundance profiles of individual faecal samples incubated with entacapone. **a.** Total microbial cell loads in incubations of samples from three different donors (compared to samples used in main text, e.g., Figure 1) with entacapone (ENT-Hi) and controls (No drug - DMSO only), at the start of the incubation (time=0 hours, t0) and after 6 and 24 hours in sM9 medium. Total cell loads were assessed by flow cytometry. *p<0.05, **p<0.01; unpaired two-sample t-test. Data points represent the mean (n=3). Error bars represent the standard error of the mean. **b, c.** Relative (b) and absolute (c) genus abundance profiles of faecal microbial communities (Donor 1: D1, Donor 2: D2 and Donor 3: D3) incubated with ENT-Hi and assessed by 16S rRNA gene amplicon sequencing. The community composition at 0 hours (inoculum) is also shown. All genera present at a relative abundance below 0.025 or absolute abundance below 5x10⁶ cells.ml⁻¹ were assigned to the category “Other”. Each bar represents the mean from triplicates.

3. The relevance of the study with only healthy individuals included. In a medical study this group will be considered a “control” group. The target group usually includes individuals that suffer of Parkinson and are at different stages of treatment, with these drugs. Following few such patients from before they are taking the drugs and at different time intervals into their treatment would reflect the impact of these drugs metabolized by the body on the microbiome of diseased patients. How the effect of the drugs on gut microbiomes of healthy young subjects can relate to the effect on gut microbiomes of sick Parkinson subjects, which generally are much older?

Thank you for this point and we understand the concern of the reviewer. Our rationale for selecting healthy adult individuals was to expose drug-naïve faecal microbiomes - that never have encountered these drugs - to capture the drug’s full potential to modify the microbiome and to simulate initiation of treatment with entacapone or loxapine. We fully agree that analysing samples from Parkinson’s and schizophrenia patients would provide additional insights on how these drugs affect gut microbiomes that were previously exposed to the drugs, but this would require at least a year of additional work (ethics, cohort building, actual analysis) and we feel that this is beyond the scope of this study.

Although there is conflicting data on the microbiome structure between PD patients and controls (Romano *et al.*, 2021), we agree that PD patients are typically older and may have a distinct microbiome. We fully agree that following PD patients would be important to evaluate the full potential of entacapone to modulate the microbiome. This is something we intend to address in future work. This point is now mentioned in the discussion of the revised manuscript, please see lines 599-603, or below.

“Our results are consistent with these findings, although they are based on short-term incubations of drugs with microbiomes of healthy individuals simulating the initiation of treatment. For a more robust assessment of the long-term effects of entacapone on the microbiome, it would be relevant to follow a cohort of Parkinson's patients before they start taking the drug and at different time intervals during their treatment.”

4. What is the significance of the observed effect during a relatively short incubation (up to 24h) time when compared to the long-term treatments that Parkinson patients usually experience?

The main point of our study was to understand what happens when a community first encounters entacapone (initiation of treatment) and to elucidate the mechanism of action of the drug on the microbiome, hence the short incubation times chosen. Our *ex vivo* system would not allow us to preserve the diversity of the microbiome for longer periods, where we could test repeated administration of drugs. Multi-bioreactor systems are available to investigate the effects of various factors on the gut microbiome for longer periods, but even optimised systems typically lead to similar issues, *i.e.* to an altered microbial community profile and loss of diversity compared to faecal sample communities (Jin *et al.*, 2022).

We believe the long-term impact of the drug on the microbiome would be better evaluated *in vivo*, in a cohort study including longitudinal samples from PD patients before taking entacapone, at the start of treatment and at regular intervals during treatment, compared with PD patients not taking the drug - as suggested by the reviewer in point 3. Of note, entacapone is not taken in isolation, it is rather prescribed in combination with levodopa and certain cases, carbidopa or benserazide. There could also be an adaptation of the microbiome to these PD drugs and other drugs, *e.g.*, laxatives or sleeping aids which are commonly taken by PD patients, and hence this investigation would make more sense in a cohort. The major advantage of our study design is that it tests the effects of two drugs on a complex microbial community without additional confounding effects, such as co-medications. Reductionist approaches like ours have been followed by others to successfully prove the mechanistic basis of interactions between drugs and the microbiome (Rekdal *et al.*, 2019, Klünemann *et al.*, 2021). In our case, this approach was proven successful to identify a mechanism of action for entacapone on the microbiome.

This point is now mentioned in the discussion of the revised manuscript, please see lines 599-603 or below.

“Our results are consistent with these findings, although they are based on short-term incubations of drugs with microbiomes of healthy individuals simulating the initiation of treatment. For a more robust assessment of the long-term effects of entacapone on the microbiome, it would be important to follow a cohort of Parkinson's patients before they start taking the drug and at different time intervals during their treatment.”

More specific comments:

1. are the high and low conc of the 2 drugs used here to follow the effect on the gut microbiome relevant for daily dosage the Parkinson patients usually take?

Drug concentrations are typically measured in blood/plasma and sometimes in urine, and very scarce data on faecal concentrations for drugs is available. The “low” concentration we chose - 20 μM - was the concentration used by a large screen testing the effect of hundreds of drugs on gut microbes (Maier *et al.* 2018). The “high” drug doses in our study were also based on the study from Maier *et al.*, as the authors later estimated concentrations for many drugs in the large intestine by taking into account the typical dose regimen and available pharmacokinetics parameters for each drug. For entacapone, 1965 μM was the estimated colon concentration by Maier *et al.* based on an oral dose of 655.1 μmol , which is the equivalent to 200 mg of entacapone. The recommended dose of entacapone is 200 mg, one to eight times a day^a. Thus, depending on the dose frequency, entacapone may reach even higher concentrations than what we have here considered. The estimated colon concentration was calculated considering the 200 mg dose and available pharmacokinetic parameters and excretion profile for entacapone, and assuming an average intestinal transit time of 24 hours and a colon volume of 0.6 L (Maier *et al.*, 2018).

For loxapine, the oral dose considered by Maier *et al.* was 56.06 μmole , equivalent to 18.3 mg of loxapine. The oral dose of loxapine for adults is up to 18.2 mg daily^b. Due to a lack of data on some of the pharmacokinetics of loxapine, Maier *et al.* were unable to estimate faecal loxapine concentrations. We predicted that loxapine would reach similar colon concentrations as its chemical and therapeutic analogues amoxapine and clozapine, which are prescribed in similar doses and were estimated to reach colon concentrations of 138 μM and 153 μM , respectively (estimated by Maier *et al.* following the same protocol as described for entacapone). Using these values as a reference, we predict that loxapine is present in the colon at concentrations of at least 100 μM . This value is close to the median colon concentration (82.9 μM) reached for all human-targeted drugs tested by Maier *et al.* Thus, we are convinced that the “high” concentrations here used for both drugs are physiologically relevant and reflect doses prescribed by clinicians.

This point is more carefully explained in materials and methods section of the manuscript, please see lines 705-712 or below.

“The colon concentration estimated for entacapone is 1965 μM ¹, which is based on an oral dose of 200 mg. For loxapine no estimate of its large concentration is available. We predicted that loxapine would reach similar colon concentrations as its chemical and therapeutic analogues amoxapine and clozapine, prescribed at similar doses (10-20 mg daily) and estimated to reach colon concentrations of 138 μM and 153 μM , respectively¹. Using these values as a reference, we predict that loxapine succinate should be present in the colon at concentrations of at least 100 μM and chose it as the LOX-Hi concentration.”

^a "Comtan: EPAR – Product Information" (PDF). European Medicines Agency. 2015. Retrieved 17 April 2017.

^b "Adasuve: EPAR – Product Information" (PDF). European Medicines Agency. 2013. Retrieved 02 July 2024.

2. Lines 202-204: how the cross-sensitization to loxapine in the context of community should work?

We thank the reviewer for this valid point. We envision major causes would be bioaccumulation of the drug by species “X”, reducing the drug concentration faced by its neighbouring species “Y”, or biotransformation of the drug by species “X” into a drug-derivative with increased toxicity towards species “Y”. Cross-sensitisation induced by drugs in the context of a community may also arise from drug-induced toxic stress responses in species “X”, which leads to changes in secreted metabolites involved in interspecies interaction with species “Y” (Yu *et al.*, 2022). In addition, as reviewer 2 pointed out, we cannot rule out that the different responses to loxapine in microbial strains grown in isolation vs. when present in the faecal community are due to *i*) differences in growth conditions between our study and the study from Maier *et al.*, including the media used (sM9/BHI in this study vs. mGAM in Maier *et al.*); or *ii*) differences in the genomic content of strains present in our samples compared to type strains used in the study of Maier *et al.*, 2018. This point is more carefully explained in the discussion of the revised manuscript, please see lines 569-577 or below.

“One way by which some taxa may sensitize or protect others to a particular drug is through chemical conversion or accumulation of the drug¹⁰. However, previous studies have shown that the gut microbiota does neither significantly bioaccumulate nor transform loxapine succinate¹⁰. Thus, we presume that cross-sensitization to loxapine is likely due to drug-induced changes in microbial metabolites that are involved in interspecies interactions. These differences may, however, not be entirely related to cross-sensitization, but instead due to variations in growth conditions or in the genomic content of strains present in the faecal sample compared to the type strains tested in previous studies¹.”

3. Lines 223-225: authors should specify distinct short-term effect.

We thank the reviewer for the remark. The text has been changed to “strong but distinct short-term effect”. Please see line 229.

4. Lines 271-274: is there an explanation or speculation that authors can offer for these results?

*“We speculate that only a few drug-resistant *P. dorei* cells remained active enough to be detected by FISH, and these cells were not strongly impacted in activity (Fig.2c,d).”*

Thank you for allowing us to explain in more detail as we did not have enough space to elaborate on this in the main manuscript. There are several plausible explanations for the low impact on the activity of some *P. dorei* cells following drug supplementation. One possibility is phenotypic heterogeneity within *P. dorei*. This heterogeneity in isogenic bacterial populations can arise from internal factors such as noise and stochastic gene expression, or from external factors like chemical gradients (Gasperotti *et al.*, 2020). Indeed, it is

possible that *P. dorei* cells experienced varying local drug concentrations in different parts of the sample, as our incubations were conducted without agitation. An additional explanation is that more than one *P. dorei* strain is targeted by the FISH probe. Indeed, the FISH probe we developed is specific for *P. dorei* but it targets three additional 16S rRNA sequences in our dataset that are closely related to the 16S rRNA gene sequence of the *P. dorei* genome, as retrieved by metagenomic analyses (Supplementary Table 10). It is possible that these 16S rRNA sequences covered by the probe belong to species with slightly different physiologies. This is now explained in the manuscript, lines 278-280:

“This observation could be explained by either phenotypic heterogeneity⁴² within the *P. dorei* isogenic population, or the FISH probe used, which may target multiple *P. dorei* strains with different physiologies.”

5. Lines 275-276: do the authors suggest a lag response between activity and abundance?

There is indeed a more pronounced decrease in the abundance of targeted taxa such as *B. uniformis*, *Clostridium sp900539375* or *Streptococcus* spp. at 24 hours in LOX treated samples compared to controls compared to 6 hours. This suggests a lag between activity and abundance. The Log₂FC in abundance for this and other taxa of interest at 24 hours is now included in the revised manuscript, Extended Data Figure 5 and we refer to this data in manuscript lines 280-286 and 562-564:

“A comparable decrease in activity was detected for *Streptococcus* spp., but in this case this decrease was surprisingly accompanied by a slight increase in abundance at 6 hours (Fig.2d,e). However, from 6 to 24 hours, we detect a decrease of the population represented by the *Clostridium sp900539375*, *B. uniformis* and *Streptococcus lutetiensis* MAGs in the presence of LOX compared to the control (Fig. 1e,f, Extended Data Fig.5), which could be an effect of the lower activities detected by SRS in the LOX conditions at 6 hours.”

“For instance, we did detect reduced *B. uniformis* and *Clostridium sp900539375* activities at 6 hours, but their abundance is not strongly affected until 24 hours (Fig.1e, Fig 2d, Extended Data Fig.5). This is a major advantage of our SRS-FISH approach as it captures drug-induced changes in short incubation times, which are ideal for ex vivo systems like the one in use.”

Extended Data Fig.5. Bubble plot denoting the fold change (FC) in absolute abundances for the taxa targeted by FISH and incubated with drugs relative to “No drug” incubations, as determined by DeSeq2, at 24 hours of incubation is sM9. Original data is from triplicates per condition.

6. Fig 3 b, d why the bioaccumulation of entacapone takes place only in random cells and not all cells show this accumulation?

We would like to thank the reviewer for this interesting question. Figure 3b shows randomly selected cells from the complex community (originating from various taxa). Thus, the large distribution in Fig. 3b is related to the fact that in a complex community, some taxa accumulate the drug, while others do not. Even different strains of *E. coli* have varying capabilities of bioaccumulating drugs, as reported for other drugs like duloxetine (Klünemann *et al.*, 2021).

Within the same taxon (Fig. 3d), this distribution could be related to phenotypic heterogeneity or external factors such as different local levels of entacapone encountered by different cells – as our incubations were not carried out with agitation to better mimic conditions in the large intestine. It could be also because our FISH probes target more than one strain/species, despite our efforts to use probes as specific as possible

for the taxa of interest (Sup. Table 10). We would also like to note that the fact that we cannot detect entacapone in certain cells does not mean these do not accumulate it – they may still do so, but it could be below our limit of detection.

We now address the heterogeneity in entacapone levels in the manuscript discussion lines 622-627 and below:

“Interestingly, using single-cell chemical imaging we further show that this bioaccumulation is heterogeneous, with some cells within a particular taxon accumulating varying amounts of the drug. This can be explained by phenotypic heterogeneity due to stochastic gene expression or to different local levels of entacapone encountered by different cells, as our incubations were performed without agitation to better mimic the gut environment.”

Are those cells showing bioaccumulation also highly enriched in D2O, e.g. highly active?

We implemented a new protocol to measure activity (as %CD_{SRS}) and entacapone levels in the same cell. This was previously not possible because entacapone has a contribution to the lock-in amplifier x channel measurement that affected accurate %CD_{SRS} activity measurements for cells exposed to ENT-Hi. However, by bleaching the entacapone photothermal (PT) signal in cells in a field of view by continuous scanning after the initial PT measurement, we were able to eliminate the contribution from entacapone to the x channel and determine activity levels for each cell with accuracy. Results demonstrate a weak negative correlation ($r = -0.17$, Pearson's correlation) between entacapone levels and single-cell activity. Cells accumulating high levels of entacapone display medium to low levels of %CD_{SRS}.

These results are now included in Fig. 3, panel d, and below. We have also added a section to the Supplementary Information Text (lines 60 to 76, Supplementary Information) and an accompanying figure (Extended data Fig. 7) with details and feasibility of this new protocol. We would like to point out that these data were acquired using a newly prepared sample mix from 2 individuals, as we were no longer able to detect strong SRS signals in the original samples used to acquire the data presented in Figures 1, 2 and 3, probably due to signal degradation over time. %CD_{SRS} and PT signal intensity detected in these newly prepared samples is comparable to samples previously analysed (Figs.2b and 3c).

Figure 3d. Correlation between levels of activity (%CD_{SRS}) and PT signal intensity for microbiota cells exposed to ENT-Hi for 6 hours in the presence of 50% D₂O. Pearson correlation coefficient (-0.17) and p-value are indicated. Incubations were established using a new mixture of samples from two donors in sM9.

Extended Data Fig.7. Recorded signal intensity of the lock-in amplifier x and y channels overtime in faecal samples subjected to continuous bleaching of the entacapone photothermal signal. a. Signal intensity of the lock-in amplifier y channel (corresponding to photothermal signal) in samples incubated with ENT-HI and D₂O. **b.** Signal intensity of the SRS x channel (corresponding to SRS signal and its background) in samples incubated with ENT-HI and D₂O. **c.** Signal intensity of the SRS x channel (corresponding to SRS signal and its background) in samples incubated with no drug, but with D₂O. Each dot represents the average signal intensity of 20 microbiota cells.

7. Lines 343-344: Is there any explanation for that?

Lines 343-344 in the previous version of the manuscript read as follows: “Interestingly, while entacapone accumulation drastically inhibited the growth of *P. dorei* as a microbiome community member and in pure culture, it did not affect *Streptococcus* growth in the community to the same extent and showed even growth promotion for *E. coli*”.

We thank the reviewer for the interesting point. Bioaccumulation of drugs by microbes has been shown to reduce and even inhibit growth of certain bioaccumulating species, while not affecting or even promoting growth in others (Klünemann *et al.*, 2021). These effects are thought to be mediated by the capacity of drugs (e.g. duloxetine) to bind key metabolic enzymes in the cell (amino acid metabolism or purine and pyrimidine synthesis, for instance) and by doing so alter cellular physiology and metabolism (Klünemann *et al.*, 2021). The reason why in some cases this results in growth inhibition and others in no effect or even growth promotion is a very interesting question, but it is outside of the scope of this manuscript. Using an *E. coli* isolated from our incubations, we show in our manuscript that entacapone does not appear to directly promote *E. coli* growth (Fig. 6b). We thus believe that the entacapone-induced increase in *E. coli* growth in the community is likely due to a greater availability of nutrients released from dead cells of taxa negatively affected by entacapone, or due to cross-feeding. Of note, drugs such as duloxetine have been also shown to alter community behaviour, promoting the expansion of particular taxa via cross-feeding (Klünemann *et al.*, 2021).

We write this in the discussion, lines 647-655:

“Among stimulated taxa, the siderophore-producing *E. coli*_D strain present in our incubations greatly benefitted from entacapone’s presence, but only in the context of the community, as entacapone supplementation to an isolate alone did not cause any significant boost in *E. coli* growth (Fig. 6b). Thus, taxa stimulated by entacapone likely acquired iron via the mechanisms mentioned above, or via siderophores, and expanded in the community at the cost of nutrients released by dead cells or supplied via cross-feeding from other species. Treatment of communities with other iron complexing agents has shown somewhat similar effects on the gut community^{71,72}.”

8. 454-456: this speculation seems rather doubtful. How the *Ruminococcus* should import the siderophores from *E. coli* exactly? Is there a precedent that can be cited?

The import and utilisation of siderophores produced by other bacteria (e.g. enterobactin and salmochelin) is well documented for the gut commensal *Bacteroides thetaiotaomicron* (Zhu *et al.*, 2020). *Ruminococcus gnavus* species from Inflammatory Bowel Disease patients have been shown to be enriched in iron transport genes and have an enterochelin/enterobactin (siderophore) esterase which is required to release ferric iron from iron-enterobactin complexes (Hall *et al.*, 2017). This suggests that some *Ruminococcus* spp. may have enhanced capacity for iron acquisition that also involve the import and release of iron from xenosiderophores. This potentially confers these strains an adaptive advantage in certain gut environments, e.g., during inflammation. We have inspected our *Ruminococcus* MAG using FeGenie, a tool for the identification of iron acquisition genes in genomes (Garber *et al.*, 2020; Sup. Table 16). This revealed the presence of several ferrous and ferric iron transporters, as expected, but also of a YfeB family membrane protein, which is involved in chelated iron transport in *Yersinia pestis* and other organisms (Bearden *et al.*, 1998). For these reasons, we hypothesize that *Ruminococcus* may have specialised iron transport systems that enable its growth in iron-limited environments. We have re-written the sentence to better reflect this additional information, manuscript lines 509-515 or below.

“Interestingly, we could not find any genes involved in the production of known siderophores in the MAG of the Ruminococcus strain thriving under ENT-Hi conditions, but we did find a gene encoding an orthologue of the membrane protein YfeB required for chelated iron transport in Yersinia pestis⁵⁶ (Supplementary Table 16). We therefore speculate that Ruminococcus expands in the entacapone supplemented medium by importing iron-loaded siderophores or other iron-chelating molecules, or by producing other high-affinity iron binding proteins⁵⁵ (Fig.6d).”

9. 474-476: the fact that the samples come from healthy individuals does not necessarily mean these have not been exposed throughout their life to antibiotics and pathogenic bacteria that will enrich their gut microbiome with AB resistance & virulence. The conclusion based on that seems rather circumstantial. The fact that at least one individual have an *E.coli* isolate resistant to AB that is able to overcome iron limitation does not mean the drug selects for such strains. Without biological replication this may be considered an overstatement.

We thank the reviewer for the remark. We have observed a significant expansion of *Escherichia/Enterobacteriaceae* after entacapone supplementation in two additional individuals (Extended Data Fig. 2) that did not take part in the initial experiments. Although this is still a relatively small number of individuals analyzed, we believe this outcome is not purely circumstantial. Despite the presence of AMR and virulence genes in the *E. coli_D* MAG, we agree that the true pathogenic potential of this *E. coli* and other expanding *Enterobacteriaceae* spp. remain to be investigated – for this reason we used the wording “associated pathogenic potential”.

10. 543-553: how lowering of entacapone conc locally by bioaccumulation (which seems to be a feature of many genera) will impact the activity/abundance of the other members of the microbiome?

We could not find a way to measure a local concentration of the drug around the accumulator *in situ*, to determine how this impacts neighbouring species. However, we have used two concentrations of entacapone in our study, and we do see differences in the impact these have on the community, with 20 μ M causing a much lower impact than 1965 μ M (Fig.1 and Extended Data Fig.1). This suggests that lowering concentrations of entacapone over time due to progressive bioaccumulation of the drug would have a lower negative impact on the activity and likely also abundance of neighbouring bacteria. As we discuss in lines 617-622 (and below), the overall decrease in entacapone concentration due to bioaccumulation may explain the microbial growth of ENT-Hi samples between 6 and 24 hours, in contrast to the complete growth inhibition in the first 6 hours of ENT-Hi exposure.

“Here, we demonstrate that entacapone is also bioaccumulated, with Bacteroides, Phocaeicola, Streptococcus and Escherichia spp. being able to accumulate sufficient amounts of entacapone for photothermal detection (Fig.3). This accumulation likely results in depletion of entacapone in the surrounding environment over time, thus possibly explaining the slight alleviation of entacapone’s inhibitory effect between 6 and 24 hours of incubation, when compared to the first 6 hours (Fig.1b).”

Reviewer #2 (Remarks to the Author):

In their study, Pereira et al. explore the effects of two drugs targeting the nervous system on the signature and metabolic activity of the human gut microbiome *ex vivo*. Employing 16S rRNA sequencing, FACS-based cell counting, and long-read metagenomics, the researchers assess the drugs' impact on abundances across six healthy donors. Through stable isotope probing and single-cell chemical imaging, they pinpoint metabolic adaptations to drug exposure. Specifically, for entacapone, an antiparkinson's drug primarily excreted in feces, the authors demonstrate its bioaccumulation by certain taxa. Crucially, the authors elucidate the growth inhibitory effect of entacapone, revealing its ability to sequester iron, thereby limiting microbial growth. Whether these mechanisms are also at play in the intestines of patients remains to be established.

For me personally, the "mechanism of action" of entacapone, the inhibition of bacterial growth through iron binding, is the eye-opening, surprising novelty of this study.

The study is presented in a clear and accessible manner, with figures aiding comprehension. The methods employed are cutting-edge and apt for addressing the research questions, and the conclusions are mostly substantiated by the presented data. The results hold immediate interest for the microbiology community and extend to pharmacologists studying nervous system-targeting compounds.

While the study is generally suitable for publication in *Nature Microbiology*, several points should be addressed or clarified:

The authors thank the reviewer for the compliments on our work and manuscript.

Major Concerns:

- Fig. 1 c/d. I assume that the data plotted here are the combined results across all six donors and all three replicates. How have these results been combined? Is this the mean, median or something else shown here? Is the observation described here true for each individual or were there differences between individuals? The same question applies to Figures 2, 3 and 4.

The data shown is from triplicates per incubation condition. Incubations were established from a faecal slurry that combined samples from six donors. Interindividual variability in microbiome response to drugs has been reported previously and was not the main focus of our study. Our rationale for mixing samples from 6 healthy donors was to assess the effect of drugs on a microbial community with greater microbial diversity than that of a single individual, to capture the full potential of these drugs to remodel the microbiome. We do observe a reproducible effect of the drugs on microbial growth and community composition of this faecal mix when using two different media: sM9 and BHI (Figures 1 and Extended Data Fig. 1). We have further demonstrated replication of the effect of entacapone when we tested whether replenishing iron levels could rescue the effect of entacapone, as shown in Figure 4. For this experiment of Figure 4, we also used a mix of faecal samples (in triplicates, as well), but the experiment was conducted independently (with new samples) of the original experiment in Figure 1. In the experiment the results of which are shown in Fig.4, we observe a decrease in the abundance of key microbial taxa such as *Bacteroides* or *Clostridium* species upon entacapone treatment and an expansion of *Escherichia* spp. and *Streptococcus*, in agreement with results shown in Fig.1 and Extended Data Fig. 1.

To confirm that the observed effect was not only present when the community was a combination of samples from different donors, we performed additional experiments assessing the effect of supplementing entacapone to individual samples from different donors, more precisely in faecal samples from three donors that did not take part in the initial experiments (triplicates were set up for each condition tested per sample). Results demonstrate that entacapone supplementation significantly reduces microbial loads in samples from all three donors tested individually (Extended Data Fig.2a). Please see our response to point 2, reviewer 1. We further show that entacapone dramatically remodels the microbiome community of all three donors and promotes an expansion of *Escherichia-Shigella* and other *Enterobacteriaceae* species, similarly to what we show in Fig.1 and Fig.4. This expansion is subtle in Donor 1, but quite pronounced in Donors 2 and 3. These results are shown in Extended Data Fig.2 and Sup. Table 6 of the revised manuscript. These replications and independent experiments support the main findings of the manuscript.

- The idea that *E. coli_D* enterobactin production allows it to grow under iron-limiting conditions induced by ENT-Hi is intriguing, but has not been convincingly demonstrated. So far, there is only an association between the presence of the enterobactin biosynthetic gene cluster and a bloom in the presence of high entacapone concentrations, not a causal relationship. The fact that the *E.coli_D* isolate can grow on iron-free minimal medium and is unaffected by ENT-Hi is not sufficient proof. It should be shown that in the absence of the enterobactin biosynthetic gene cluster (natural isolate or genetically engineered knockout in any *E. coli* strain), growth and/or metabolic activity/iron requirements are altered in the presence of entacapone.

The authors thank the reviewer for this point and the experiments suggested. In response to this comment, we followed the growth of wild type *E. coli* K-12 BW25113 and isogenic *entB* and *fepA* deletion mutants in MOPS medium supplemented or not with entacapone (Fig.6c). EntB is an isochorismatase required for enterobactin synthesis and FepA is a TonB-dependent receptor for ferric:enterobactin complexes (Fig.6a). While all three strains grew equally well in MOPS medium, the growth of *entB* and *fepA* mutants was significantly reduced when entacapone (ENT-Hi) was added to the medium. The *entB* mutant shows residual growth, likely by using iron stored internally. *E. coli* with a *fepA* siderophore receptor mutant grows to higher levels than the *entB* mutant, possibly because other cell receptors such as Fiu enable uptake of enterobactin breakdown products (*i.e.* 2,3-dihydroxybenzoyl-L-serine (DHBS)) in complex with iron (Yang *et al.*, 2024). The growth defect of the *entB* mutant is rescued when commercial enterobactin is supplemented to the medium. As expected, enterobactin supplementation does not rescue the growth of the *fepA* deletion mutant, as FepA is necessary for the uptake of ferric:enterobactin complexes. These results strongly support that enterobactin synthesis and/or import is required for optimal growth in the presence of entacapone, and thus that the presence of entacapone drives iron limitation.

These results are now included in the main manuscript Fig.6c and below, and are described in lines 496-506:

“Similarly, ENT-Hi supplementation did not impact the growth of wild-type *E. coli* K12 strain BW25113, but it significantly reduced growth of a BW25113 deletion mutant deficient for enterobactin production (*entB* mutant) or enterobactin uptake (*fepA*, encoding a Ton-B enterobactin receptor) (Fig.6c). The *fepA* siderophore receptor mutant grows to higher levels than *entB* in the presence of entacapone, likely due to uptake of enterobactin breakdown products in complex with iron, which may enter the cell via other receptors⁵⁴. Supplementation of the growth medium with enterobactin rescued the entacapone-induced growth defect of *entB*, but not of the *fepA* receptor mutant (Fig.6c), as expected. These results strongly support that enterobactin synthesis and/or import is required for optimal growth in the presence of entacapone, and thus that the presence of entacapone drives iron limitation.”

Figure 6c. Growth of *E. coli* K12 BW25113 wild-type (wt), *entB* ($\Delta entB$) or *fepA* ($\Delta fepA$) mutants determined by optical density measurements of cultures supplemented or not with 1965 μM entacapone (ENT-Hi) and/or 2 μM of enterobactin. Data points represent the mean ($n=3$). Error bars represent the standard error of the mean. * $p<0.05$, ** $p<0.01$; *** $p<0.001$; unpaired two-sample t-test with “*E. coli* wt” used as the reference.

- Line 500 - 508: "Our results show that loxapine succinate exert an effect on a broader range of taxa in the context of the community than its effects on microbes grown in isolation". One explanation, as the authors mention, is indeed cross-sensitisation to the drug in the community context. However, other possibilities include media differences (sM9/BHI in this study vs. mGAM in Maier *et al.*, 2018) or differences between natural isolates at higher taxonomic levels, as in this study here, compared to type strains (in Maier *et al.*, 2018). As there are alternative explanations for their observations, the authors should also discuss / mention these possibilities.

We thank the reviewer for these points. We agree with the reviewer, and the sentence has been rephrased to include the options above as equally plausible explanations. Please see manuscript lines 574-577 and below:

“These differences may, however, not be entirely related to cross-sensitization, but instead could be caused by variations in growth conditions or in the genomic content of strains present in the faecal sample strains compared to the type strains tested in a previous study¹.”

Minor Concerns:

- Figure 2b: Please add the number of cells analysed per condition. Was this done for all six donors?

The authors apologize for the lack of information in Figure 2. We have analysed between 510 and 1090 cells, the number of cells analysed is now shown in Fig 2b and Supplementary Table 9. Single-cell measurements are presented in Supplementary Table 9. These samples originated from incubations of drugs with a mixture of samples from six individuals, set in triplicates per condition (see also our response above to point 1 of this reviewer).

- Line 313, figure legend 3a: It is unclear to me why the concentration here is 10 mM. Please either explain or delete.

We did use a concentration of 10 mM entacapone in DMSO only for Fig. 3 panel a and Extended Fig.6. We needed a concentrated solution of entacapone to acquire the absorption spectra of entacapone and photothermal signal traces and we chose 10 mM, for no particular reason. For all other panels in Fig.3, the concentration of entacapone used was 1965 μ M (ENT-Hi). This is now explained in the legend of Fig. 3.

- Line 318: Please correct: Single cell photothermal signal intensity distribution in samples shown in (b).

Apologies, this has been corrected, please see line 349.

- Figure 4: Which medium was used for the faecal incubations? BHI or sM9?

We have used sM9 medium, this is now indicated in the legend of Fig. 4, please see line 394.

- Figure 4b: How many independent replicates were tested? What does the error bar represent? SD, SE,...?

The authors would like to apologize for the lack of information in Figure 4. Data shown is from six independent growths for ENT-Hi and No drug samples, and three replicates for all other conditions. Error bars indicate standard error of the mean. This has been added to Fig.4 legend.

- Line 387: The given concentration of iron (31 μ M) refers to which medium?

This concentration refers to sM9 medium. This information is now added, please see line 411.

- Figure 5d: This could be due to biases in knowledge and databases, with better annotation for Enterobacteriaceae compared to other taxa. This should be discussed/mentioned in the text.

We agree with the reviewer. We have added this information to lines 544-551, and below:

“ENT-Hi drives an increase in the abundance of AMR and virulence within faecal microbiomes (Fig.6e). This increase is driven in great part by the increase in abundance of E. coli_D, whose genome encodes a total of 14 AMR and virulence genes, in addition to at least one siderophore production cluster, thus suggesting a pathogenic potential of this organism (Supplementary Table 17). While we cannot rule out the possibility that the increased AMR and virulence index is due to improved genomic characterization of E. coli compared to other taxa, these results reveal that ENT-Hi promotes the growth of iron scavenging organisms with an associated pathogenic potential.”

- Line 547: over time

This information is now corrected, line 615 of the revised version.

- Line 956: Why are monoculture growth assays performed in a different medium than faecal incubation assays? Would an increase in E. coli growth be observed with ENT-Hi in sM9 or BHI?

For monoculture growths, we chose to use defined media instead of undefined media such as BHI, so we could control iron levels in the medium. sM9 would be the preferred option to keep consistency across the study, but most *Bacteroides* and *Phocaeicola* (formerly *Bacteroides*) spp. including *B. thetaiotaomicron* and *P. dorei* do not grow well in sM9 medium when in isolation. This is due to a lack of essential nutrients such as vitamin B12, ammonia or hemin in sM9 (Varel and Bryant, 1974). These nutrients may be present in faecal samples in sufficient amounts to support *Bacteroides* growth in faecal slurries prepared in sM9 (Fig. 1c-e), but not in the sM9 medium alone. Therefore, to grow *B. thetaiotaomicron* (or *P. dorei*) in monoculture we

chose BMM (*Bacteroides* minimal medium; Bacic and Smith, 2008), a defined medium tailored for *Bacteroides* spp. and for which the concentration of iron is known and can be controlled (27.5 μ M).

E. coli does grow well in sM9. However, as *E. coli* can produce the siderophore enterobactin, we opted to use a MOPS-based medium, which is the medium of choice to induce iron limitation in *Escherichia coli* K-12 and evaluate its potential to scavenge iron via siderophores. Growth media derived from commercially available M9 salts, as is the case of sM9, are unsuitable for studies of iron-limited growth, likely through the contamination of the sodium phosphate or other components with traces of iron that siderophore-producers can easily scavenge (Southwell *et al.*, 2024). MOPS medium includes minimal levels of phosphates compared to sM9 or BMM and thus, reduces free iron concentration to growth-limiting levels, as demonstrated by Fig. 6b.

The reasons for the choice of media have now been added to materials and methods, please see lines 1120-1122 and 1135-1137.

Reviewer #3 (Remarks to the Author):

In this manuscript the authors evaluate the impact of two clinically relevant nervous system targeting drugs on commensal microbes. The authors report that these drugs affect microbiome composition and abundance, and these experiments were performed across two distinct media increasing the rigor of these observations. Through the application of an innovative SRS-FISH assay the authors determined nervous system-targeting drugs alter microbial metabolism without impacting microbial abundance. Using photothermal imaging approaches, the authors report that entacapone bioaccumulates in microbial cells. Finally, the authors report that entacapone chelates iron and induces iron starvation in whole microbiome populations and promotes growth of iron-scavenging *E. coli*. The authors put forward an exciting and innovative model but the presented data fall short of conclusively determining the mechanism by which entacapone inhibits microbial growth. I have the following suggestions to strengthen the conclusions and improve the readability of the manuscript.

We thank the reviewer for the constructive feedback and questions. We provide clarifications below based on new experimental data that are included in the revised manuscript. To increase the readability, we respond to the different sections separately as below.

1. The inclusion of the CAS assay is informative to evaluate the ability of entacapone to bind ferric iron. However, due to the focus of the paper on the gut microbiota, the binding of entacapone to ferrous iron should be evaluated, the predicted affinity is not convincing without experimental data to support this prediction.

We would like to thank the reviewer for the constructive comments, which were very helpful to better understand the interaction between entacapone and iron in our incubations. We started our additional experiments by determining the ability of entacapone to complex Fe(II), as suggested by the reviewer. We used the ferrozine chromogenic method, in which ferrozine reacts with free divalent Fe to form a stable magenta complex species with a maximum absorbance at 562 nm (Stookey, 1970), to follow the presence of Fe(II) in our incubations with entacapone. This revealed that a significant part of Fe(II) from FeSO₄ remained free to form complexes with ferrozine following incubation with entacapone (in DMSO:degassed water 1:1). This contrasted with results obtained in parallel using a known Fe(II) chelating agent - Na₂H₂EDTA - which was able to chelate nearly 100% of all Fe(II) present. These results indicate limited ability of entacapone to chelate Fe(II) in our conditions, and are shown below and in the new Extended data Fig. 9a (also shown below). We then proceeded to determine how much iron in our incubation set up was in the Fe(II) form. We found that despite the anaerobic conditions, most of the FeSO₄ added to the sM9 medium was quickly oxidised to Fe(III) (Extended Data Fig.9c). We believe the presence of phosphates in the medium catalyses the oxidation of Fe(II) to Fe(III), as documented by others (Mitra and Matthews, 1985). Phosphates are indeed present in the millimolar range (10-35 mM) in the minimal media used. We also measured the levels of Fe(II)-ferrozine when FeSO₄ was incubated with degassed DMSO or water. Pure DMSO was used to dissolve FeSO₄ and entacapone powder to generate the ENT-Hi:Fe(II) complex (or Fe(II) pre-loaded ENT-Hi) in the rescue experiments, as shown in Fig.4b-e. Limited absorbance from a Fe(II)-ferrozine complex was detected after FeSO₄ salts were dissolved in pure DMSO (Extended Data Fig. 9b). In contrast, when degassed water was used instead of DMSO, a steady increase in absorbance was observed with increasing ferrozine concentration (Extended Data Fig. 9b). From these data we concluded that pure DMSO oxidizes Fe(II) to Fe(III). Concentrations of DMSO of 50% or below are unlikely to oxidize Fe(II), as these conditions were used to dissolve entacapone in Extended Data Fig. 9a, and we observed that nearly all Fe(II) added remained free to complex ferrozine. Thus, both pre-complexation of ENT-Hi with FeCl₃ or FeSO₄ resulted in

Fe(III)-loaded entacapone. These results are now reflected in the main manuscript, and we have also modified the legend of Fig. 4b,c accordingly.

In conclusion, these new results suggest that a significant part of iron in our incubations was present as Fe(III) – ferric iron - even when supplied in the form of FeSO₄, and is therefore susceptible to chelation by entacapone. Both ferric and ferrous iron are expected to be present in the colon (Kortman *et al.*, 2014). Most iron is expected to enter the large intestine in Fe(III) form, as the increase in pH in the duodenum favours the oxidation of ferrous iron in the presence of oxygen (Jacobs and Miles, 1969). *E. coli* mutants affected in the production or uptake of catechol-type siderophores (that exclusively bind ferric iron) showed impaired colonisation of the mouse intestine compared to wild-type counterparts (Pi *et al.*, 2012), supporting the idea that ferric iron is an important iron source for the gut microbiota. Consequently, ferric iron chelation by entacapone is of physiological relevance in the colon. We now include this information in manuscript lines 641-644:

“Most iron is expected to enter the large intestine in Fe(III) form, as the increase in pH in the duodenum favors the oxidation of Fe(II) in the presence of oxygen⁶⁷. In fact, ferric iron is an important source of iron for the gut microbiota⁶⁸, and we expect Fe(III) chelation by entacapone to impact microbiome homeostasis in the colon.”

Extended Data Fig.9. Determination of Fe(II) concentrations by spectrophotometric measurements of the Fe(II)–ferrozine complex. The ferrozine reagent reacts with divalent Fe to form a stable magenta complex species with a maximum absorbance at 562 nm. **a.** Percentage of total Fe(II) in anaerobic solutions of FeSO₄ containing increasing concentrations of Na₂H₂EDTA (a known Fe(II) chelator) or entacapone, as determined using the ferrozine method. **b.** Absorbance at 562 nm, indicating the presence of ferrozine-Fe(II) complexes in solutions of FeSO₄ in degassed DMSO (100%) or degassed double distilled water, following reaction with increasing concentrations of ferrozine. **c.** Fe(II) concentrations in water or sM9 solutions of FeSO₄ (1mM FeSO₄). FeSO₄ was dissolved in degassed water or sM9 and incubated for 0, 1, 5, 10, 15 and 30 minutes, as indicated, before the ferrozine reagent was added. Concentrations of Fe(II) were determined using a calibration curve obtained by

measuring absorbance at 562 nm in solutions of known Fe(II) concentration (see Methods) supplemented with 2 mM ferrozine. In *a,b,c* error bars denote standard deviation from triplicates per condition.

2. The presented data do not distinguish between the possibility that entacapone inhibits microbial growth through iron sequestration vs. the possibility that iron binding to entacapone inhibits its antibacterial activity in a manner independent of microbial iron starvation. In this regard, the manuscript falls short of convincingly supporting the proposed model. To clearly show that entacapone inhibits microbial growth through nutrient iron restriction, a series of additional experiments are required. First, the authors should test whether entacapone exposure induces a gene expression profile in microbes of interest that is consistent with iron starvation

In the revised version of the manuscript, we demonstrate that *E. coli* K-12 BW25113 mutants for enterobactin siderophore production ($\Delta entB$) or uptake ($\Delta fepA$) have impaired growth in the presence of entacapone compared to the wild-type strain (new Fig. 6c). The growth defect of the *entB* mutant is fully rescued by external addition of enterobactin. These results are explained in detail in our response to reviewer 2 point 2. We believe this data strongly supports our conclusion that entacapone reduces bacterial growth by inducing iron limitation.

We would also like to highlight that we only detect a weak negative correlation between entacapone bioaccumulation and single-cell C-D activity (Fig. 3d), suggesting no strong inhibitory/toxicity effect of intracellular entacapone on microbial activity.

Second, the authors should measure iron levels within entacapone exposed bacteria.

Using nanoscale secondary ion mass spectrometry (NanoSIMS), we evaluated the relative iron content of cells from faecal incubations and from a pure culture of *Phocaeicola dorei* (a confirmed entacapone bioaccumulator, Extended Data Fig.8) exposed or not to ENT-Hi. These results are now included in a new figure - Fig. 5 - and below. In both cases, cells exposed to entacapone display significantly higher levels of iron than non-exposed cells. These data corroborate our results showing that ENT-Hi complexed with iron bioaccumulates in cells (Extended Data Fig. 8e,f), strongly suggesting that ENT-Hi-Fe(III) complexes are stable and reach the cellular space. We postulate microbial cells lack enzymes that enable the release of iron from these ENT-Hi-Fe(III) complexes, resulting in iron starvation. This is now discussed in the manuscript, lines 459-463.

Figure 5. Entacapone-exposed cells display higher ^{56}Fe signal intensity. **a, b.** Nanoscale secondary ion mass spectrometry (NanoSIMS) overlay images of the $^{56}\text{Fe}^+$ (color scale) and $^{12}\text{C}^+$ (grey scale) signal intensities of a faecal sample incubated for 6 hours with no drug (**a**) or ENT-Hi (**b**). Scale bar, 10 μm . **c.** Quantification of $^{56}\text{Fe}^+ / ^{12}\text{C}^+$ intensity ratios in individual cells present in faecal samples in sM9 medium. **d, e.** NanoSIMS overlay images of the $^{56}\text{Fe}^+$ (color scale) and $^{12}\text{C}^+$ (grey scale) signal intensities in *Phocaeicola dorei* cells incubated for 24 hours with no drug (**d**) or ENT-Hi (**e**). For individual ^{12}C images used for merging, the minimum and maximum intensities (grey scale) are the following: 1-16 counts/pixel (**a**), 2-25 counts/pixel (**b**), 4-80 counts/pixel (**d**) and 0-25

counts/pixel (e). For individual ^{56}Fe images used for merging, the minimum and maximum intensities (color scale) are the following: 3-10 counts/pixel (a), 3-35 counts/pixel (b), 7-17 counts/pixel (d) and 4-80 counts/pixel (e). Scale bar, 10 μm . f. Quantification of $^{56}\text{Fe}^+ / ^{12}\text{C}^+$ intensity ratios in individual *P. doerei* cells. In c, f, p-values were calculated using an unpaired two-sample Wilcoxon test. Each dot represents a cell and boxes represent the median, first and third quartile. Whiskers extend to the highest and lowest values that are within one and a half times the interquartile range.

Finally, the authors should test if microbial strains defective in siderophore production (siderophore synthesis mutants) are more susceptible to entacapone than isogenic wildtype bacteria.

We thank the reviewer for this suggestion. We performed the suggested experiment, which demonstrated that *E. coli* mutants for enterobactin siderophore production (ΔentB) or uptake (ΔfepA) have impaired growth in the presence of entacapone compared to the wild-type strain (new Fig. 6c). Please see our detailed response to reviewer 2 point 2.

3. If the authors convincingly show that entacapone inhibits microbial growth through iron restriction, they should determine if entacapone bioaccumulation within microbes occurs through microbial siderophore receptors as this would significantly increase the impact of the work.

We have measured entacapone accumulation in *E. coli* K-12 BW25113 wild-type and *fepA* mutant cells. FepA is a TonB-dependent receptor for ferric:enterobactin complexes, and this would enable us to determine if entacapone bioaccumulation occurs through microbial siderophore receptors. However, we could not detect enough levels of entacapone bioaccumulation using photothermal imaging (PT) in the wild-type *E. coli* K-12 BW25113, nor in the *fepA* mutant, as shown below (Figure R1, lack of PT signal in green). Our results from Fig.6c indicate that FepA is functional in wild-type BW25113 (Fig.6c). Thus, the lack of entacapone signal within these cells indicates that other receptors/transporters may be required for entacapone bioaccumulation, or that intracellular entacapone is efficiently exported out of the cell in this strain, resulting in little or no accumulation.

Figure R1. Entacapone photothermal signal (PT, in green) and transmission (grey) images of *E. coli* K-12 BW25113 wild type and *fepA* mutant cells exposed (middle and right columns) or not (left column) to ENT-Hi for 24 hours. Scale bar 8 μm .

References:

Bacic MK, Smith CJ. 2008. Laboratory Maintenance and Cultivation of Bacteroides Species. 1. *Current Protocols in Microbiology* Chapter 13: Unit 13C.1.

Bearden SW, Staggs TM, Perry RD. 1998. An ABC Transporter System of Yersinia pestis Allows Utilization of Chelated Iron by Escherichia coliSAB11. *Journal of Bacteriology* 180:1135–1147.

Garber AI, Neelson KH, Okamoto A, McAllister SM, Chan CS, Barco RA, Merino N. 2020. FeGenie: A Comprehensive Tool for the Identification of Iron Genes and Iron Gene Neighborhoods in Genome and Metagenome Assemblies. *Frontiers in Microbiology* 11:37.

Gasperotti A, Brameyer S, Fabiani F, Jung K. 2020. Phenotypic heterogeneity of microbial populations under nutrient limitation. *Current Opinion in Biotechnology* 62:160–167.

Hall AB, Yassour M, Sauk J, Garner A, Jiang X, Arthur T, Lagoudas GK, Vatanen T, Fornelos N, Wilson R, Bertha M, Cohen M, Garber J, Khalili H, Gevers D, Ananthakrishnan AN, Kugathasan S, Lander ES, Blainey

- P, Vlamakis H, Xavier RJ, Huttenhower C. 2017. A novel Ruminococcus gnavus clade enriched in inflammatory bowel disease patients. *Genome Medicine* 9:103.
- Hua S. 2020. Advances in Oral Drug Delivery for Regional Targeting in the Gastrointestinal Tract - Influence of Physiological, Pathophysiological and Pharmaceutical Factors. *Frontiers in Pharmacology* 11:524.
- Jacobs A, Miles PM. 1969. Intraluminal transport of iron from stomach to small-intestinal mucosa. *British Medical Journal* 4:778–781.
- Jin Z, Ng A, Maurice CF, Juncker D. 2022. The Mini Colon Model: a benchtop multi-bioreactor system to investigate the gut microbiome. *Gut Microbes* 14:2096993.
- Keränen T, Gordin A, Karlsson M, Korpela K, Pentikäinen PJ, Rita H, Schultz E, Seppälä L, Wikberg T. 1994. Inhibition of soluble catechol-O-methyltransferase and single-dose pharmacokinetics after oral and intravenous administration of entacapone. *European Journal of Clinical Pharmacology* 46:151–157.
- Klünemann M, Andrejev S, Blasche S, Mateus A, Phapale P, Devendran S, Vappiani J, Simon B, Scott TA, Kafkia E, Konstantinidis D, Zirngibl K, Mastroilli E, Banzhaf M, Mackmull M-T, Hövelmann F, Nesme L, Brochado AR, Maier L, Bock T, Periwal V, Kumar M, Kim Y, Tramontano M, Schultz C, Beck M, Hennig J, Zimmermann M, Sévin DC, Cabreiro F, Savitski MM, Bork P, Typas A, Patil KR. 2021. Bioaccumulation of therapeutic drugs by human gut bacteria. *Nature* 597:533–538.
- Kortman GAM, Raffatellu M, Swinkels DW, Tjalsma H. 2014. Nutritional iron turned inside out: intestinal stress from a gut microbial perspective. *FEMS Microbiology Reviews* 38:1202–1234.
- Luo JP, Vashishtha SC, Hawes EM, McKay G, Midha KK, Fang J. 2011. In vitro identification of the human cytochrome p450 enzymes involved in the oxidative metabolism of loxapine. *Biopharmaceutics & Drug Disposition* 32:398–407.
- Maier L, Pruteanu M, Kuhn M, Zeller G, Telzerow A, Anderson EE, Brochado AR, Fernandez KC, Dose H, Mori H, Patil KR, Bork P, Typas A. 2018. Extensive impact of non-antibiotic drugs on human gut bacteria. 7698. *Nature* 555:623–628.
- Maini Rekdal V, Bess EN, Bisanz JE, Turnbaugh PJ, Balskus EP. 2019. Discovery and inhibition of an interspecies gut bacterial pathway for Levodopa metabolism. *Science* 364:eaau6323.
- Mitra AK, Matthews ML. 1985. Effects of pH and phosphate on the oxidation of iron in aqueous solution. *International Journal of Pharmaceutics* 23:185–193.
- Pi H, Jones SA, Mercer LE, Meador JP, Caughron JE, Jordan L, Newton SM, Conway T, Klebba PE. 2012. Role of Catecholate Siderophores in Gram-Negative Bacterial Colonization of the Mouse Gut. *PLOS ONE* 7:e50020.
- Romano S, Savva GM, Bedarf JR, Charles IG, Hildebrand F, Narbad A. 2021. Meta-analysis of the Parkinson's disease gut microbiome suggests alterations linked to intestinal inflammation. *npj Parkinsons Dis* 7:1–13.
- Simpson GM, Cooper TB, Lee JH, Young MA. 1978. Clinical and plasma level characteristics of intramuscular and oral loxapine. *Psychopharmacology* 56:225–232.
- Southwell JW, Wilson KS, Thomas GH, Duhme-Klair A-K. 2024. Enhancement of growth media for extreme iron limitation in Escherichia coli. *Access Microbiology* 6:000735.v4.
- Stookey LL. 1970. Ferrozine---a new spectrophotometric reagent for iron. *Analytical Chemistry* 42:779–781.
- Varel VH, Bryant MP. 1974. Nutritional features of Bacteroides fragilis subsp. fragilis. *Applied Microbiology* 28:251–257.
- Wikberg T, Vuorela A, Ottoila P, Taskinen J. 1993. Identification of major metabolites of the catechol-O-methyltransferase inhibitor entacapone in rats and humans. *Drug Metabolism and Disposition* 21:81–92.
- Yang T, Zou Y, Ng HL, Kumar A, Newton SM, Klebba PE. 2024. Specificity and mechanism of TonB-dependent ferric catecholate uptake by Fiu. *Frontiers in Microbiology* 15:1355253.

Yu JSL, Correia-Melo C, Zorrilla F, Herrera-Dominguez L, Wu MY, Hartl J, Campbell K, Blasche S, Kreidl M, Egger A-S, Messner CB, Demichev V, Freiwald A, Müllereder M, Howell M, Berman J, Patil KR, Alam MT, Ralser M. 2022. Microbial communities form rich extracellular metabolomes that foster metabolic interactions and promote drug tolerance. *Nature Microbiology* 7:542–555.

Zhu W, Winter MG, Spiga L, Hughes ER, Chanin R, Mulgaonkar A, Pennington J, Maas M, Behrendt CL, Kim J, Sun X, Beiting DP, Hooper LV, Winter SE. 2020. Xenosiderophore Utilization Promotes *Bacteroides thetaiotaomicron* Resilience during Colitis. *Cell Host & Microbe* 27:376-388.e8.

Zimmermann M, Zimmermann-Kogadeeva M, Wegmann R, Goodman AL. 2019. Mapping human microbiome drug metabolism by gut bacteria and their genes. 7762. *Nature* 570:462–467.

We would like to thank all the reviewers for their constructive feedback, which has helped us to improve the manuscript.

Reviewer #1:

Remarks to the Author:

Upon careful review of this revised version of the manuscript by Pereira et al., I consider that all my major concerns regarding i) the biotransformation of the entacapone and loxapine that may occur post-drug absorption; ii) biological replication iii) relevance of the target group considering that only healthy individuals were considered in the study; IV) significance considering the short timeframe of maximum 24h have been addressed and answered, either by plausible explanations supported by references or by addition of supplementary results and profs.

I consider the study excellently suited for the publication in Nature Microbiology. I would like to acknowledge the authors efforts to improve the initial work by considering in detail all reviewers comments and by carried out several additional experiments during revision.

We thank the reviewer for the positive feedback. We are pleased to see that all of the reviewer's concerns have been successfully addressed in the revised version of the manuscript.

Reviewer #2:

Remarks to the Author:

The authors have done an exceptional job addressing my feedback. The revised manuscript is well-suited for publication in Nature Microbiology, and I eagerly await its publication.

We thank the reviewer for the positive feedback. We are pleased to see that all of the reviewer's concerns have been successfully addressed in the revised version of the manuscript.

Reviewer #3:

Remarks to the Author:

The authors performed a significant amount of new experimentation to address my prior concerns. I think the inclusion of these new data strengthens the conclusions and supports the authors' model. I have no further suggestions for improvement. This is an exciting paper that will be of broad significance.

We thank the reviewer for the positive feedback. We are pleased to see that all of the reviewer's concerns have been successfully addressed in the revised version of the manuscript.